# Long-chain ceramides are cell non-autonomous signals linking lipotoxicity to endoplasmic reticulum stress in skeletal muscle

Ben D. McNally[1], Dean F. Ashley [1], Lea Hänschke [2], Hélène N. Daou[3,8], Nicole T. Watt[3,8], Steven A. Murfitt[1], Amanda D. V. MacCannell[3], Anna Whitehead [3], T. Scott Bowen [4], Francis W. B. Sanders [1], Michele Vacca[1,5], Klaus K. Witte [3], Graeme R. Davies[6], Reinhard Bauer[2], Julian L. Griffin [1,7] & Lee D. Roberts [3✉]

The endoplasmic reticulum (ER) regulates cellular protein and lipid biosynthesis. ER dysfunction leads to protein misfolding and the unfolded protein response (UPR), which limits protein synthesis to prevent cytotoxicity. Chronic ER stress in skeletal muscle is a unifying mechanism linking lipotoxicity to metabolic disease. Unidentified signals from cells undergoing ER stress propagate paracrine and systemic UPR activation. Here, we induce ER stress and lipotoxicity in myotubes. We observe ER stress-inducing lipid cell non-autonomous signal(s). Lipidomics identifies that palmitate-induced cell stress induces long-chain ceramide 40:1 and 42:1 secretion. Ceramide synthesis through the ceramide synthase 2 de novo pathway is regulated by UPR kinase Perk. Inactivation of *CerS2* in mice reduces systemic and muscle ceramide signals and muscle UPR activation. The ceramides are packaged into extracellular vesicles, secreted and induce UPR activation in naïve myotubes through dihydroceramide accumulation. This study furthers our understanding of ER stress by identifying UPR-inducing cell non-autonomous signals.

[1] Department of Biochemistry, University of Cambridge, Cambridge CB2 1GA, UK. [2] Life & Medical Sciences Institute (LIMES) Development, Genetics & Molecular Physiology Unit, University of Bonn, Carl-Troll-Straße, 31, 53115 Bonn, Germany. [3] School of Medicine, University of Leeds, Leeds LS2 9JT, UK. [4] Faculty of Biological Sciences, University of Leeds, Leeds LS2 9JT, UK. [5] Clinica Medica "Frugoni", Interdisciplinar Department of Medicine, University of Bari "Aldo Moro", Bari, Italy. [6] Bioscience Metabolism, Research and Early Development, Cardiovascular, Renal and Metabolism, BioPharmaceuticals R&D, AstraZeneca, Cambridge, UK. [7] Department of Metabolism, Digestion and Reproduction, Imperial College London, London, UK. [8] These authors contributed equally: Hélène N. Daou, Nicole T. Watt. ✉email: L.D.Roberts@leeds.ac.uk

The endoplasmic reticulum (ER) is a cellular organelle with a key role in both protein synthesis and folding, and lipid biosynthesis. Disruption to ER function results in organelle stress and the accumulation of misfolded proteins. Unchecked accretion of unfolded proteins can result in cell death. ER stress has been implicated in a wide range of pathologies including aging, certain cancers and metabolic diseases including dyslipidaemia, obesity and type 2 diabetes (T2D)[1]. Metabolic diseases including obesity and T2D are characterised by elevated plasma concentrations of saturated fatty acids, particularly palmitate[2,3]. These lipid species are thought to induce metabolic dysfunction in insulin-sensitive tissues such as liver, adipose tissue and skeletal muscle through effects termed lipotoxicity[4–6]. Skeletal muscle is a key regulator of systemic metabolic homoeostasis[7]. Recently, ER stress has emerged as a potential unifying mechanism linking lipotoxicity to metabolic dysfunction in metabolic disease[8]. Lipotoxicity-induced ER stress in skeletal muscle causes metabolic dysfunction and contributes to the development of metabolic disease[9–13]. However, the mechanisms linking palmitate with the induction of ER stress remain unclear.

The cell has adaptive responses to maintain protein homoeostasis and survival during ER stress. ER stress triggers the unfolded protein response (UPR), a protective signalling cascade. The UPR is composed of three arms, mediated by the kinases protein kinase R-like endoplasmic reticulum kinase (Perk; encoded by *EIF2AK3*) and inositol-requiring enzyme 1 (Ire1; encoded by *ERN1*), and the transmembrane transcription factor, activating transcription factor 6 (Atf6). Signalling through these proteins increases cellular protein chaperones and disulphide isomerases, activates protein degradation pathways, and inhibits protein translation[14]. These protective responses reduce protein load on the ER and improve protein folding. The intracellular mechanisms of ER stress and regulation of the UPR are well understood. However, recent studies suggest that UPR signalling can be propagated in both a paracrine manner and systemically by cell non-autonomous signals[15,16]. The nature of these extracellular paracrine and endocrine signals remains unknown.

In this study, we hypothesise that cell non-autonomous signalling may be important in the communication of lipotoxicity-induced ER stress in skeletal muscle. We demonstrate the presence of lipotoxicity-induced long-chain ceramides secreted in response to lipotoxicity, which function as a paracrine signal to activate the UPR in myotubes. We describe the mechanism through which the long-chain ceramide signal is linked to UPR activation, synthesised, transported extracellularly and initiates UPR activation. This study identifies the nature of a cell non-autonomous signal capable of propagating ER stress. We develop our understanding of the control of UPR signalling beyond a cell-autonomous state, with clear implications in the development and propagation of metabolic disease.

## Results

**Myotubes secrete a non-autonomous ER stress-inducing lipid signal in response to chronic lipotoxicity.** To confirm UPR induction in response to chronic lipotoxicity in muscle cells, murine C2C12 myotubes were exposed to physiological concentrations of palmitate for the final 6 days of an 8 day differentiation protocol[17,18]. Cells were treated with either 100 or 200 μM palmitate conjugated to fatty-acid-free bovine serum albumin (BSA), or BSA vehicle control. Concentrations of 100 μM and 200 μM palmitate reflect the physiological palmitate plasma concentration range in mice (100–300 μM)[19,20], and chronic treatment (6 days) models continual palmitate exposure in vivo. Palmitate-treatment increased expression of the UPR genes activating transcription factor 3 (*Atf3*), activating

transcription factor 4 (*Atf4*), Heat Shock Protein Family A (Hsp70) Member 5 (*Hspa5*) and ER Degradation Enhancing Alpha-Mannosidase Like Protein 1 (*Edem1*) (Supplementary Fig. 1a).

To assess a role for cell non-autonomous secreted signals during lipotoxicity-induced cell stress, C2C12 myotubes were exposed to 200 μM palmitate or the BSA vehicle during differentiation to induce ER stress. Culture media was then changed to serum-free and palmitate-free media for 24 h. Compounds accumulating in the media would therefore originate from the myotubes. Lipidomic profiling confirmed that the conditioned media was exogenous palmitate-free (Supplementary Fig. 1b). This conditioned media was transferred to naïve differentiated C2C12 myotubes (Fig. 1a). Conditioned media from palmitate-treated C2C12 myotubes increased expression of UPR genes *Atf4*, *Hspa5* and *Edem1* (Fig. 1b). To determine the relevance of this phenotype in a human model, primary human skeletal muscle cells (HSkMCs) were exposed to 50 or 100 μM palmitate, or the BSA vehicle during a 6 day differentiation. Concentrations of 50 μM and 100 μM palmitate reflect the physiological palmitate plasma concentration range in humans (50–150 μM)[17,18,21], and chronic treatment (6 days) models continual palmitate exposure in vivo. Palmitate-treatment increased the expression of UPR genes *ATF3*, *ATF4*, *HSPA5* and *EDEM1* (Supplementary Fig. 1c). Conditioned serum- and palmitate-free media collected from HSkMCs treated with 100 μM palmitate or BSA was transferred to naïve HSkMCs. Conditioned media from palmitate-treated HSkMCs increased UPR gene expression in human myocytes (Fig. 1c). These data suggest the presence of cell non-autonomous UPR-inducing signal secreted by both murine and human myocytes in response to chronic lipotoxicity.

Next we sought to determine the nature of the UPR-inducing secreted signal. Conditioned media was boiled to denature proteins prior to transfer to naïve myotubes. Boiled conditioned media from palmitate-treated myotubes retained UPR-inducing properties, suggesting that there are non-protein secreted signal(s) (Fig. 1d). The UPR-inducing signal was preserved in boiled media conditioned from palmitate-treated cells with no evidence of altered UPR-inducing activity in boiled control media. However, boiling denaturation has the potential to destabilise small molecules. Therefore, to further characterise the physicochemical nature of the UPR-inducing signal, molecular weight cut-off dialysis of 1 kDa was used to dialyse conditioned media from control and palmitate-treated myocytes. Dialysate (containing the dialysed molecules of 1 kDa and below) was dried down under $N_2$, reconstituted in serum-free media and transferred to naïve myocytes. The UPR-inducing signal in conditioned media from palmitate-treated myocytes was observed in the dialysate reconstituted media (Supplementary Fig. 1d). These data suggest the signal(s) is a small molecule of 1 kDa or less.

We hypothesised that the UPR-inducing signal may be a bioactive lipid. The lipid-containing organic component and metabolite containing aqueous component of the conditioned media from both control and palmitate-treated myocytes was extracted using protein-denaturing organic-aqueous solvent partition. The lipids, isolated from the organic fractions of conditioned media from both control and palmitate-treated myocytes, were reconstituted in serum-free media and transferred on to naïve differentiated myotubes. The organic lipid fraction of media from palmitate-treated myotubes increased the expression of UPR target genes compared to the organic lipid fraction of media from control myotubes (Fig. 1e). Next, we examined whether aqueous soluble metabolites may represent additional UPR-inducing signals. The metabolites isolated from the aqueous

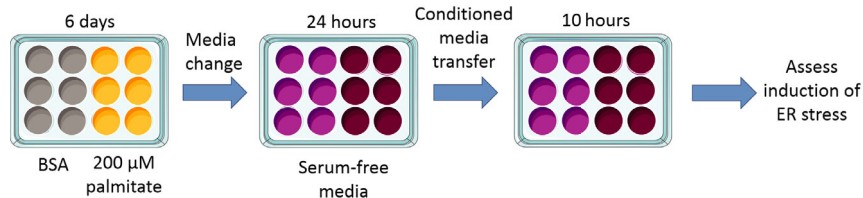

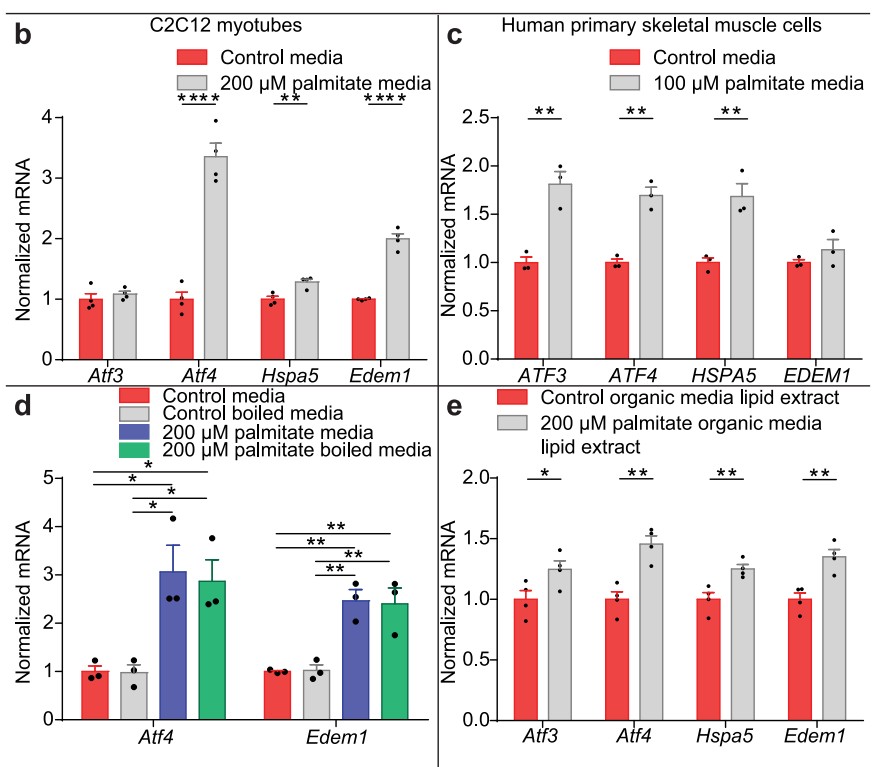

**Fig. 1 Myotubes secrete a cell non-autonomous unfolded protein response (UPR) -inducing lipid signal in response to lipotoxicity. a** C2C12 myotubes were treated with 200 μM palmitate or bovine serum albumin (BSA) vehicle, serum-free media was conditioned on these cells (24 h) and transferred to naïve myocytes. **b** Activating transcription factor 3 (*Atf3*), activating transcription factor 4 (*Atf4*), Heat Shock Protein Family A (Hsp70) Member 5 (*Hspa5*) and ER Degradation Enhancing Alpha-Mannosidase Like Protein 1 (*Edem1*) UPR gene expression in C2C12 myotubes receiving conditioned media from myotubes treated with 200 μM palmitate (grey) or BSA (red) (*n* = 4; two-tailed Student's *t* test; *Atf4 P* = 0.00009, *Hspa5 P* = 0.005, *Edem1 P* = 0.000024). **c** UPR gene expression in human skeletal muscle cells (HSkMCs) receiving conditioned media from HSkMCs treated with 100 μM palmitate (grey) or BSA (red) (*n* = 3, two-tailed Student's *t* test; *ATF3 P* = 0.005, *ATF4 P* = 0.0017, *Hspa5 P* = 0.0088). **d** UPR gene expression in myotubes receiving boiled media conditioned on C2C12 myotubes treated with 200 μM palmitate or BSA (control = red; control boiled = grey; 200 μM palmitate = blue; 200 μM palmitate boiled = green; *n* = 3; one-way ANOVA; control media vs 200 μM palmitate media, *Atf3 P* = 0.017, *Atf4* P = 0.005; control vs 200 μM palmitate boiled, *Atf3 P* = 0.03, *Atf4 P* = 0.0065; control boiled vs 200 μM palmitate, *Atf3 P* = 0.016, *Atf4 P* = 0.005; control boiled vs 200 μM palmitate boiled, *Atf3 P* = 0.027, *Atf4 P* = 0.007). **e** UPR gene expression in myotubes receiving reconstituted lipid extract isolated from media conditioned on C2C12 myotubes treated with 200 μM palmitate (grey) or BSA (red) (*n* = 4; two-tailed Student's *t* test: *Atf3 P* = 0.05, *Atf4 P* = 0.0026, *Hspa5 P* = 0.0097, *Edem1 P* = 0.0048). *P ≤ 0.05, **P < 0.01, ***P < 0.001, ****P < 0.0001. Data are expressed as mean ± SEM with individual data points. Source data are provided as a Source Data file.

fractions of conditioned media from both control and palmitate-treated myocytes were reconstituted in serum-free media and transferred on to naïve differentiated myotubes. The aqueous metabolite fraction of media from palmitate-treated myotubes also increased the expression of UPR target genes compared to the aqueous fraction of media from control myotubes (Supplementary Fig. 1e). These data suggest that both a bioactive lipid(s) and metabolite(s) may function as cell non-autonomous UPR-inducing secreted signals.

**Lipotoxicity-induced UPR-activating long-chain ceramide secretion from myotubes.** To identify the cell non-autonomous UPR-inducing lipid species we performed lipidomic profiling of serum and palmitate-free conditioned media from palmitate-treated myotubes. Multivariate analysis of the media lipidomic data identified lipid species discriminating 100 μM and 200 μM treated myocyte media from control (Supplementary Fig. 2). Palmitate increased media concentrations of diacylglycerides (DG), lysophosphatidylcholines (LPC) and ceramides (Fig. 2a;

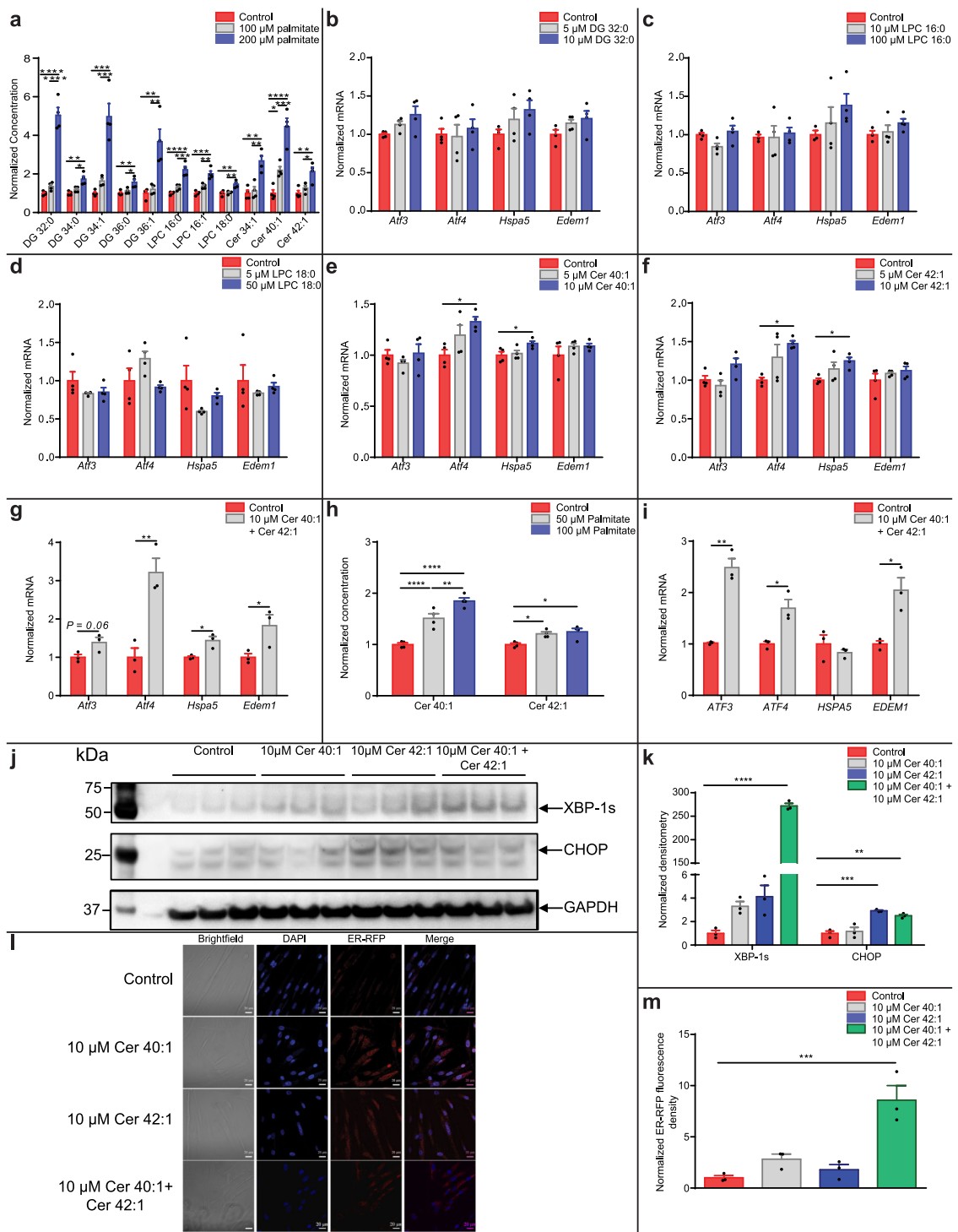

Supplementary Fig 3a). Specific species within these lipid classes - LPC 16:0, LPC 18:0, DG 32:0, ceramide 40:1 (N-(docosanoyl)-ceramide; Ceramide d18:1/22:0) and ceramide 42:1 (N-(tetracosanoyl)-ceramide; Ceramide d18:1/24:0) - were selected as candidate signals based on their increased concentration in conditioned media following palmitate treatment (post lipid class numbering represents carbon acyl-chain length and unsaturation). DG 32:0 (0–10 μM) (Fig. 2b), LPC 16:0 (0–100 μM) (Fig. 2c) and LPC 18:0 (0–50 μM) (Fig. 2d), at a dose response of physiological concentrations found in plasma[17,22], had no significant effect on UPR gene expression. However, treatment of myotubes with ceramides 40:1 (Fig. 2e) and 42:1 (Fig. 2f), at a

dose response in the physiological range found in plasma[22–24], increased expression of *Atf4* and *Hspa5*. Combined treatment of myotubes with both ceramides 40:1 and 42:1 increased expression of *Atf4*, *Hspa5* and *Edem1*, (increased expression of *Atf3* trended to significance, p = 0.06) (Fig. 2g). However, treatment of myocytes with ceramide 34:1 had a limited effect on UPR gene expression, suggesting specificity within lipid class (Supplementary Fig. 3b). Ceramides 40:1 and 42:1 (and ceramide 34:1) were also increased in concentration in media collected from HSkMCs treated with palmitate (Fig. 2h; Supplementary Fig. 3c). Treatment of HSkMCs with ceramides 40:1 and 42:1, was consistent with effects on murine myotubes with increased expression of

**Fig. 2 Long-chain ceramides activate the unfolded protein response in myotubes. a** Diacyglycerols (DG), lysophosphocholines (LPC) and ceramides (Cer) are enriched in conditioned media collected from C2C12 myotubes treated with vehicle control (red) 100 μM (grey) and 200 μM (blue) palmitate ($n = 4$; one-way ANOVA; control vs 200 μM palmitate, DG 32:0 $P < 0.0001$, DG 34:0 $P = 0.002$, DG 34:1 $P = 0.0002$, DG 36:0 $P = 0.0068$, DG 36:1 $P = 0.0025$, LPC 16:0 $P < 0.0001$, LPC 16:1 $P = 0.0001$, LPC 18:0 $P = 0.0029$, Cer 34:1 $P = 0.0012$, Cer 40:1 $P < 0.0001$, Cer 42:1 $P = 0.004$; control vs 100 μM palmitate, Cer 40:1 $P = 0.043$; 100 μM palmitate vs 200 μM palmitate, DG 32:0 $P < 0.0001$, DG 34:0 $P = 0.013$, DG 34:1 $P = 0.0006$, DG 36:0 $P = 0.023$, DG 36:1 $P = 0.0043$, LPC 16:0 $P = 0.0008$, LPC 16:1 $P = 0.0036$, LPC 18:0 $P = 0.0043$, Cer 34:1 $P = 0.002$, Cer 40:1 $P = 0.001$, Cer 42:1 $P = 0.02$). UPR gene expression in C2C12 myotubes treated with vehicle control (red) and lipids: **b** 5 μM (grey) and 10 μM (blue) DG 32:0 ($n = 4$), **c** 10 μM (grey) and 100 μM (blue) LPC 16:0 (control $n = 3$; 10 μM and 100 μM LPC 16:0 $n = 4$), **d** 5 μM (grey) and 50 μM (blue) LPC 18:0 ($n = 4$), **e** 5 μM (grey) and 10 μM (blue) Cer 40:1 ($n = 4$; one-way ANOVA; control vs 10 μM Cer 40:1, *Atf4* $P = 0.023$, *Hspa5* $P = 0.05$) and **f** 5 μM (grey) and 10 μM (blue) Cer 42:1 ($n = 4$; one-way ANOVA; control vs 10 μM Cer 42:1, *Atf4* $P = 0.02$, *Hspa5* $P = 0.036$). **g** UPR gene expression in C2C12 myotubes treated with vehicle control (red) and a combination of 10 μM Cer 40:1 and 10 μM Cer 42:1 (grey) ($n = 3$; two-tailed Student's $t$ test, *Atf4* $P = 0.007$, *Hspa5* $P = 0.017$, *Edem1* $P = 0.05$). **h** Cer 40:1 and Cer 42:1 concentrations in conditioned media from human skeletal muscle cells (HSkMCs) treated with 50 μM (grey) and 100 μM (blue) palmitate or vehicle control (red) ($n = 4$; one-way ANOVA; control vs 50 μM palmitate, Cer 40:1 $P = 0.001$, Cer 42:1 $P = 0.025$; control vs 100 μM palmitate Cer 40:1 $P < 0.0001$, Cer 42:1 $P = 0.01$; 50 μM palmitate vs 100 μM palmitate Cer 40:1 $P = 0.014$). **i** UPR gene expression in HSkMCs treated with vehicle control (red) or a combination of 10 μM Cer 40:1 and 10 μM Cer 42:1 (grey) ($n = 4$; two-tailed Student's $t$ test; *ATF3* $P = 0.001$, *ATF4* $P = 0.016$, *EDEM1* $P = 0.014$). **j** Western blot for C/EBP Homologous Protein (CHOP), X-box binding protein 1 active splice variant (XBP-1s) and Glyceraldehyde-3-phosphate dehydrogenase (GAPDH) expression control from HSkMCs treated with 10 μM ceramides 40:1 and 42:1 and a combination of both ceramides. **k** Normalised Western blot densitometry for CHOP and XBP-1s from HSkMCs treated with vehicle control (red) 10 μM ceramides 40:1 (grey) and 42:1 (blue) and a combination of both ceramides (green) ($n = 3$; one-way ANOVA; Control vs 10 μM Cer 40:1 and Cer 42:1, XBP-1s $P < 0.0001$, CHOP $P = 0.0018$; Control vs 10 μM Cer 42:1, CHOP $P = 0.0005$). **l** Confocal microscopy of HSkMCs expressing endoplasmic reticulum-targeted red fluorescent protein (ER-RFP; red) and treated with vehicle control, 10 μM Cer 40:1 or 10 μM Cer 42:1, or a combination of both 10 μM Cer 40:1 and 42:1. Nuclear staining with 4′,6-diamidino-2-phenylindole (DAPI; blue). Scale bar = 20 μm. **m** The normalised ER-RFP fluorescence density in HSkMCs expressing ER-RFP and treated with vehicle control (red), 10 μM Cer 40:1 (grey) or 10 μM Cer 42:1 (blue) or a combination of both 10 μM Cer 40:1 and 42:1 (green) ($n = 3$; one-way ANOVA; control vs 10 μM Cer 40:1 and Cer 42:1, $P = 0.0005$). $*P ≤ 0.05$, $**P < 0.01$, $***P < 0.001$, $****P < 0.0001$. Data are expressed as mean ± SEM with individual data points. Source data are provided as a Source Data file.

*ATF3*, *ATF4* and *EDEM1* (Fig. 2i). We confirmed induction of the UPR at the protein level using immunoblotting for the C/EBP Homologous Protein (CHOP) transcription factor (downstream of ATF3 and ATF4 in the UPR) and the active splice variant of the UPR transcription factor X-box binding protein 1 (XBP-1s) from protein extracts of HSkMCs treated with ceramides 40:1 and 42:1 and a combination of both ceramide species (Fig. 2j, k). Using ER-targeted red fluorescent protein and confocal microscopy imaging, we identified ER swelling, indicative of ER stress, in HSkMCs treated with ceramide 40:1 and ceramide 42:1 (Fig. 2l, m)[25]. These data demonstrate that human and mouse skeletal myocytes secrete UPR-activating long-chain ceramides in response to lipotoxicity.

**Long-chain ceramides are enriched in murine and human skeletal muscle during metabolic disease.** We then explored whether the long-chain ceramide 40:1 and ceramide 42:1 species were increased in the skeletal muscle of murine models and human patients with obesity and insulin resistance, metabolic diseases characterised by dyslipidaemia and lipotoxicity. Mice were fed either a western diet (WD) or a low-fat control diet (LFD) for 12 weeks. Western diet increased body weight and fat mass in the mice, while lean mass was unaffected (Supplementary Fig. 4a, b). Blood glucose concentrations were elevated in mice fed a western diet following an insulin tolerance test (ITT, Supplementary Fig. 4c, d) and glucose tolerance tests (GTT, Supplementary Fig 4e, f) at week 10 of the study. Therefore, mice fed a western diet exhibited increased glucose intolerance and insulin resistance. The long-chain ceramide signals were increased in the skeletal muscle (Fig. 3a, Supplementary Fig. 5a) and blood plasma (Fig. 3b, Supplementary Fig. 5b) of western diet-fed mice. Muscle and plasma concentrations of ceramide 40:1 and 42:1 were significantly positively correlated in the mice (Supplementary Fig 5c, d).

Next, we examined the association of skeletal muscle long-chain ceramides 40:1 and 42:1 with metabolic disease in human volunteers. Skeletal muscle biopsies of the pectoralis major were taken from age- and sex-matched patients with and without T2D

(Supplementary Table 1). Lipidomic analysis of the biopsies demonstrated greater concentrations of ceramides 40:1 and 42:1 in the muscle from patients with T2D (Fig. 3c, Supplementary Fig. 5e). Lipidomic analysis of the blood plasma from a random subset of patients identified increased concentrations of ceramides 40:1 and 42:1 in people with T2D (Fig. 3d, Supplementary Fig. 5f). Muscle and plasma concentrations of ceramide 40:1 and 42:1 were significantly positively correlated in the patients (Supplementary Fig 5g, h). These data highlight that increases in skeletal muscle and blood plasma long-chain ceramides are associated with metabolic disease in mice and humans.

**Palmitate-induced lipotoxicity increases ceramide synthesis by the ceramide synthase 2 de novo pathway through the UPR kinase Perk.** Ceramides are synthesised either via the salvage pathway or the de novo pathway, which begins with the condensation of palmitoyl-CoA with serine (Fig. 4a). To confirm that increases in ceramide secretion are coupled to increased intracellular synthesis, lipid extracts from C2C12 myotubes treated with 100 or 200 μM palmitate, or the BSA vehicle, were analysed using lipidomics. Palmitate increased intra-myocyte concentrations of ceramide 40:1 and 42:1 consistent with increased secretion of these species (Fig. 4b, Supplementary Fig 6a). Similar increases in intra-myocyte ceramide 40:1 and 42:1 concentration following palmitate treatment were observed in primary HSkMCs (Fig. 4c, Supplementary Fig 6b).

We hypothesised that exogenous palmitate would increase flux through the de novo ceramide synthesis pathway through conversion to palmitoyl-CoA. C2C12 myotubes, with and without palmitate treatment, were exposed to either myriocin (Fig. 4d), which inhibits serine palmitoyl transferase (SPT), or fumonisin B1 (FB1) (Fig. 4e), an inhibitor of multiple ceramide synthases (CerS). Inhibition of the de novo pathway at both SPT and CerS greatly abrogated palmitate-induced increases in the intracellular concentrations of ceramide 40:1 and ceramide 42:1, suggesting that palmitate-induced ceramide 40:1 and ceramide 42:1 synthesis is via the de novo pathway. There are six CerS isoforms, which exhibit a preference to produce ceramides of differing acyl-chain

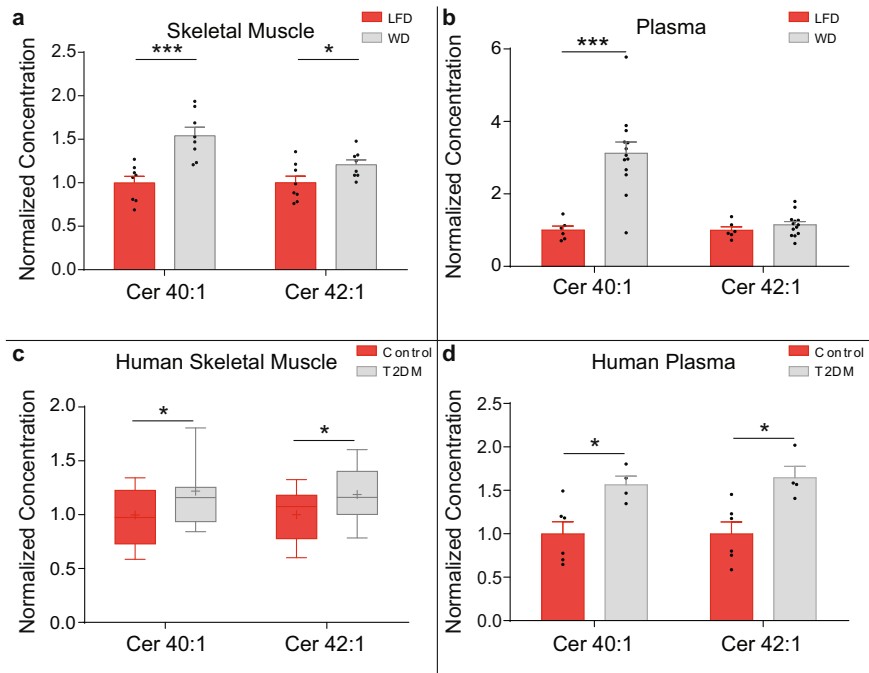

**Fig. 3 Long-chain ceramides are enriched in murine and human skeletal muscle during metabolic disease. a** Ceramide (Cer) 40:1 and Cer 42:1 in soleus muscle of low-fat diet (LFD) (red) and western diet (WD) (grey) fed mice ($n = 8$; two-tailed Student's $t$ test; Cer 40:1 $P = 0.00056$, Cer 42:1 $P = 0.045$). **b** Cer 40:1 and Cer 42:1 in plasma of LFD (red) and WD (grey) fed mice (LFD $n = 6$, WD $n = 13$; two-tailed Student's $t$ test; Cer 40:1 $P = 0.00031$). **c** Cer 40:1 and Cer 42:1 in skeletal muscle biopsies from patients with type 2 diabetes (T2D) (grey) and controls (red) (control $n = 52$, T2D $n = 21$; two-tailed Student's $t$ test; Cer 40:1 $P = 0.033$, Cer 42:1 $P = 0.017$). **d** Cer 40:1 and Cer 42:1 in blood plasma from patients with T2D (grey) and controls (red) (control $n = 6$, T2D $n = 4$; two-tailed Student's $t$ test; Cer 40:1 $P = 0.0118$, Cer 42:1 $P = 0.012$). *$P \leq 0.05$, **$P < 0.01$, ***$P < 0.001$, ****$P < 0.0001$. Data in bar charts are expressed as mean ± SEM with individual data points. Box and whisker plots show 25th to 75th percentile (box) min to max (whiskers), mean (+) and median (−). Source data are provided as a Source Data file.

length[26]. Our results suggest palmitate stimulates increases in long-chain ceramide 40:1 and ceramide 42:1 production and secretion. Long-chain ceramides, including ceramide 40:1 and 42:1, are predominantly synthesised by CerS2[27]. However, CerS4 overlaps with CerS2 in its specificity for the production of long-chain ceramides[28]. CerS4 production of long-chain ceramides is unaffected by FB1[28], this suggests that the mechanism for palmitate-induced ceramide 40:1 and 42:1 production is independent of CerS4. The expression of CerS4 was decreased in response to palmitate treatment in C2C12 myocytes (Supplementary Fig 6c). In contrast palmitate-treatment of C2C12 myotubes and HSkMCs induced *CerS*2 expression (Fig. 4f, g). To confirm long-chain ceramides 40:1 and 42:1 are generated through CERS2 in response to palmitate treatment in myocytes, we decreased *CERS2* expression in HSkMCs by 70% using siRNA (Fig. 4h). Knockdown of *CERS2* expression decreased intracellular (Fig. 4i) and extracellular media (Fig. 4j) ceramide 40:1 and 42:1 concentrations and inhibited the palmitate-induced increase in the myocyte media concentration of the lipids. These data suggest increased ceramide 40:1 and ceramide 42:1 concentrations in response to palmitate are generated through the de novo pathway. We then transferred conditioned serum- and palmitate-free media collected from HSkMCs treated with 100 μM palmitate or BSA control, with and without siRNA-mediated *CERS2* knockdown, to naïve HSkMCs. The induction of UPR gene expression induced by media conditioned on palmitate-treated HSkMCs was inhibited by *CERS2* knockdown (Fig. 4k) in agreement with the effect of *CERS2* knockdown on ceramide 40:1 and ceramide 42:1 media concentrations. These data confirm that the CERS2 generated long-chain ceramides are cell non-autonomous UPR-inducing lipid signals.

Next, we hypothesised that long-chain ceramide synthesis in response to lipotoxicity was directly regulated by specific UPR pathways. Intracellular ceramide 40:1 and 42:1 concentrations were analysed in C2C12 myotubes co-treated with 200 μM palmitate or the BSA vehicle, and either 10 μM 4μ8C, which inhibits Ire1 activity (Fig. 5a) or 10 μM AMG PERK 44, an inhibitor of the UPR kinase Perk (Fig. 5b). While inhibition of Ire1 had no effect on ceramide concentrations, inhibition of Perk abrogated palmitate-induced increases in ceramides 40:1 and 42:1. To confirm PERK contributed to the palmitate-mediated secretion of ceramide 40:1 and 42:1 from myocytes, we decreased *PERK* expression in HSkMCs by greater than 85% using siRNA (Fig. 5c). Knockdown of *PERK* decreased intracellular (Fig. 5d) and the extracellular media (Fig. 5e) ceramide 40:1 and 42:1 concentrations and inhibited the palmitate-induced increase in the myocyte media concentration of the lipids. These data identify a role for the UPR, and specifically Perk, in regulating sphingolipid synthesis.

**In vivo transgenic inactivation of Cers2 decreases the plasma and muscle long-chain ceramide signals and muscle UPR activation.** CerS2 was implicated in the biosynthesis of the UPR-inducing ceramide signals in myocytes in vitro. We next explored whether transgenic catalytic inactivation of CerS2 in the *CerS2* H/A mouse (two consecutive histidine for alanine substitutions in the CerS2 catalytic centre)[29] affected synthesis of the ceramides. Lipidomics was used to analyse the concentration of ceramide species in the skeletal muscle (Fig. 6a, Supplementary Fig. 7a) and plasma (Fig. 6b, Supplementary Fig. 7b) of mice heterozygous for *CerS2* H/A (Het *CerS2* H/A) and mice homozygous for *CerS2* H/A (Homo

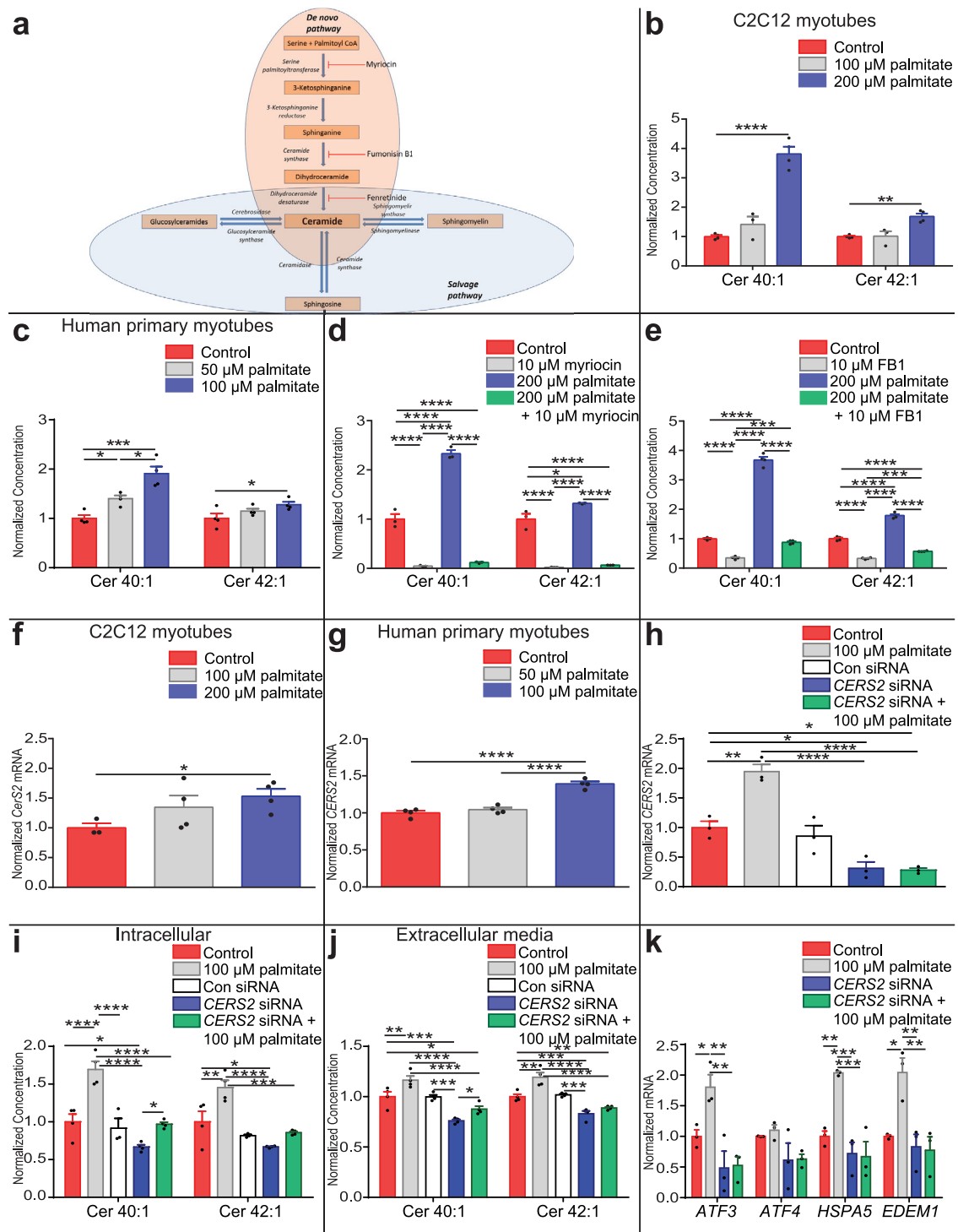

CerS2 H/A) compared to wild type littermate controls. Tissues of the Homo CerS2 H/A mouse exhibit a compensatory increase in ceramide 34:1 concentration (Supplementary Fig. 7a, b)[29]. To limit compensatory effects and generate a more physiological model we also used the Het CerS2 H/A mouse which is not characterised by compensatory increases in shorter chain ceramides (Supplementary Fig. 7a, b). The concentrations of ceramide 40:1 and 42:1 were depleted in plasma and muscle of Het CerS2 H/A, and Homo CerS2 H/A mice in a gene-dosage responsive manner. We then assessed whether heterozygous and homozygous catalytic inactivation of CerS2 and subsequent decreases in the long-chain ceramides altered skeletal muscle expression of UPR genes. The

expression of both Atf4 and Hspa5 was significantly decreased in the skeletal muscle of Het CerS2 H/A, and Homo CerS2 H/A mice (Fig. 6c).

These data indicate that CerS2 regulates the systemic and muscle concentration of the long-chain ceramide signals and disruption in biosynthesis of the ceramides decreases muscle UPR activation in vivo.

**Extracellular vesicles mediate secretion and transport of the UPR-inducing long-chain ceramide signal.** Ceramides are hydrophobic. If ceramides are to function as a paracrine or

**Fig. 4 Long-chain ceramides are synthesised via the Cers2 de novo pathway. a** The de novo synthesis pathway and salvage pathway of ceramide synthesis *with* de novo synthesis pathway inhibitors (red). Intracellular ceramides (Cer) in **b** C2C12 myotubes (control, 200 μM palmitate n = 4; 100 μM palmitate n = 3; one-way ANOVA; control vs 200 μM palmitate, Cer 40:1 $P < 0.0001$, Cer 42:1 $P = 0.0026$) (control = red, 100 μM palmitate = grey, 200 μM palmitate = blue) and **c** human skeletal muscle cells (HSkMCs) (control = red, 50 μM palmitate = grey, 100 μM palmitate = blue) (n = 4; one-way ANOVA; control vs 100 μM palmitate, Cer 40:1 $P = 0.043$; control vs 200 μM palmitate, Cer 40:1 $P = 0.0003$, Cer 42:1 $P = 0.05$; 100 μM palmitate vs 200 μM palmitate, Cer 40:1 $P = 0.014$) treated with palmitate. Long-chain ceramides in C2C12 myotubes co-treated with either **d** 10 μM myriocin (serine palmitoyl transferase inhibitor) (control = red, 10 μM myriocin = grey, 200 μM palmitate = blue, 200 μM palmitate + 10 μM myriocin = green) (n = 3; one-way ANOVA; control vs 10 μM myriocin, Cer 40:1 $P < 0.0001$, Cer 42:1 $P < 0.0001$; control vs 200 μM palmitate, Cer 40:1 $P < 0.0001$, Cer 42:1 $P = 0.014$; control vs 200 μM palmitate + 10 μM myriocin, Cer 40:1 $P < 0.0001$, Cer 42:1 $P < 0.0001$; 200 μM palmitate vs 10 μM myriocin, Cer 40:1 $P < 0.0001$, Cer 42:1 $P < 0.0001$; 200 μM palmitate vs 200 μM palmitate + 10 μM myriocin, Cer 40:1 $P < 0.0001$, Cer 42:1 $P < 0.0001$) or **e** 10 μM fumonisin B1 (FB1) (ceramide synthase (CerS) inhibitor) (control = red, 10 μM FB1 = grey, 200 μM palmitate = blue, 200 μM palmitate + 10 μM FB1 = green) (n = 4; one-way ANOVA; control vs 10 μM FB1, Cer 40:1 $P < 0.0001$, Cer 42:1 $P < 0.0001$; control vs 200 μM palmitate, Cer 40:1 $P < 0.0001$, Cer 42:1 $P < 0.0001$; 200 μM palmitate vs 10 μM FB1, Cer 40:1 $P < 0.0001$, Cer 42:1 $P < 0.0001$; 10 μM FB1 vs 200 μM palmitate + 10 μM FB1, Cer 40:1 $P = 0.0001$, Cer 42:1 $P = 0.0005$; 200 μM palmitate vs 200 μM palmitate + 10 μM FB1, Cer 40:1 $P < 0.0001$, Cer 42:1 $P < 0.0001$) and either 200 μM palmitate or bovine serum albumin (BSA). CerS2 expression in **f**. C2C12 myotubes (control = red; 100 μM palmitate = grey; 200 μM palmitate = blue) (control n = 3; 100 μM and 200 μM palmitate n = 4; one-way ANOVA; control vs 200 μM palmitate $P = 0.044$) and **g**. HSkMCs treated with palmitate or BSA (control = red; 50 μM palmitate = grey; 100 μM palmitate = blue) (n = 4; one-way ANOVA; control vs 100 μM palmitate $P < 0.0001$; 50 μM palmitate vs 100 μM palmitate $P < 0.0001$). **h** *CERS2* expression in control HSkMCs (red) and HSkMCs treated with 100 μM palmitate (grey), scrambled control siRNA (con siRNA; white), siRNA against *CERS2* (*CERS2* siRNA; blue) or *CERS2* siRNA and 100 μM palmitate (green) (n = 3; one-way ANOVA; control vs 100 μM palmitate $P = 0.0016$; control vs *CERS2* siRNA $P = 0.014$; control vs *CERS2* siRNA + 100 μM palmitate $P = 0.011$; 100 μM palmitate vs *CERS2* siRNA $P < 0.0001$; 100 μM palmitate vs *CERS2* siRNA + 100 μM palmitate $P < 0.0001$). **i** Cer 40:1 and Cer 42:1 normalised concentrations in control HSkMCs (red) and HSkMCs treated with 100 μM palmitate (grey), con siRNA (white), *CERS2* siRNA (blue) or *CERS2* siRNA and 100 μM palmitate (green) (n = 4; con siRNA n = 3; one-way ANOVA; control vs 100 μM palmitate, Cer 40:1 $P < 0.0001$, Cer 42:1 $P = 0.001$; control vs *CERS2* siRNA Cer 40:1 $P = 0.013$, Cer 42:1 $P = 0.01$; 100 μM palmitate vs *CERS2* siRNA Cer 40:1 $P < 0.0001$, Cer 42:1 $P < 0.0001$; 100 μM palmitate vs *CERS2* siRNA + 100 μM palmitate Cer 40:1 $P < 0.0001$, Cer 42:1 $P = 0.0001$; *CERS2* siRNA vs *CERS2* siRNA + 100 μM palmitate Cer 40:1 $P = 0.023$). **j** Cer 40:1 and Cer 42:1 normalised concentrations in media conditioned on control HSkMCs (red) and HSkMCs treated with 100 μM palmitate (grey), con siRNA (white), *CERS2* siRNA (blue) or *CERS2* siRNA and 100 μM palmitate (green) (n = 4; one-way ANOVA; control vs 100 μM palmitate, Cer 40:1 $P = 0.003$, Cer 42:1 $P = 0.0001$; control vs *CERS2* siRNA Cer 40:1 $P = 0.0001$, Cer 42:1 $P = 0.0005$; control vs *CERS2* siRNA + 100 μM palmitate, Cer 40:1 $P = 0.019$, Cer 42:1 $P = 0.011$; 100 μM palmitate vs *CERS2* siRNA Cer 40:1 $P < 0.0001$, Cer 42:1 $P < 0.0001$; 100 μM palmitate vs *CERS2* siRNA + 100 μM palmitate Cer 40:1 $P < 0.0001$, Cer 42:1 $P < 0.0001$; *CERS2* siRNA vs *CERS2* siRNA + 100 μM palmitate Cer 40:1 $P = 0.024$). **k** UPR gene expression in HSkMCs receiving conditioned media from 100 μM palmitate or BSA-treated HSkMCs with and without *CERS2* siRNA (control = red, 100 μM palmitate = grey, control + *CERS2* siRNA = blue, 100 μM palmitate + *CERS2* siRNA = green) (n = 3; one-way ANOVA; control vs 100 μM palmitate, ATF3 $P = 0.045$, HSPA5 $P = 0.0043$, EDEM1 $P = 0.012$; 100 μM palmitate vs *CERS2* siRNA, ATF3 $P = 0.003$, HSPA5 $P = 0.001$, EDEM1 $P = 0.0052$; 100 μM palmitate vs *CERS2* siRNA + 100 μM palmitate ATF3 $P = 0.004$, HSPA5 $P = 0.0008$, EDEM1 $P = 0.004$). $*P \leq 0.05$, $**P < 0.01$, $***P < 0.001$, $****P < 0.0001$. Data are expressed as mean ± SEM with individual data points. Source data are provided as a Source Data file.

systemic signal, propagating the induction of the UPR, then there must be an appropriate physiological mechanism to overcome limitations to solubility and mediate extracellular transport. Due to their physicochemical nature this is unlikely to be via canonical membrane transporters. Extracellular vesicles (EVs) have emerged as important vehicles for signalling mediators[30]. There is developing recognition that bioactive lipid signals may be partitioned into EVs[31–33]. We investigated whether EVs function to transport the long-chain ceramide signals extracellularly. Conditioned media was ultracentrifuged to isolate small EV (exosome) and larger EV (microvesicle) populations. Populations were analysed by NanoSight tracking (Fig. 7a), which demonstrated an increase in both small and large EV release following palmitate treatment (Fig. 7b, c, Supplementary Fig. 8). To confirm increased EV release from myocytes in response to palmitate in accordance with the minimal information for studies of extracellular vesicles[34], the total protein concentration of EV isolates from media conditioned on BSA-treated control and 200 μM palmitate-treated myocytes was measured (Supplementary Fig 9a). To confirm the nature and degree of purity of the EV preparation we used immunoblotting to determine the presence of the plasma membrane associated transmembrane protein, transferrin receptor 2 (Tfr2), the cytosolic protein superoxide dismutase 1 (Sod) recovered in EVs, and the mitochondrial protein carnitine palmitoyl-transferase 1 (Cpt1), a component of the non-EV co-isolated structure in EV isolations, conditioned media and whole cell myocyte protein isolations (Supplementary Fig. 9b). These data indicate palmitate induces myocyte EV release.

We then explored whether small EVs or large EVs released from palmitate-treated myocytes contained the UPR-inducing signal. Small EVs (Fig. 7d) and large EVs (Fig. 7e) isolated from palmitate-treated myotubes increased UPR gene expression in naïve myotubes following their reconstitution in serum-free media. EV-depleted conditioned media, however, did not induce UPR gene expression in naive myotubes (Fig. 7f). Lipidomic analysis of small EVs (Fig. 7g) and large EVs (Fig. 7h) showed that palmitate-treated myotubes produced EVs enriched in ceramides 40:1 and 42:1. Reciprocally, LC-MS analysis of EV-depleted media demonstrated a ~92% reduction in long-chain ceramides 40:1 and 42:1 (Fig. 7i). Therefore, EVs represent a mechanism for long-chain ceramide transport between myotubes and subsequent induction of the UPR.

**Long-chain ceramides activate the UPR through dihydroceramides.** We investigated the mechanism by which ceramide-enriched EVs induce UPR activation. Myotubes were treated with 10 μM ceramide 40:1, 10 μM ceramide 42:1, a combination of both species, or the vehicle control. Cells were then analysed using lipidomics. Ceramide 40:1 treatment increased intracellular concentrations of ceramide 40:1, and ceramide 42:1 treatment increased intracellular concentrations of ceramide 42:1 (Fig. 8a). This indicates ceramides are imported into the myotubes. Lipidomic analysis also demonstrated an increase in the intracellular concentration of the cognate dihydroceramides 40:0 and 42:0 (Fig. 8b). Dihydroceramides are ceramide precursors. Therefore, exogenous ceramides may be imported and recycled within the

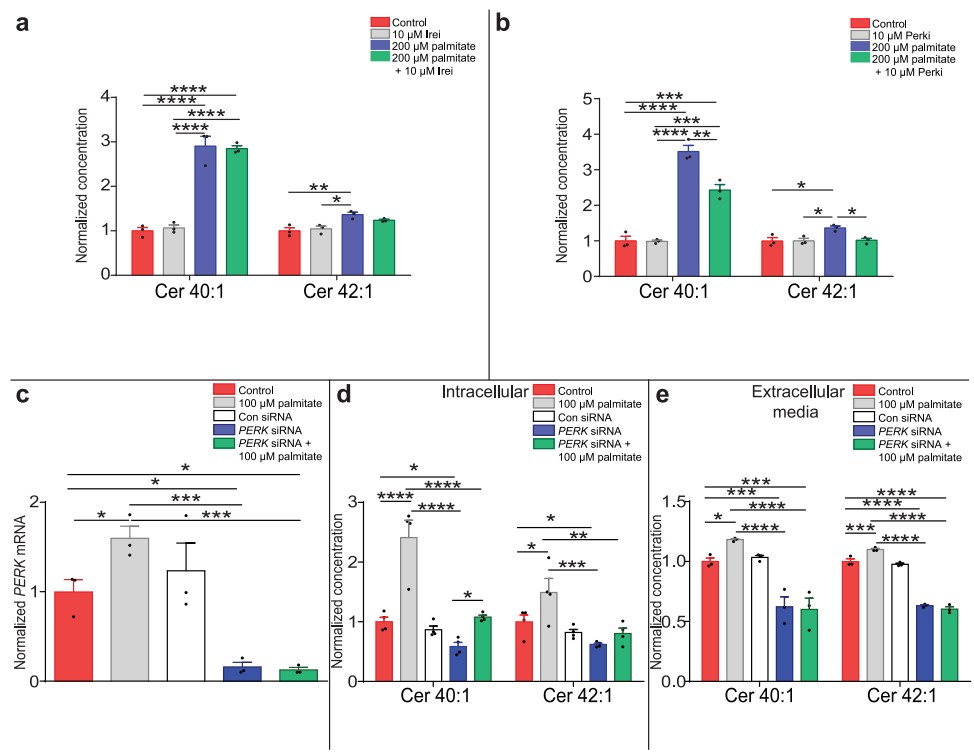

**Fig. 5 Palmitate-induced long-chain ceramide synthesis requires Perk. a** Ceramides (Cer) in C2C12 myotubes treated with palmitate or Bovine Serum Albumin (BSA), in combination with 10 μM 4μ8c, an inhibitor of Ire1 (Ire1i) (control = red, Ire1i = grey, palmitate = blue, palmitate + Ire1i = green) (n = 3; one-way ANOVA; control vs 200 μM palmitate Cer 40:1 P < 0.0001, Cer 42:1 P = 0.0062; control vs 200 μM palmitate + 10 μM Ire1i Cer 40:1 P < 0.0001; 10 μM Ire1i vs 200 μM palmitate Cer 40:1 P < 0.0001, Cer 42:1 P = 0.013; 10 μM Ire1i vs 200 μM palmitate + 10 μM Ire1i Cer 40:1 P < 0.0001). **b** Ceramides in C2C12 myotubes treated with palmitate or BSA, in combination with 10 μM AMG PERK 44, an inhibitor of PERK (Perki) (control = red, Perki = grey, palmitate = blue, palmitate + Perki = green) (n = 3; one-way ANOVA; control vs 200 μM palmitate Cer 40:1 P < 0.0001, Cer 42:1 P = 0.022; control vs 200 μM palmitate + 10 μM Perki Cer 40:1 P = 0.0003; 10 μM Perki vs 200 μM palmitate Cer 40:1 P < 0.0001, Cer 42:1 P = 0.023; 10 μM Perki vs 200 μM palmitate + 10 μM Perki Cer 40:1 P = 0.0002; 200 μM palmitate vs 200 μM palmitate + 10 μM Perki Cer 40:1 P = 0.0017, Cer 42:1 P = 0.028). **c** PERK expression in control HSkMCs (red) and HSkMCs treated with 100 μM palmitate (grey), scrambled control siRNA (con siRNA; white), siRNA against PERK (PERK siRNA; blue) or PERK siRNA and 100 μM palmitate (green) (n = 3; one-way ANOVA; control vs 100 μM palmitate P = 0.05; control vs PERK siRNA P = 0.016; control vs PERK siRNA + 100 μM palmitate P = 0.016; 100 μM palmitate vs PERK siRNA P = 0.0004; 100 μM palmitate vs PERK siRNA + 100 μM palmitate P = 0.0004). **d** Cer 40:1 and Cer 42:1 normalised concentration in control HSkMCs (red) and HSkMCs treated with 100 μM palmitate (grey), con siRNA (white), PERK siRNA (blue) or PERK siRNA and 100 μM palmitate (green) (n = 4; one-way ANOVA; control vs 100 μM palmitate, Cer 40:1 P < 0.0001, Cer 42:1 P = 0.015; control vs PERK siRNA Cer 40:1 P = 0.05, Cer 42:1 P = 0.05; 100 μM palmitate vs PERK siRNA Cer 40:1 P < 0.0001, Cer 42:1 P = 0.0002; 100 μM palmitate vs PERK siRNA + 100 μM palmitate Cer 40:1 P < 0.0001, Cer 42:1 P = 0.0015; PERK siRNA vs PERK siRNA + 100 μM palmitate Cer 40:1 P = 0.0262). **e** Cer 40:1 and Cer 42:1 normalised concentration in media conditioned on control HSkMCs (red) and HSkMCs treated with 100 μM palmitate (grey), con siRNA (white), PERK siRNA (blue) or PERK siRNA and 100 μM palmitate (green) (n = 3; one-way ANOVA; control vs 100 μM palmitate, Cer 40:1 P = 0.0487, Cer 42:1 P = 0.0007; control vs PERK siRNA Cer 40:1 P = 0.0009, Cer 42:1 P < 0.0001; control vs PERK siRNA + 100 μM palmitate, Cer 40:1 P = 0.0006, Cer 42:1 P < 0.0001; 100 μM palmitate vs PERK siRNA Cer 40:1 P < 0.0001, Cer 42:1 P < 0.0001; 100 μM palmitate vs PERK siRNA + 100 μM palmitate Cer 40:1 P < 0.0001, Cer 42:1 P < 0.0001). *P ≤ 0.05, **P < 0.01, ***P < 0.001, ****P < 0.0001. Data are expressed as mean ± SEM with individual data points. Source data are provided as a Source Data file.

cell. Similar effects on intracellular ceramide 40:1 and 42:1, and dihydroceramide 40:0 and 42:0 were seen in long-chain ceramide-treated primary HSkMCs (Fig. 8c, d).

Dihydroceramides drive activation of the UPR[35]. We assessed the role of dihydroceramides in lipotoxicity-induced ER stress. Palmitate-treated myotubes were co-incubated with 5 μM fenretinide, an inhibitor of dihydroceramide desaturase 1 (Des1), the enzyme catalysing the conversion of dihydroceramides to ceramides. Co-treatment with palmitate and fenretinide decreased ceramide 42:1 concentration without effect on ceramide 40:1 concentration (Supplementary Fig. 10a). However, the intracellular concentrations of dihydroceramides were greatly increased by palmitate and fenretinide co-treatment (Supplementary Fig. 10b). Concomitant palmitate treatment and Des1 inhibition synergistically enhanced expression of UPR genes (Supplementary Fig. 10c). To confirm a role for dihydroceramides

in lipotoxicity-induced UPR activation we used siRNA to decrease DEGS1 expression (the gene encoding Des1) by > 50% in the translational primary HSkMC model (Fig. 8e). DEGS1 knockdown in HSkMCs decreased the intracellular concentration of ceramides 40:1 and 42:1 (Fig. 8f). Concomitant palmitate treatment of human primary myotubes with DEGS1 knockdown increased the intracellular concentrations of dihydroceramides 40:0 and 42:0 (Fig. 8g). Palmitate treatment in combination with DEGS1 knockdown synergistically enhanced UPR gene expression in human myotubes (Fig. 8h). These data implicate dihydroceramides in the cell non-autonomous ceramide-induced activation of UPR signalling in lipotoxicity.

## Discussion

ER stress has emerged as a potentially unifying mechanism underpinning the development of lipotoxicity and metabolic

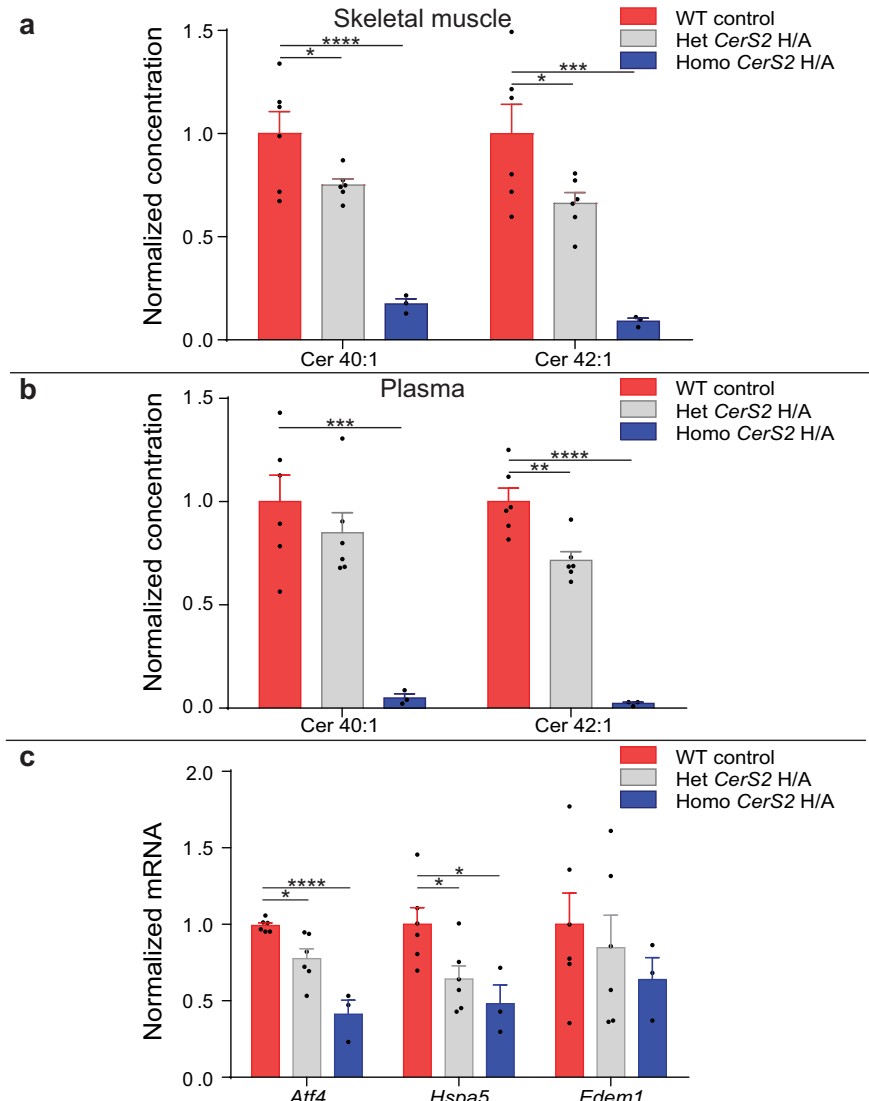

**Fig. 6 Transgenic CerS2 inactivation decreases plasma and muscle long-chain ceramides and muscle UPR gene expression. a** Ceramides (Cer) 40:1 and Cer 42:1 in the soleus muscle of transgenic mice heterozygous (Het *CerS2* H/A; grey) and homozygous (Homo *CerS2* H/A; blue) for catalytically inactivated ceramide synthase 2 compared to wildtype (WT; red) (WT, Het *CerS2* H/A $n = 6$; Homo *CerS2* H/A $n = 3$; one-way ANOVA; WT vs Het *CerS2* H/A, Cer 40:1 $P = 0.03$, Cer 42:1 $P = 0.03$; WT vs Homo *CerS2* H/A, Cer 40:1 $P < 0.0001$, Cer 42:1 $P = 0.0005$). **b** Cer 40:1 and Cer 42:1 in the plasma of Het *CerS2* H/A (grey) and Homo *CerS2* H/A (blue) mice compared to WT (red) (WT, Het *CerS2* H/A $n = 6$; Homo *CerS2* H/A $n = 3$; one-way ANOVA; WT vs Het *CerS2* H/A, Cer 42:1 $P = 0.0018$; WT vs Homo *CerS2* H/A, Cer 40:1 $P = 0.0006$, Cer 42:1 $P < 0.0001$). **c** Activating transcription factor 4 (*Atf4*), Heat Shock Protein Family A (Hsp70) Member 5 (*Hspa5*) and ER Degradation Enhancing Alpha-Mannosidase Like Protein 1 (*Edem1*) soleus muscle UPR gene expression in Het *CerS2* H/A (grey) and Homo *CerS2* H/A (blue) mice compared to WT (red) (WT, Het *CerS2* H/A $n = 6$; Homo *CerS2* H/A $n = 3$; one-way ANOVA; WT vs Het *CerS2* H/A, *Atf4* $P = 0.019$, *Hspa5* $P = 0.02$; WT vs Homo *CerS2* H/A, *Atf4* $P < 0.0001$, *Hspa5* $P = 0.019$) *$P \leq 0.05$, **$P < 0.01$, ***$P < 0.001$, ****$P < 0.0001$. Data are expressed as mean ± SEM with individual data points. Source data are provided as a Source Data file.

dysfunction in metabolic disease. Cell non-autonomous signalling represents a mode of regulation in the control of UPR activation[16]. The identity of these UPR-inducing signals remained unknown. This study identifies the nature of a cell non-autonomous signal that propagates the activation of the UPR and describes such a signal in the context of lipotoxicity (Supplementary Fig. 11). We show that palmitate-mediated activation of the Perk-arm of the UPR stimulates an increase in the de novo synthesis, via CerS2, of long-chain ceramides (ceramide 40:1 and 42:1). These ceramides are packaged into extracellular vesicles, secreted, and propagate activation of the UPR, potentially via the accumulation of dihydroceramides in naïve myotubes. The ER is the key cellular site of de novo ceramide biosynthesis[36]. Using long-chain ceramides, a product of this pathway, as signals of ER

stress to induce cell non-autonomous UPR activation appears advantageous. However, our work does not preclude the presence of other metabolite factors, protein signals, or bioactive lipids that may contribute to cell non-autonomous UPR activation. Indeed, our data suggest the presence of an, as yet unidentified, aqueous soluble metabolite factor(s) secreted from palmitate-treated myocytes which also activate UPR gene expression. These findings have consequences for our understanding of both the mechanisms of lipotoxicity and the regulation of ER stress and UPR activation.

Ceramides have previously been implicated in both the development of metabolic disease and the regulation of ER stress. Blood plasma and tissue ceramide and dihydroceramide concentrations are associated with a range of cardiometabolic

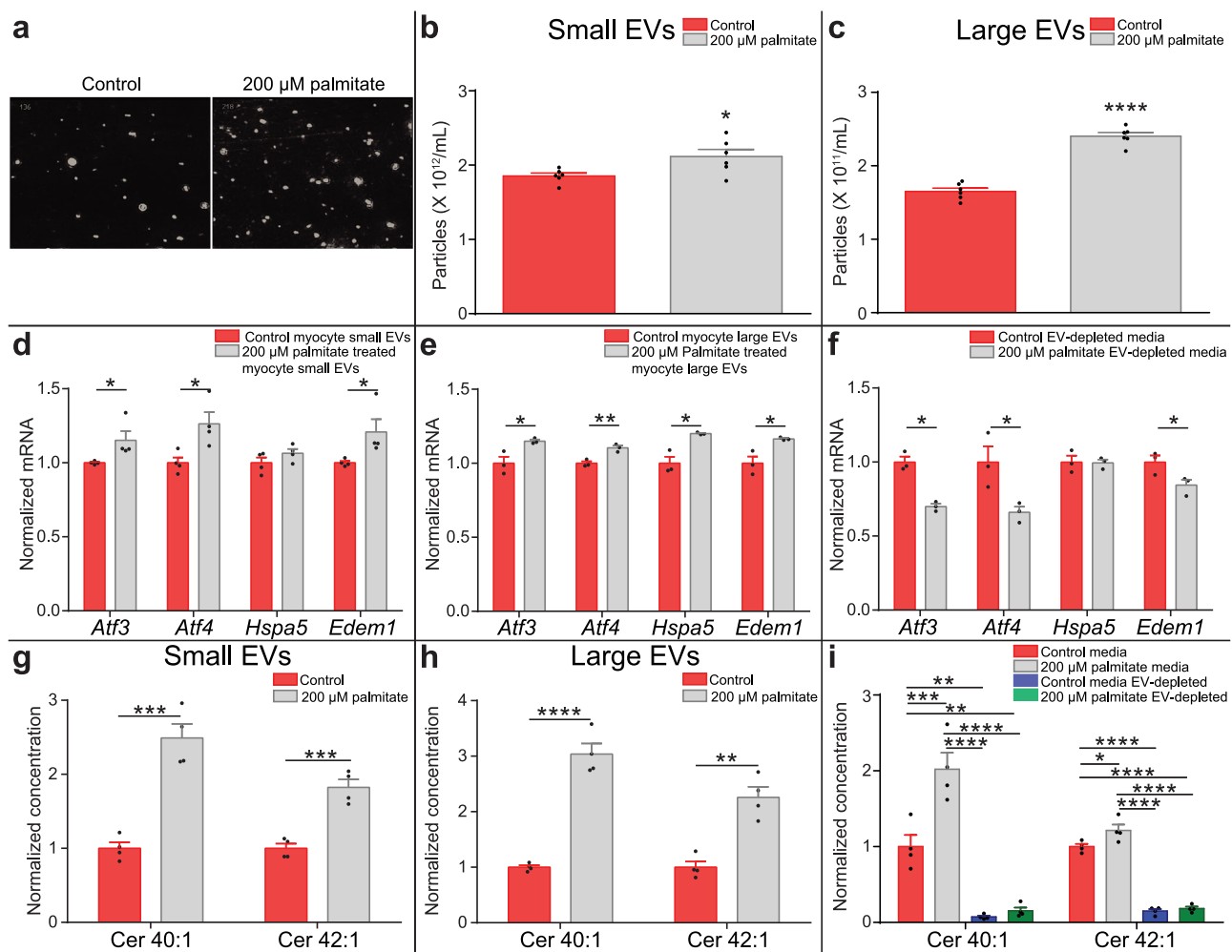

**Fig. 7 UPR-inducing ceramides are secreted via extracellular vesicles. a** Extracellular vesicle (EV) population images in conditioned media from C2C12 myotubes treated with either 200 μM palmitate (grey) or bovine serum albumin (BSA) control (red). **b** The small EV population secreted by C2C12 myotubes treated with either 200 μM palmitate (grey) or BSA vehicle (red) ($n = 4$; two-tailed Student's $t$ test; $P = 0.02$). **c** The large EV population secreted by C2C12 myotubes treated with either 200 μM palmitate (grey) or the BSA vehicle (red) ($n = 4$; two-tailed Student's $t$ test; $P < 0.0001$). Activating transcription factor 3 (*Atf3*), activating transcription factor 4 (*Atf4*), Heat Shock Protein Family A (Hsp70) Member 5 (*Hspa5*) and ER Degradation Enhancing Alpha-Mannosidase Like Protein 1 (*Edem1*) UPR gene expression in myotubes treated with **d**. small EVs ($n = 4$; two-tailed Student's $t$ test; *Atf3* $P = 0.05$, *Atf4* $P = 0.02$; *Edem1* $P = 0.05$) and **e**. large EVs ($n = 3$; two-tailed Student's $t$ test; *Atf3* $P = 0.03$, *Atf4* $P = 0.007$; *Hspa5* $P = 0.01$, *Edem1* $P = 0.02$) isolated from the media of palmitate (grey) and BSA-treated (red) myotubes. **f** UPR gene expression in myotubes treated with conditioned EV-depleted media from 200 μM palmitate-treated (grey) and control (red) myotubes ($n = 3$; two-tailed Student's $t$ test; *Atf3* $P = 0.002$, *Atf4* $P = 0.04$, *Edem1* $P = 0.05$). Ceramide (Cer) 40:1 and Cer 42:1 concentration in **g**. small EVs ($n = 4$; two-tailed Student's $t$ test; Cer 40:1 $P = 0.0004$, Cer 42:1 $P = 0.0006$) and **h**. large EVs ($n = 4$; two-tailed Student's $t$ test; Cer 40:1 $P < 0.0001$, Cer 42:1 $P = 0.001$) from the media of C2C12 myotubes treated with either 200 μM palmitate (grey) or BSA control (red). **i** Cer 40:1 and Cer 42:1 concentration in EV-containing and EV-depleted media from C2C12 myotubes treated with either 200 μM palmitate or BSA control (control = red, control EV-depleted = grey, palmitate = blue, palmitate EV-depleted = green) ($n = 4$; one-way ANOVA; control vs 200 μM palmitate, Cer 40:1 $P = 0.0007$, Cer 42:1 $P = 0.012$; control vs control EV-depleted, Cer 40:1 $P = 0.0011$, Cer 42:1 $P < 0.0001$; control vs 200 μM palmitate EV-depleted, Cer 40:1 $P = 0.0016$, Cer 42:1 $P < 0.0001$; 200 μM palmitate vs control EV-depleted, Cer 40:1 $P < 0.0001$, Cer 42:1 $P < 0.0001$; 200 μM palmitate vs 200 μM palmitate EV-depleted, Cer 40:1 $P < 0.0001$, Cer 42:1 $P < 0.0001$). *$P \leq 0.05$, **$P < 0.01$, ***$P < 0.001$, ****$P < 0.0001$. Data are expressed as mean ± SEM with individual data points. Source data are provided as a Source Data file.

diseases[37–41]. Here, we observe an increase in the concentrations of long-chain ceramides 40:1 and 42:1 in the blood plasma and muscle of western diet-induced obese mice and humans with T2D. Confirming that ceramides 40:1 and 42:1 are enriched in muscle by metabolic disease in mouse and humans. We do not preclude that circulating concentrations of long-chain ceramides in this setting may be contributed to by additional tissues besides skeletal muscle. Ceramides have been identified as key mechanistic contributors to the lipotoxicity-induced development of adipose tissue and hepatic insulin resistance[42]. Inhibition of

ceramide synthesis ameliorates palmitate-induced ER stress in pancreatic β-cells[43]. Similarly, palmitate-induced the production of long-chain ceramides, and siRNA knockdown of CerS 5 and 6 prevented myristate-induced ER stress, in intestinal epithelial cells[44]. A short-chain ceramide analogue, infused into the hypothalamus of rats increases hypothalamic ER stress, and reduces the thermogenic capacity of brown adipose tissue, thus, highlighting that ceramide-induced ER stress in one tissue can have a metabolic impact on distal tissues[45]. Evidence is emerging that skeletal muscle ceramide metabolism has a key role in the

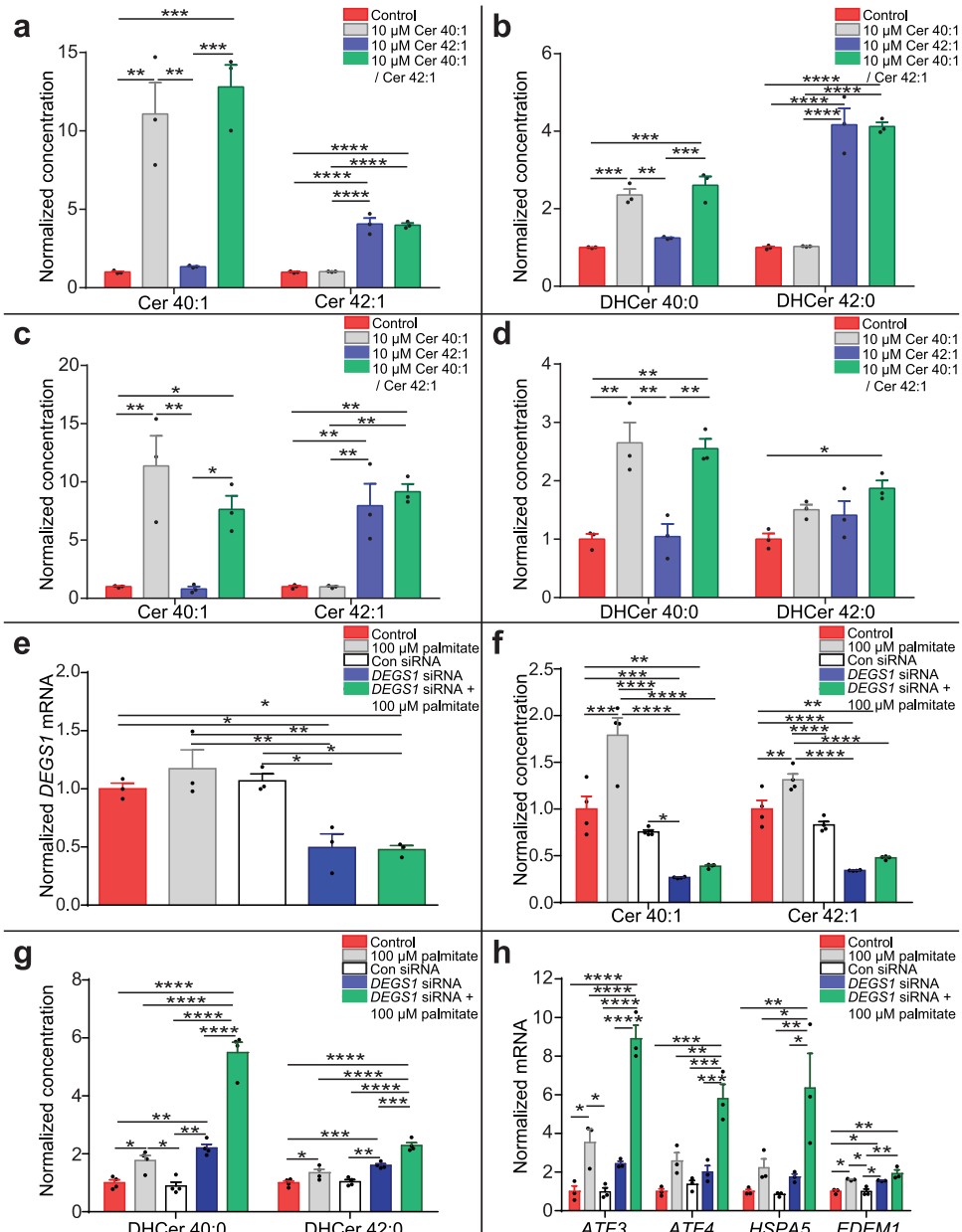

regulation of systemic physiology. Skeletal muscle ceramides are elevated in both patients with, and animal models of, obesity and diabetes and negatively correlate with insulin sensitivity[46–48]. In addition, muscle-specific disruption of ceramide synthesis by CerS1 improves systemic insulin sensitivity in diet-induced obese mice[49]. Here we demonstrate that lipotoxicity-induced ER stress can stimulate biosynthesis and secretion of ceramides which regulate UPR activation in a cell non-autonomous manner in skeletal muscle.

In the present study, we focus on skeletal muscle as a key regulator of systemic metabolic homoeostasis[7], where lipotoxicity-induced ER stress causes metabolic dysfunction and contributes to the development of metabolic disease[9–13]. This led to our hypothesis that cell non-autonomous signalling may be important in the communication of lipotoxicity-induced ER stress in skeletal muscle. Our work in skeletal muscle may have relevance to the mechanisms of lipotoxcity-induced ER stress in other tissues such as adipose tissue, liver, pancreas and the hypothalamus, where future work may identify that cell non-autonomous ceramide signalling contributes to dysfunction in metabolic disease in these tissues.

Ceramides are highly hydrophobic molecules and therefore unlikely to be secreted via canonical membrane transporters. Our data suggests that EVs represent a vehicle for long-chain ceramide extracellular transport in ER stress. This is consistent with a role for ceramides in small EV biogenesis in which inhibition of neutral sphingomyelinase 2 (nSMase2) reduces small EV secretion[50]. The transport of ceramides by EVs in propagating cell non-autonomous UPR activation further highlights the important role of extracellular vesicles in skeletal muscle endocrine and paracrine signalling[51]. This study describes the signalling of bioactive ceramide lipids delivered through EVs.

We highlight the importance of acyl-chain length in ceramide signalling. Six mammalian CerS isoforms have been identified (CerS1-6). Each CerS enzyme produces ceramides with a unique, but over-lapping, acyl-chain distribution[52]. Long-chain ceramides, ceramide 40:1 and ceramide 42:1 are produced by CerS2 and were identified as the UPR-inducing signals. This is consistent with studies which emphasise the importance of long-chain, rather than C16 and other shorter chain, ceramides in muscle metabolic disease[53,54]. For instance, despite ceramide 36:1

**Fig. 8 Ceramide-induced dihydroceramide accumulation increases UPR gene expression. a** Ceramide (Cer) 40:1 and Cer 42:1 concentrations in C2C12 myotubes treated with either vehicle control (red), 10 μM Cer 40:1 (grey), 10 μM Cer 42:1 (blue) or a combination of both Cer 40:1 and Cer 42:1 (green) (n = 3; one-way ANOVA; control vs 10 μM Cer 40:1, Cer 40:1 $P = 0.0015$; control vs Cer 42:1, Cer 42:1 $P < 0.0001$; control vs Cer 40:1/Cer 42:1, Cer 40:1 $P = 0.0008$, Cer 42:1 $P < 0.0001$; 10 μM Cer 40:1 vs 10 μM Cer 42:1, Cer 40:1 $P = 0.015$, Cer 42:1 $P < 0.0001$; 10 μM Cer 42:1 vs 10 μM Cer 40:1/Cer 42:1, Cer 40:1 $P = 0.0008$; 10 μM Cer 40:1 vs 10 μM Cer 40:1/Cer 42:1, Cer 42:1 $P < 0.0001$). **b** Dihydroceramide (DHCer) 40:0 and DHCer 42:0 concentrations in C2C12 myotubes treated with either vehicle control (red), 10 μM Cer 40:1 (grey), 10 μM Cer 42:1 (blue) or a combination of both Cer 40:1 and Cer 42:1 (green) (n = 3; one-way ANOVA; control vs 10 μM Cer 40:1, DHCer 40:0 $P = 0.0005$; control vs 10 μM Cer 42:1, DHCer 42:0 $P < 0.0001$; control vs 10 μM Cer 40:1/Cer 42:1, DHCer 40:0 $P = 0.0002$, DHCer 42:0 $P < 0.0001$; 10 μM Cer 40:1 vs 10 μM Cer 42:1, DHCer 40:0 $P = 0.0013$, DHCer 42:0 $P < 0.0001$; 10 μM Cer 40:1 vs 10 μM Cer 40:1/Cer 42:1, DHCer 42:0 $P < 0.0001$, 10 μM Cer 42:1 vs 10 μM Cer 40:1/Cer 42:1, DHCer 40:0 $P = 0.0005$). **c** Cer 40:1 and Cer 42:1 in human skeletal muscle cells (HSkMCs) treated with either vehicle control (red), 10 μM Cer 40:1 (grey), 10 μM Cer 42:1 (blue) or a combination of both Cer 40:1 and Cer 42:1 (green) (n = 3; one-way ANOVA; control vs 10 μM Cer 40:1, Cer 40:1 $P = 0.0047$; control vs Cer 42:1, Cer 42:1 $P = 0.0048$; control vs Cer 40:1/Cer 42:1, Cer 40:1 $P = 0.038$, Cer 42:1 $P = 0.0026$; 10 μM Cer 40:1 vs 10 μM Cer 42:1, Cer 40:1 $P = 0.0047$, Cer 42:1 $P = 0.0048$; 10 μM Cer 42:1 vs 10 μM Cer 40:1/Cer 42:1, Cer 40:1 $P = 0.038$; 10 μM Cer 40:1 vs 10 μM Cer 40:1/Cer 42:1, Cer 42:1 $P = 0.0026$). **d** DHCer 40:0 and DHCer 42:0 in HSkMCs treated with either vehicle control (red), 10 μM Cer 40:1 (grey), 10 μM Cer 42:1 (blue) or a combination of both Cer 40:1 and Cer 42:1 (green) (n = 3; one-way ANOVA; control vs 10 μM Cer 40:1, DHCer 40:0 $P = 0.005$; control vs 10 μM Cer 40:1/Cer 42:1, DHCer 40:0 $P = 0.005$, DHCer 42:0 $P = 0.02$; 10 μM Cer 40:1 vs 10 μM Cer 42:1, DHCer 40:0 $P = 0.005$; 10 μM Cer 42:1 vs 10 μM Cer 40:1/Cer 42:1, DHCer 40:0 $P = 0.005$). **e** DEGS1 expression in control HSkMCs (red) and HSkMCs treated with 100 μM palmitate (grey), scrambled control siRNA (con siRNA; white), siRNA against DEGS1 (DEGS1 siRNA; blue) or DEGS1 siRNA and 100 μM palmitate (green) (n = 3; one-way ANOVA; control vs DEGS1 siRNA $P = 0.02$; control vs DEGS1 siRNA + 100 μM palmitate $P = 0.02$; 100 μM palmitate vs DEGS1 siRNA $P = 0.005$; 100 μM palmitate vs DEGS1 siRNA + 100 μM palmitate $P = 0.0047$; Con siRNA vs DEGS1 siRNA $P = 0.013$; Con siRNA vs DEGS1 siRNA + 100 μM palmitate $P = 0.012$). **f** Cer 40:1 and Cer 42:1 normalised concentrations in control HSkMCs (red) and HSkMCs treated with 100 μM palmitate (grey), con siRNA (white), DEGS1 siRNA (blue) or DEGS1 siRNA and 100 μM palmitate (green) (n = 4; one-way ANOVA; control vs 100 μM palmitate, Cer 40:1 $P = 0.0005$, Cer 42:1 $P = 0.0024$; control vs DEGS1 siRNA Cer 40:1 $P = 0.0009$, Cer 42:1 $P < 0.0001$; control vs DEGS1 siRNA + 100 μM palmitate, Cer 40:1 $P = 0.004$, Cer 42:1 $P < 0.0001$; 100 μM palmitate vs DEGS1 siRNA Cer 40:1 $P < 0.0001$, Cer 42:1 $P < 0.0001$; 100 μM palmitate vs DEGS1 siRNA + 100 μM palmitate Cer 40:1 $P < 0.0001$, Cer 42:1 $P < 0.0001$; Con siRNA vs DEGS1 siRNA Cer 40:1 $P = 0.017$, Cer 42:1 $P < 0.0001$; 100 μM palmitate vs Con siRNA, Cer 40:1 $P < 0.0001$, Cer 42:1 $P < 0.0001$). **g** DHCer 40:0 and DHCer 42:0 normalised concentrations in control HSkMCs (red) and HSkMCs treated with 100 μM palmitate (grey), con siRNA (white), DEGS1 siRNA (blue) or DEGS1 siRNA and 100 μM palmitate (green) (n = 4; one-way ANOVA; control vs 100 μM palmitate, DHCer 40:0 $P = 0.047$, DHCer 42:0 $P = 0.036$; control vs DEGS1 siRNA DHCer 40:0 $P = 0.0035$, Cer 42:0 $P = 0.0007$; control vs DEGS1 siRNA + 100 μM palmitate, DHCer 40:0 $P < 0.0001$, Cer 42:1 $P < 0.0001$; 100 μM palmitate vs DEGS1 siRNA + 100 μM palmitate DHCer 40:0 $P < 0.0001$, DHCer 42:0 $P < 0.0001$; Con siRNA vs DEGS1 siRNA DHCer 40:0 $P = 0.0019$, DHCer 42:0 $P = 0.001$; Con siRNA vs DEGS1 siRNA + 100 μM palmitate, DHCer 40:0 $P < 0.0001$, Cer 42:1 $P < 0.0001$, DEGS1 siRNA vs DEGS1 siRNA + 100 μM palmitate, DHCer 40:0 $P < 0.0001$, Cer 42:1 $P = 0.0002$). **h** Activating transcription factor 3 (ATF3), activating transcription factor 4 (ATF4), Heat Shock Protein Family A (HSP70) Member 5 (HSPA5) and ER Degradation Enhancing Alpha-Mannosidase Like Protein 1 (EDEM1) UPR gene expression in control HSkMCs (red) and HSkMCs treated with 100 μM palmitate (grey), con siRNA (white), DEGS1 siRNA (blue) or DEGS1 siRNA and 100 μM palmitate (green) (n = 3; one-way ANOVA; control vs 100 μM palmitate, ATF3 $P = 0.019$, EDEM1 $P = 0.019$; control vs DEGS1 siRNA, EDEM1 $P = 0.024$; control vs DEGS1 siRNA + 100 μM palmitate, ATF3 $P < 0.0001$, ATF4 $P = 0.0001$, HSPA5 $P = 0.009$, EDEM1 $P = 0.0011$; 100 μM palmitate vs Con siRNA, ATF3 $P = 0.019$, EDEM1 $P = 0.019$; 100 μM palmitate vs DEGS1 siRNA + 100 μM palmitate, ATF3 $P < 0.0001$, ATF4 $P = 0.002$, HSPA5 $P = 0.038$; Con siRNA vs DEGS1 siRNA, EDEM1 $P = 0.024$; Con siRNA vs DEGS1 siRNA + 100 μM palmitate, ATF3 $P < 0.0001$, ATF4 $P = 0.0002$, HSPA5 $P = 0.009$, EDEM1 $P = 0.0011$; DEGS1 siRNA vs DEGS1 siRNA + 100 μM palmitate, ATF3 $P < 0.0001$, ATF3 $P = 0.0006$, HSPA5 $P = 0.023$). *$P ≤ 0.05$, **$P < 0.01$, ***$P < 0.001$, ****$P < 0.0001$. Data are expressed as mean ± SEM with individual data points. Source data are provided as a Source Data file.

being the predominate ceramide species in muscle[49], we do not identify this ceramide species in the media of skeletal myocytes in response to palmitate treatment, suggesting it is not a key signal in the palmitate-induced cell non-autonomous activation of the UPR in muscle. The importance of different chain length ceramides in metabolic disease may alter depending on the tissue type and context. Through the use of heterozygous transgenic catalytic inactivation of CerS2, a model which represents a more physiological condition than total loss of function, preventing compensatory increases in shorter chain ceramides, we are able to directly manipulate the concentration of the long-chain ceramide signals in vivo and show that this modulates the expression of UPR genes in muscle.

Lipidomic profiling of ceramide-treated myotubes showed increases in cognate dihydroceramides. Inhibition of Des1, or knockdown of DEGS1, increased the dihydroceramide:ceramide ratio, and exacerbated palmitate-induced ER stress. Although a role for dihydroceramide accumulation in ER stress has been reported in several cancers including gastric carcinoma and glioblastoma[35,55,56], here we present data suggesting the conversion of ceramide to dihydroceramide. We present evidence of an emerging role for dihydroceramides in the induction of ER stress.

This study may have repercussions beyond the understanding of ER stress. Mitochondria also contain a discreet UPR initiated when mitochondrial integrity and function are impaired. This mitochondrial UPR may also be regulated by paracrine signals with recent work identifying Wnt proteins in paracrine activation of the mitochondrial UPR[57]. However, a role for bioactive lipids in this process remains unexplored.

We identify a mechanism for cell-cell lipid-mediated paracrine communication of UPR activation in ER stress. This has consequences across pathophysiological activation of ER-stress and cell-cell signalling. In addition, by defining the mechanisms of ER-stress activated long-chain ceramide production, packaging, secretion, and UPR activation we highlight potential targets for therapeutic intervention in lipotoxicity-associated metabolic diseases.

## Methods

**C2C12 myoblast culture and differentiation.** C2C12 myoblasts (91031101, Sigma Aldrich) were grown to confluence in DMEM media (D6429, Sigma Aldrich) containing 10% foetal bovine serum (FBS, Sigma Aldrich) and 1% penicillin-streptomycin (P/S, Sigma Aldrich). Approximately 100,000 cells were seeded per well in 12-well collagen I-coated plates (Corning Biocoat, 734-0295). Myoblasts were then differentiated to myotubes from time of plating in differentiation media, which constituted DMEM containing 10% horse serum (HS, Sigma Aldrich) and

1% P/S. Cells were differentiated for 8 days. After 2 days of differentiation, cells were treated with either 200 μM palmitate, or the BSA vehicle control, for 6 days or ceramide 40:1 or 42:1 for the final 10 h. Media was changed every day during differentiation. Furthermore, cells were co-treated with the following pharmacological inhibitors: 10 μM AMG PERK 44 (Tocris Bioscience), 10 μM 4μ8c (Tocris Bioscience), 10 μM Fumonisin B1 (Tocris Bioscience), 5 μM Fenretinide (Tocris Bioscience), 10 μM Myriocin (Sigma Aldrich).

**Primary human skeletal muscle cell culture and differentiation.** HSKMCs were purchased from Cell Applications Inc (S150a-05a). They were grown to confluence in Skeletal Muscle Cell Growth Medium (Cell Applications) in collagen I-coated 12-well plates (approximately 100,000 cells were seeded per well). Once confluent, HSKMCs were differentiated for 6 days using Skeletal Muscle Differentiation Media (Cell Applications, 151D-250). During differentiation cells were treated with either 50 μM or 100 μM palmitate, or the BSA vehicle control, to set up a chronic model of ER stress, or with ceramide 40:1 or 42:1 for the final 10 h of differentiation. Media was changed every 2 days during differentiation.

Informed consent was obtained from donors. The cells were approved and complied with ethics according to:

– Collection, generation, research purpose and sale: Cell Applications, Inc. 5820 Oberlin Dr. Suite 101, San Diego, CA 92121.
– Use in compliance with Human Tissue Act (UK) by Leeds Institute of Cardiovascular and Metabolic Medicine, University of Leeds, Leeds, LS2 9JT UK.

**Conjugation of palmitate to BSA.** The conjugation of palmitate to BSA was carried out as previously described[58,59]. PBS (75 ml) was heated to 37 °C. Fatty-acid-free BSA (20 g) (Sigma Aldrich) was added to the PBS. Separately deionised water (15 ml), 95% ethanol solution (10 ml) and Na2CO3 (0.106 g) (Sigma Aldrich) were mixed under nitrogen. Palmitate (0.2115 g) (Sigma Adrich) was added, and heated to ~78 °C to evaporate the ethanol. The fatty-acid solution was added to the albumin solution and mixed. The palmitate/albumin mixture was dialysed (Spectra/Por dialysis tubing; Molecular weight cut off 6000-8000) in deionised water overnight at 4 °C.

**Preparation of ceramides.** Ceramide 40:1 (N-(docosanoyl)-ceramide; Ceramide d18:1/22:0) (860501) and ceramide 42:1 (N-(tetracosanoyl)-ceramide; Ceramide d18:1/24:0) (860524) were purchased from Avanti Polar Lipids Inc. Preparation of ceramides followed standard protocols in the literature[60,61]. Ethanol and dodecane were mixed at a ratio of 98:2, followed by vortexing and pre-warming to 37 °C. Meanwhile, ceramide 40:1 and ceramide 42:1 (Avanti Polar Lipids Inc) were dissolved in chloroform-methanol 1:1. The required volume was then dried down under N₂ gas. The pre-warmed ethanol-dodecane mixture was added to the dried ceramide 40:1 and ceramide 42:1 such that the final stock concentration was 2.5 mM. This mixture was thoroughly vortexed and incubated at 37 °C for a further 20 min followed by further vortexing. The stock solution was then diluted to the required concentration using ethanol/dodecane. The stock solution of ceramide 40:1 or ceramide 42:1 were diluted in cell culture medium (37 °C) to the required treatment concentration followed by vortex mixing. Ethanol/dodecane was included as a vehicle control at the same concentration as the treatment.

**Preparation of lysophosphocholines and diacylglycerols.** LPC 16:0 (855675), LPC 18:0 (855775), DG 32:0 (800816) (Avanti Polar Lipids) were dissolved in 50% ethanol in sterile water at a stock concentration of 50 mM. LPC and DG stocks were made up fresh for each media change during cell culture experiments. The stock solution of LPC 16:0, LPC 18:0 and DG 32:0 were diluted in cell medium (37 °C) to the required treatment concentration followed by vortex mixing. Ethanol/water was included as a vehicle control at the same concentration as the treatment.

**Conditioned media collection and processing.** Following treatment with either palmitate or the BSA vehicle for 6 days, culture media was removed and cells washed with PBS. Serum-free media, free of any agonists, was then added for 24 h. This conditioned media was collected and frozen at −80 °C. For boiling experiments, conditioned media was incubated at 100 °C in a water bath for 5 min. For molecular weight cut off dialysed experiments, conditioned media was dialysed for 24 h using molecular weight cut off dialysis of 1000 Da and deionised water as dialysate (Spectrum™ Spectra/Por™ 7 Membrane Tubing 1000 Dalton molecular weight cut off; Fischer Scientific). Dialysate (containing the dialysed molecules of 1 kDa and below) was dried down under N₂, and reconstituted in equivalent volumes of serum-free media before transfer to naïve myocytes.

For extracted conditioned media experiments, 600 μL of methanol:chloroform (1:1) was added to 1 mL of media. Samples were centrifuged (16,100 g, 10 min) and the aqueous and organic layers were separated and dried. The aqueous fraction was resuspended in 1 mL serum-free media prior to transfer on untreated myotubes. The organic fraction was reconstituted in a small volume of ethanol, and diluted in 1 mL of serum-free media prior to transfer. For transfer experiments, conditioned media was added to differentiated C2C12 or human primary skeletal myotubes for 10 h.

**siRNA PERK, CERS2 and DEGS1 knockdown.** FlexiTube siRNA against *EIF2AK3* (PERK) (SI00069062), *CERS2* (SI04711588) and *DEGS1* (SI03109946), AllStars negative control siRNA, and HiPerFect Transfection Reagent were purchased from Qiagen. Human myocyte transfection was performed according to the manufacturer's instructions (75 ng siRNA, 3 μL transfection reagent per well, 60 nmol/L final siRNA concentration for PERK, 30 nmol/L final siRNA concentration for *CERS2* and *DEGS1*) on days 2, and 4 of differentiation.

**Cers2 H/A mice.** Het *Cers2* H/A mice, Homo *CerS2* H/A mice and corresponding wild type littermate controls were generated as previously described[29]. All animals were kept under standard housing conditions in a specific pathogen free environment at room temperature with humidity maintained at 40–60% and a 12 h dark to light cycle, and water and food ad libitum. At eight weeks old experimental animals were anaesthetised, blood was collected by cardiac puncture and soleus skeletal muscle was dissected, snap-frozen and stored at −80 °C. Animals were raised and dissected at the University of Bonn, performed in accordance with the instructions of local and state authorities.

**Western diet mouse model.** Five-week-old male mice (C57Bl6j) were purchased from Harlan Laboratory Ltd. Four animals were housed per cage and had an acclimatisation period of 7 days. The temperature was maintained at 20 ± 4 °C with humidity maintained at 40–60% and a 12 h light/dark cycle. Mice were fed either a western diet (WD, TD88137) or a low-fat control diet (LFD, TD08485), for 12 weeks, with diets supplied by Teklad Custom Research Diets (Envigo). Each mouse was subject to an intraperitoneal glucose tolerance test (IP GTT) at the end of week 10, as well as an insulin tolerance test (ITT) at the end of week 11. Body composition was analysed by time domain nuclear magnetic resonance (TD-NMR) at the end of week 12.

Following the conclusion of the study, mice were sacrificed by cervical dislocation, and subsequently dissected. Tissue was flash frozen in liquid nitrogen and stored at −80 °C.

All animal studies complied with national and local ethical regulations for animal research. Animal studies were regulated under the Animals (Scientific Procedures) Act 1986 following ethical review by the University of Cambridge Animal Welfare and Ethical Review Body. All procedures were carried out in accordance with U.K. Home Office protocols by a UK Home Office Personal License Holder.

**Intraperitoneal glucose tolerance tests.** Intraperitoneal glucose tolerance tests were performed as previously described[62]. Mice were fasted for 8 h with access to water prior to baseline glucose measurement. Administration of glucose (Sigma-Aldrich) was performed by intraperitoneal injection (glucose 1.5 mg/g of body weight; glucose solution 150 mg/ml). Blood was obtained from the tail vein immediately prior to glucose injection and then at 10, 20, 30, 60, 90 and 120 min post injection. Glucose levels were measured using a Bayer Contour Glucose Meter (Bayer Healthcare).

**Human skeletal muscle biopsies.** 75 eligible and consecutive patients undergoing routine pacemaker therapy at Leeds General Infirmary volunteered to participate in the study following written informed consent. Eligible patients were grouped into two age and sex-matched cohorts based on the presence (n = 22) or absence (n = 53) of T2DM. The T2DM cohort had a previous diagnosis of T2DM (>3 months) and/or was receiving treatment for diabetes. In such cases, T2DM was defined as a documented history of diabetes, fasting plasma glucose ≥ 7.0 mmol/L, plasma glucose ≥ 11.1 mmol/L 2 h after the OGTT or a glycated haemogolobin (HbA1c) ≥ 6.5% (≥48 mmol/L). Patient demographic data is given in Supplementary Table 1.

Human skeletal muscle (*pectoralis major*) biopsies were obtained from eligible participants during their pacemaker operation. Pacemaker procedures were otherwise entirely routinely carried out: lidocaine was initially injected to anaesthetise the area and a small incision was made under the left clavicle. During creation of the pre-pectoral pocket for the generator above the *pectoralis major* a portion of the muscle (approximately 100 mg) was sampled. This sample was immediately placed into a 1.5 mL Eppendorf tube and snap frozen in liquid nitrogen. The study was approved by the Leeds West Research Ethics Committee (11/YH/0291) and Leeds Teaching Hospitals R&D committee (CD11/10015) and conforms to the Declaration of Helsinki.

**RNA handling and reverse transcription quantitative polymerase chain reaction.** RNA was extracted and purified in situ, using the RNeasy mini kit (Qiagen), following the manufacturer's instructions. cDNA was synthesised by a two-step process with the RT² First Strand Kit (Qiagen), following the manufacturer's instructions using a PrimeG thermal cycler (Techne). Samples were analysed using a Step One Plus Real Time Cycler (Applied Biosystems), quantified using the ΔΔ-CT method with expression normalised to the housekeeping gene.

For murine samples, *Rn18s* (PPM72041A) was used as the housekeeping gene, while *ACTB* (PPH00073G) was used for analysis of primary HSKMCs. The induction of ER stress and biosynthesis of long-chain ceramides was assessed throughout this study using primers for *Atf3* (PPM04669C), *Atf4* (PPM04670E), *Hspa5* (PPM03586B), *Edem1* (PPM26189B), *CerS2* (PPM32482A) and *CerS4* (PPM28333A) for murine samples, and *ATF3* (PPH00408C), *ATF4* (PPH02016A), *HSPA5* (PPH00158E), *EDEM1* (PPH12935B), *EIF2AK3* (PERK) (PPH10874A), *CERS2* (PPH16029F) and *DEGS1* (PPH16415A) for human samples. Data were collected and analysed using StepOne™ Software (version 2.1 Applied Biosystems). Details of primers are given in Supplementary Table 2.

**Metabolite extraction**. Metabolites from cell pellets, media (1 mL), tissue (approximately 25 mg) and plasma (20 µl) were extracted using a modified Bligh and Dyer method as previously described[58]. Methanol-chloroform (2:1 600 µl) was added to the samples. Tissue and cells were homogenised in a TissueLyser II (Qiagen) (2 min, 25 Hz) and the samples were sonicated for 15 min. Chloroform-water (1:1) was then added (200 µl of each). Samples were centrifuged (16,100 *g*, 20 min) and the aqueous and organic phases were separated. For each sample, 250 µL of a deuterated internal standard mixture was added to organic fraction samples prior to drying under nitrogen. The internal standard mix was composed of deuterated standards sourced from Avanti Polar Lipids (C16-d31 ceramide, 16:0-d31-18:1 phosphatidic acid, 16:0-d31-18:1 phosphatidylcholine, 16:0-d31-18:1 phosphatidylethanolamine, 16:0-d31-18:1 phosphatidylglycerol, 16:0-d31-18:1 phosphatidylinositol, 14:0 phosphatidylserine-d54, and 16:0-d31 sphingomyelin) and CDN Isotopes/QMX Laboratories (18:0-d6 cholesteryl ester, 15:0-d29 fatty acid, 17:0-d33 fatty acid, 20:0-d39 fatty acid, 14:0-d29 lysophosphatidylcholine-d13, 45:0-d87 triacylglyceride, 48:0-d83 triacylglyceride, and 54:0-d105 triacylglyceride). The internal standards were dissolved in methanol/chloroform (1:1) at a final concentration of 2.5 µg/mL.

**LC-MS lipidomic profiling**. Metabolites from the organic phase were reconstituted in 50 µL of methanol/chloroform (1:1) and vortexed thoroughly. Ten µL of the sample was then diluted into 190 µL of propan-2-ol/acetonitrile/water (IPA:ACN:H₂O; 2:1:1). For both positive and negative ionisation modes, 5 µL of sample was injected onto a C18 CSH column, 2.1 × 100 mm (1.7 µm pore size; Waters), which was held at 55°C using an Ultimate 3000 UHPLC system (Thermo Fisher Scientific). The mobile phase comprised solvents A (ACN/H₂O; 60:40) and B (ACN/IPA; 10:90), run through the column in a gradient (40% B, increased to 43% B after 2 min, 50% B at 2.1 min, 54% B at 12 min, 70% B at 12.1 min, raised to 99% B at 18 min before returning to 40% for 2 min). The total run time was 20 min, with a flow rate of 0.400 µL/min. In positive mode, 10 mM ammonium formate (Fisher Scientific) was added to solvents A and B. In negative mode, 10 mM ammonium acetate (Sigma Aldrich) was added to the solvents. Solvent additives were chosen based on current literature[63].

Mass spectrometry was then carried out using the LTQ Orbitrap Elite Mass Spectrometer (Thermo Fisher Scientific) in positive (for detection of LPCs and DGs) and negative mode (for detection of ceramides and dihydroceramides). Metabolites were ionised by heated electrospray before entering the spectrometer. The source temperature was set to 420 °C, and the capillary temperature to 380 °C. In positive mode, the spray voltage was set to 3.5 kV. Negative mode spray voltage was 2.5 kV. Data were collected using the Fourier transform mass spectrometer (FTMS) analyser. The resolution was set to 60,000 and the data were obtained in profile mode. The full scan was performed across an *m/z* range of 110–2000.

To overcome limitations in LC-MS annotation of isobaric lipid species, and to facilitate the identification of the fatty acyl chains esterified to complex lipids, a tandem mass spectrometry fragmentation method was used in a second scan event. Ions were fragmented using collision-induced dissociation at a normalised collision energy of 35. Precursor ions were selected from a mass list, with the most intense ion on the list fragmented in each scan. The minimum signal required was set to 5000 counts and the isolation width was set to 1. The activation time was 10 ms and the activation Q was set to 0.25. Fragmentation spectra were acquired in centroid mode at a resolution of 15,000 using the FTMS analyser. The subsequent fragmentation spectra were used to confirm the identity of the lipid assignments. Assignment of lipid identities for select species were also confirmed by comparison with commercially available standards for LPC 16:0 (855675), LPC 18:0 (855775), DG 32:0 (800816), ceramide 40:1 (N-(docosanoyl)-ceramide; Ceramide d18:1/22:0) (860501) and ceramide 42:1 (*N*-(tetracosanoyl)-ceramide; Ceramide d18:1/24:0 (860524), dihydroceramide 42:0 (860628) (Avanti Polar Lipids Inc) and dihydroceramide 40:0 (sc-210991) (Santa Cruz Biotechnology) and therefore meet Metabolomics Standards Initiative level one identification criteria[64,65].

Chromatograms were collected using Xcaliber (version 2.2; Thermo Scientific). For data processing, spectra were converted from Xcalibur.raw files into.mzML files using MS Convert (version 3; Proteowizard) (https://proteowizard.sourceforge.io/tools.shtml#) for analysis by XCMS (https://xcmsonline.scripps.edu/landing_page.php?pgcontent=mainPage) (version 3.4, Scripps Institute) within R studio. XCMS software was used to process data and identify peaks. Peaks were identified based on an approximate full width at half maximum of 5 s and a signal to noise threshold of 5. To improve peak identification, peaks had to be present in a minimum of 25% of the samples. Peaks were annotated by accurate mass comparison to the LipidMaps database (https://www.lipidmaps.org/)[66] and

confirmed by data-directed fragmentation and comparison with commercial standards as described above. Intensity was normalised to an appropriate internal standard, and cell number (or dry protein pellet weight for animal tissue).

**Extracellular vesicle isolation and quantification**. Conditioned media was centrifuged first at 1000 *g* for 5 min and the pellet discarded. This was followed by a second spin at 10,000 *g* for 20 min to pellet large EVs. The supernatant was then spun at 110,000 *g* for 75 min. The supernatant was discarded and the pellet resuspended in a small volume of PBS prior to a final ultracentrifugation at 110,000 *g* for 75 min to pellet small EVs. The supernatant was discarded and the pellet resuspended in a small volume of PBS. EV particles were quantified using a NS300 system (Malvern). Six movies of 60 seconds were acquired per sample at room temperature, using a camera level of 15. The processing threshold was set to 6. Data was collected, and particle number and size distribution of each sample was analysed, using NTA3.0 software (NanoSight, Malvern Panalytical).

**Preparation of EV protein lysate for protein quantification and SDS-PAGE electrophoresis**. After isolation, an aliquot of the EV sample was lysed using RIPA buffer (containing protease (cOmplete ULTRA, 05-892 791 001 Roche, supplied by Merck Dorset, UK) and phosphatase inhibitors (PhosSTOP, 490 684 5001 Roche, supplied by Merck Dorset, UK) on ice for 30 min. The samples were centrifuged at 13,000 *g* for 15 min and the clarified supernatant used for protein quantification using a bicinchoninic acid protein assay kit (ThermoFisher Scientific; 23227) as per the manufacturer's instructions.

**SDS-PAGE and western blot analysis**. Samples (containing 25 µg protein for EV analysis, 30 µg for UPR markers) were resolved by electrophoresis through a 4–12% Bis-Tris gradient gel (Invitrogen, Paisley UK). The resolved proteins were transferred to nitrocellulose membrane (Bio-rad, Hertfordshire UK) and blocked by incubation for 1 h with TBS containing 0.1% (v/v) Tween 20 at room temperature. Incubations with primary antibodies (TrfR (1:1000, H68.4), GAPDH (1:1000, 1D4) —Invitrogen, Paisley UK), CPT1a (1:1000, EPR21843-71-2F), SOD (1:1000, 71G8), XBP-1s (1:1000, E9V3E), CHOP (1:1000, L63F7) (All Cell Signalling Technologies, London UK) and peroxidase-conjugated secondary antibodies (sheep anti-mouse (1:3000, NA931V, GE Healthcare, Amersham, UK) for GAPDH and TrfR and goat anti-rabbit (1:3000, G21234, Invitrogen, Paisley UK) for all other antibodies) were performed in the same relevant solution as the blocking step. Bound peroxidase conjugates were visualised using an enhanced chemiluminescence detection system (WBKLSO500 Millipore, supplied by Merck Dorset, UK). Images were recorded using Syngene GBox gel doc system and GeneSys software version 1.8.3 (Syngene, Cambridge UK) for image collection and densitometry analysis. Uncropped and unprocessed blot images are presented in the Source Data file.

**Cellular confocal microscopy imaging of ER swelling**. HSkMCs were plated onto collagen-coated coverslips at a density of $4 \times 10^4$ cells/ml. Once 90% confluent, the growth media was changed to differentiation media and the cells were allowed to differentiate for 5 days. On day 5 of differentiation, CellLight ER-RFP BacMam reagent (C10591, Invitrogen, Paisley UK) was added to the cells at a multiplicity of infection of 30. After 18 h of treatment (day 6 of differentiation), 10 µM ceramide 40:1, 10 µM ceramide 42:1 or 10 µM of both ceramide 40:1 and 42:1, or vehicle only (98:2 Ethanol:Dodecane) was added to the cells. The coverslips were fixed in 4% paraformaldehyde for 10 mins at room temperature after 10 h of exposure, washed and mounted onto slides using fluoromount-G containing DAPI (0100-20, Southern Biotech, Birmingham, Alabama).

HSkMCs were then imaged using a ZEISS LSM 880 confocal microscope in the Cy3 (excitation 555 nm) and DAPI channels (excitation 358 nm, emission 460 nm) to visualise ER and nuclei respectively. Images were obtained using a 40X objective lens.

The amount of ER-specific staining was recorded using the ZEN software (Version 2.60, ZEISS). 25 fields of view were imaged from 3-4 slides taken from 3 biological replicates. Each view had a threshold applied to establish the ER-specific staining and a mean fluorescence intensity recorded. This was then corrected to the number of nuclei present in that field.

**Statistical analysis**. Multivariate data analysis was performed using SIMCA-P + 13.0 (Umetrics AB, Umeå, Sweden) as previously described[58]. LC-MS data sets were mean-centred and Pareto-scaled prior to analysis. Data sets were analyzed using principal components analysis and partial least squares-discriminant analysis. Lipids responsible for clustering or regression trends within the pattern recognition models were identified by interrogating the corresponding loadings plot. Lipids identified in the variable importance in projections/coefficients plots were deemed to have changed globally if they contributed to separation in the models with a confidence limit of 95%.

Univariate statistics was used to assess the significance of changes in individual metabolites. Sample sizes were calculated using power calculations. Samples were randomly assigned to experimental groups. Group variance was analysed with an *F* test. Statistical significance was assessed using one-way ANOVA (Tukey's, Holm–Sidak's and Dunnett's post hoc test) or two-tailed Student's *t* test, as

detailed. In each case, $n \geq 3$ and the significance level was set to $P \leq 0.05$. All univariate analysis was conducted using GraphPad Prism (version 6) software.

**Reporting summary**. Further information on research design is available in the Nature Research Reporting Summary linked to this article.

## Data availability

The Lipid Maps database was used in lipid identification from mass spectrometry data (https://www.lipidmaps.org/). All other data generated or analysed during this study are included in this published article (and its supplementary information files). Source data are provided with this paper.

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

## Acknowledgements

B.D.M. was funded by an MRC PhD studentship. L.D.R. is supported by the Diabetes UK RD Lawrence Fellowship (16/0005382). This research was supported by funding to L.D.R. from the Biotechnology and Biological Sciences Research Council (BB/T004231/1, BB/R013500/1) and Diabetes UK (18/0005846) and to J.L.G. from the Biotechnology and Biological Sciences Research Council (BB/R013500/1) and the Medical Research Council (MC_UP_A090_1006, MC_PC_13030, MR/P011705/1 and MR/P01836X/1). The authors are indebted to Prof Vidal-Puig, Dr Samuel Virtue, and the MRC Metabolic Diseases Unit (the Disease Model Core and the Biochemistry Assay Lab - MC_UU_00014/5) at the WT/MRC Institute of Metabolic Science for assistance with the WD murine experiments.

## Author contributions

B.D.M. designed and carried out the majority of experiments, interpreted the results and wrote the paper with input from all co-authors. D.F.A. performed LC-MS analysis on CerS2 H/A mice and EV isolations. L.H. performed CerS2 H/A mouse experiments. H.N.D. designed and performed siRNA knockdown experiments in human myocytes. N.T.W. designed and performed protein assays for UPR protein analysis and extracellular vesicle characterisation. S.A.M. provided experimental support in mass spectrometry experiments. A.D.V.M. and A.W. performed gene expression analysis for CerS2 mouse studies. T.S.B. contributed to the design of protein quantitation experiments. F.W.B.S. and M.V. provided tissue from murine western diet studies. K.W. provided human skeletal muscle biopsies. RB led CerS2 H/A mouse research. J.L.G. supported experiment design and interpretation and secured funding. L.D.R. led the study and experiment design, interpretation and manuscript writing and secured funding. All authors commented on and approved the final manuscript.

## Competing interests

The authors declare no competing interests.
