## [Peer Review File · Nature Communications]

Reviewers' Comments:

Reviewer #1:

Remarks to the Author:

McNally and colleagues have investigated the exciting concept of skeletal muscle ceramide export to induce UPR activation in neighbouring cells. By using in vitro culturing experiments, the authors have performed co-culturing experiments to determine that exosomes released from palmitate-treated myotubes contained a distinct ceramide profile with very long ceramide species being solely responsible for the UPR activation in neighbouring cells. Whilst this is an intriguing concept, the experiments performed do not fully convince the reviewer with the conclusions drawn as there are numerous omissions in the collected data set and what is presented in the manuscript.

Major concerns:

1. Lipidomic profiles – all lipidomic profiles need to be presented in this manuscript, specifically all the dihydro- and ceramide profiles. The reason for this is that the lipidomic profiles of in vitro muscle cultures differ from in vivo skeletal muscle samples as well as primary muscle cultures. Therefore, the full profiles for all lipidomic analysis need to be presented as to ascertain if there are changes in the full profiles as opposed to only C40:1 and C42:1. The other concern that these are not being presented is that the CerS2 H/A animals are known to have compensatory increases in C34:1 ceramide when C40:1 and C42:1 are decreased. This cannot be ascertained in experiments as the data is not presented here.
2. The authors do not acknowledge that C36:1 are the predominate ceramide species in skeletal muscle. This has been documented by numerous animal and human studies. This is not acknowledged in the manuscript and the measurement of this ceramide species is not shown or discussed.
3. The ceramides C40:1 and C42:1 are presented to induce key gene markers of the UPR arms of ER stress. Does this then translate into function? No experiments are presented to show more than gene expression, does protein expression also change? Can ER stress be visualised or any functional experiments be performed to confirm gene expression findings?
4. The conclusion of PERK regulating sphingolipid synthesis is based on one weak experiment. Authors must prove that this connection is regulated by PERK by using PERK KO myotubes and confirming that C40:1 and C42:1 ceramides are altered.
5. The experiments regarding the CerS2 H/A mice do use a good animal model, they do not adequately answer the questions proposed or conclusions drawn. The CerS2 H/A mice are known to have a compensatory increase in C34:1 ceramides in response to the reduction of C40:1 and C42:1 ceramides. Does this occur in the soleus muscles used in these experiments? Also, the plasma measures of C40:1 and C42:1 ceramides are not adequate to conclude that the plasma and soleus muscle ceramide concentrations were related. The plasma concentrations of ceramides would be predominately derived from the liver, not the muscle. Therefore, experiments extracting muscle specific exported ceramides need to be performed or exosomes/EVs derived from muscle cultures could be performed in coculturing experiments to confirm that the C40:1 and C42:1 ceramides do indeed activate the muscle UPR.
6. The conclusions drawn from Figure 7 do not fit the data presented. The gene expression pattern of intracellular ceramides produced by palmitate does not fit with the UPR gene expression. Authors conclude that dihydroceramides drive UPR however, the UPR genes increase only with palmitate or palmitate + fenretinide. If ceramides were solely responsible for UPR activation, then why is there no increase in UPR gene expression in fenretinide only controls? Fenretinide would inhibit all dihydroceramides and therefore should alter some of the UPR gene expression as the DES1 inhibitor increases the intracellular C40:1 and C42:1 ceramides around 8 fold in Figure 7f.

Minor concerns:

7. Methods are missing for the addition of ceramide derivatives to the cell culture media. It is not clear how the lipophobic ceramides were added to cells, or if they even incorporated into the cells.
8. The introduction lacks a section on ceramides and specifically, ceramides in skeletal muscle.
9. The authors use the term "lipotoxic lipid species" however, numerous studies have now shown that most metabolic tissues are remarkably plastic and are able to adapt to increased lipid concentrations of TAGs, or even distinct ceramide species.

10. Why does the palmitic acid concentration change in each experiment? Ie 50um, 100um and 200um concentrations are used throughout different experiments but it is not explained why.
11. The controls for lipid extraction experiments regarding the culture media are lacking. Control lipid extracts and control organic extracts of media in Figure 1e should be performed for comparisons, not just comparisons of control organic to palmitate media. Therefore, please include the aqueous extract gene expression to confirm it is palmitate media lipid extracts that are increasing gene expression.
12. The data presented in Figure 3C does not appear to be different as stated in the text.
13. Line 175 – Needs to be corrected to “multiple Ceramide synthases”.
14. It would be informative for the authors to include the expression levels of the Ceramide synthases and other enzymes of the de novo ceramide synthesis pathway to demonstrate that the “increased generation of ceramides through the de novo synthesis pathway results in lipotoxicity”.
15. Line 217-8. The statement that the bioactive lipid profiles of EVs has not previously been reported is false. Please refer to the published article.

Burrello J, Biemmi V, Dei Cas M, Amongero M, Bolis S, Lazzarini E, et al. Sphingolipid composition of circulating extracellular vesicles after myocardial ischemia. *Scientific reports*. 2020;10(1):16182.

16. Line 226. The authors state that the microvesicle release was unaffected by CERT inhibition but Figure 6c shows that the CERT inhibitor HPA-12 increased microvesicle particles in controls.
17. Ceramide concentrations were not shown in the depleted media experiments, therefore how are we to know that ceramide levels were not altered in these experiments and therefore would account for the reduction in UPR signalling?

Reviewer #2:

Remarks to the Author:

In the current manuscript, the authors reported that the long-chain ceramide was extracellularly transported via EVs in ER stress. They also showed that the ceramide, which was transported by EVs, induced UPR activation in naïve myotubes. Overall, the experiment was well performed, and the manuscript was well organized. However, there are several technical issues that need to be addressed below.

1. MISEV 2018 guidelines currently encourage to use of operational terms for EV subtypes that refer to physical characteristics of EVs, such as size (“small EVs” and “medium/large EVs,” with ranges defined, respectively). The reviewer recommends changing the expression of “exosome” and “microvesicles” to “small EVs” and “large EVs”, respectively.

2. The manuscript needs more information of the EV characterization. Please refer to the following, and add more information to the manuscript.

- 2-1. In Fig.2 b and c, although the authors analyzed particle number of EVs, quantification of EVs should be addressed by at least 2 methods: protein amount, particle number, lipid amount. Please analyze the protein or lipid amount of EVs-derived from control and palmitate treated myocyte.

- 2-2. At least one protein of categories 1-3 (1. Transmembrane or GPI-anchored proteins, 2. Cytosolic proteins recovered in EVs, 3. Major components of the non-EV co-isolated structure) must be analyzed to demonstrate the EV nature and the degree of purity of an EV preparation. (Please refer to Table 3 of MISEV2018.)

3. In Fig.6 d-f, the authors should investigate whether the mRNA levels of UPR genes are upregulated in a dose-dependent manner by treating EVs-derived from palmitate treated myocyte.

Reviewer #3:

Remarks to the Author:

The manuscript describes a study focusing on the connection of long-chain ceramides, lipotoxicity and reticulum stress in skeletal muscle. Overall, the study is well planned experimentally and it gives novel insights of the role of ceramides. However, there are several areas in the manuscript that require further consideration. The strength of the work is that it combines cell-based studies with mice and human data including also a more mechanistic study. However, in humans, only muscle biopsy data was shown; it would be relevant to show (as is done for mice) the data also on the matching blood-based samples. The conditions chosen for the experiments (concentrations, exposure times) have not been well justified/explained, making it more challenging to evaluate whether the results present a realistic situation. In addition, the literature cited misses several key publications in the area. For example, an article by S. Choi et al (2018) showing that palmitate increased several long-chain ceramides in intestinal epithelial cells, while a shorter chain fatty acid myristate resulted in changes in shorter chain ceramides; the study also concentrating on ER stress. Another limitation is that the study focuses on the skeletal muscle only, while discussing the link with metabolic syndrome. The other vital tissues/organs are not covered, and there is also very little discussion on that. This would be important, as the results from the mice indicated larger changes in circulation that were observed in the muscle, thus suggesting that there are other tissues that may have a stronger contribution.

Detailed comments:

Results, first chapter: Here, the presentation of the results is not fully clear, and there is a mix of discussion and results. The discussion parts should be moved to the Discussion section. The explanation of the myotube differentiation and exposure is not well described, and as it now reads, it sounds that the same experiment was carried out twice. In experimental section it is stated that the differentiation was first carried out for 2 days, after which the myotubes were exposed to palmitate but in the results, it is stated that the differentiation is done for six days and then they were exposed to palmitate. It is also unclear how the palmitate concentrations and the exposure times were chosen; this is crucial information and should be explained in detail. Figure 1 b vs c, the difference for several genes seem to be larger with lower palmitate concentration, except for Atf4 and EDEM1. Thus, it would be of interest to see the impact at even lower concentration.

Page 6, lines 104-113: The treatment of the media for protein denaturation: boiling of the media will have a major impact on the metabolites, as many of them are (particularly bioactive lipids) not stable at elevated conditions so this is not most likely a suitable method. Protein precipitation with an organic solvent is better. In the experimental section, it is stated that also the aqueous fraction was used in the exposure studies, but there are no results stated in the results section. Please explain also what was the impact of the aqueous fraction. Did you consider a more precise separation of the lipid fraction, using e.g. LC based fractionation (e.g. by HILIC)? This would enable the separation of the other species that were showing difference in the overall lipid profiles (DG, LPC), but did not seem to be the main drivers.

Page 6-7: The concentrations of DG32:0 and LPC16:0 and ceramides in the treatment of myotubes, how were these selected? Based on the experimental results?

Page 8, Fig 3: The enrichment of the ceramides in murine samples, the difference for murine was larger in serum than in muscle for Cer 40:1, suggesting that this difference is related to something else than changes in the muscle metabolome. In humans, no results are shown for blood-based samples, this should be added in the results. In humans, the difference in the muscle does not seem to be significant?

Page 11-12, analysis of ceramides and dihydroceramides. How did you verify the identity of the dihydroceramides? Overall, more data on the accuracy of the lipidomic measurements should be given here.

Discussion:

Here, the discussion is focusing on the ceramides, however, in the current study the focus was only on the muscle tissue. Liver is one of the key organs in lipid homeostasis and it is tightly linked with the metabolic diseases. Why was to focus solely on skeletal muscle? What is the potential contribution of the other tissues, particularly liver?

Page 11, line 217-218: The authors state that the partition of bioactive lipids into EVs has not been previously reported. This statement is not very clear. There are several studies of the lipid composition of the EVs, and the difference in the composition e.g. in cancer cells.

Page 15, line 309-3010: The authors state that the dihydroceramides have been linked with gastric carcinoma cells. There are other studies as well, linking these compounds with ER stress, e.g. in glioblastomas

Supplementary Figure 2. The text of lipid species is too small to be readable

REVIEWER RESPONSES

(Page and line numbers refer to position in the marked copy of the manuscript.)

Reviewer #1 (Remarks to the Author):

McNally and colleagues have investigated the exciting concept of skeletal muscle ceramide export to induce UPR activation in neighbouring cells. By using in vitro culturing experiments, the authors have performed co-culturing experiments to determine that exosomes released from palmitate-treated myotubes contained a distinct ceramide profile with very long ceramide species being solely responsible for the UPR activation in neighbouring cells. Whilst this is an intriguing concept, the experiments performed do not fully convince the reviewer with the conclusions drawn as there are numerous omissions in the collected data set and what is presented in the manuscript.

We are pleased that the reviewer found our work exciting and intriguing and thank them for their thoughtful comments which have enhanced and strengthened our manuscript.

Major concerns:

1. Lipidomic profiles – all lipidomic profiles need to be presented in this manuscript, specifically all the dihydro- and ceramide profiles. The reason for this is that the lipidomic profiles of in vitro muscle cultures differ from in vivo skeletal muscle samples as well as primary muscle cultures. Therefore, the full profiles for all lipidomic analysis need to be presented as to ascertain if there are changes in the full profiles as opposed to only C40:1 and C42:1.

We thank the reviewer for their point. We feel it is important to reiterate why we chose to focus on ceramide 40:1 and 42:1 as we progress through our investigation and previously did not report the ceramide lipid species profile in each experiment. Our aim was to investigate cell non-autonomous signalling in the communication of lipotoxicity-induced ER stress in skeletal muscle. In the data reported in **Fig 1** we identified the presence of a bioactive lipid released into the media of palmitate treated myocytes. In **Fig 2** we used our lipidomic screen to examine the conditioned, serum-free, media from palmitate treated myocytes in order to narrow down the potential secreted UPR-inducing bioactive lipid species. The lipid content

and variety of species in myocyte conditioned serum-free media would be expected to be far less complex than the lipid contents of the cell or skeletal muscle tissue, since it will only contain lipid species secreted from the myocytes. We performed multivariate analysis of this lipidomic screen in **Supplementary Fig 2** of the manuscript (**line 148, page 7, paragraph 3**):

“To identify the cell non-autonomous UPR-inducing lipid species we performed lipidomic profiling of serum and palmitate-free conditioned media from palmitate-treated myotubes. Multivariate analysis of the media lipidomic data identified lipid species discriminating 100 μ M and 200 μ M treated myocyte media from control (Supplementary Fig. 2).”

To further focus on identifying the UPR-activating bioactive lipid signal secreted in response to palmitate we took the logical step of then focussing on the lipid species that were significantly increased in the conditioned media of both murine muscle cultures and human primary myocytes in response to palmitate treatment. This narrowed our potential signals to 11 lipid species including 5 diacylglycerols, 3 lysophosphocholines and 3 ceramides (ceramide 34:1, 40:1 and 42:1) (Fig 2a). Of these lipid classes only ceramides demonstrated activity in inducing UPR gene expression in myocytes and only the longer chain species ceramide 40:1 and 42:1 demonstrated activity to induce multiple markers of the UPR (Fig 2e,f,g,h & i) (ceramide 34:1 shows activity in inducing only Atf4 expression)

We believe this rationale provides good reasoning in choosing to focus on ceramide 40:1 and 42:1 in the skeletal muscle from murine and human disease models in **Fig 3** and the muscle of CerS 2 H/A mouse models in **Fig 6** later in our studies. These lipids are identified in our screens as being secreted from myocytes in response to palmitate treatment and to show bioactivity in increasing UPR gene expression, and our aim was to identify whether these long-chain ceramide lipid signals were increased in muscle in response to metabolic disease, rather than study the total ceramide lipidome in response to metabolic disease.

However, in response to the reviewers concerns, we understand that there may be a more general interest in what happens to the ceramide species throughout our manuscript so we have now included the ceramide and dihydroceramide species profile identified in the conditioned media of palmitate treated C2C12 myocytes (**page 8, paragraph 1, line 151 Supplementary Fig 3a**):

“Palmitate increased media concentrations of diacylglycerides (DG), lysophosphatidylcholines (LPC) and ceramides (Fig. 2a; Supplementary Fig 3a).”

Likewise, we now include the ceramide and dihydroceramide species profile identified in the conditioned media of palmitate treated human primary skeletal muscle cells (**page 8, paragraph 1, line 166, Supplementary Fig 3c**):

“Ceramides 40:1 and 42:1 (and ceramide 34:1) were also increased in concentration in media collected from HSkMCs treated with palmitate (Fig. 2h; Supplementary Figure 3c).”

As observed in new **Supplementary Figures 3a & c**, the ceramide content and variety of species in both C2C12 and human primary myocyte conditioned serum-free media is less complex than the ceramide contents of a cell or skeletal muscle tissue, since it will only contain ceramide species secreted from the myocytes. However, the detected ceramide species, secreted in response to palmitate, are consistent between the C2C12 and human primary myocyte media with ceramide 34:1, 40:1, 42:1 and 42:2 detected in media from both models. Consistently in both C2C12 and human myocyte media palmitate treatment exclusively increased the concentrations of ceramide 34:1, 40:1 and 42:1. The dihydroceramides 36:0, 38:0 and 40:0 are also consistently detected between the media from the C2C12 and human myocytes, but media dihydroceramide species do not increase in response to palmitate treatment and therefore do not constitute the bioactive UPR-inducing lipid signal.

In further response to the reviewer’s suggestion we now include the ceramide profiles for:

-the muscle and plasma of diet-induced obese mice (**line 190, page 9, New Supplementary Fig 5a & b**):

“The long-chain ceramide signals were increased in the skeletal muscle (Fig. 3a, Supplementary Figure 5a) and blood plasma (Fig. 3b, Supplementary Figure 5b) of western diet-fed mice.”

-the muscle and plasma of humans with diabetes (**page 9, line 198, New Supplementary Fig 5e & f**):

“Lipidomic analysis of the biopsies demonstrated greater concentrations of ceramides 40:1 and 42:1 in the muscle from patients with T2D (Fig. 3c, Supplementary Figure 5e).

Lipidomic analysis of the blood plasma from a random subset of patients identified increased concentrations of ceramides 40:1 and 42:1 in people with T2D (Fig. 3d, Supplementary Figure 5f).

-the intracellular ceramide profile for C2C12 myocytes and human primary skeletal myocytes treated with palmitate (**page 10, line 213, Supplementary Fig 6 a & b**):

“Palmitate increased intra-myocyte concentrations of ceramide 40:1 and 42:1 consistent with increased secretion of these species (Fig. 4b, Supplementary Fig 6a). Similar increases in intra-myocyte ceramide 40:1 and 42:1 concentration following palmitate treatment were observed in primary HSkMCs (Fig. 4c, Supplementary Fig 6b).”

-the muscle and plasma of heterozygous CerS2 H/A mice, and now also we include the full ceramide profiles of homozygous CerS2 H/A mice as a new mouse model added to this iteration of the manuscript (**page 12, line 266, Supplementary Fig 7a & 7b**):

“Lipidomics was used to analyse the concentration of the ceramide species in the skeletal muscle (Fig. 6a, Supplementary Figure 7a) and plasma (Fig. 65b, Supplementary Fig 7b) of mice heterozygous for CerS2 H/A (Het CerS2 H/A Het) and mice homozygous for CerS2 H/A (Homo CerS2 H/A) compared to wild type littermate controls.”

The other concern that these are not being presented is that the CerS2 H/A animals are known to have compensatory increases in C34:1 ceramide when C40:1 and C42:1 are decreased. This cannot be ascertained in experiments as the data is not presented here.

As the reviewer states, there is a compensatory increase in ceramide 34:1 concentration in homozygous CerS2 H/A mouse liver¹ (although this has not previously been observed in skeletal muscle). However, this is not the mouse model originally used in our studies. To limit compensatory effects and generate a more physiological model we used the heterozygous CerS2 H/A mouse. A compensatory increase in ceramide 34:1 has not previously been observed in the Het CerS2 H/A mice. In response to the reviewer’s comment we recognise, therefore, the importance of including the ceramide profile data for the skeletal muscle and plasma of the Het CerS2 H/A mice which we now include alongside new data for the homozygous CerS2 H/A mouse (Homo CerS2 H/A) (**page 12, line 266, Supplementary Fig 7a & 7b**):

“Lipidomics was used to analyse the concentration of the ceramide species in the skeletal muscle (Fig. 6a, Supplementary Figure 7a) and plasma (Fig. 65b, Supplementary Fig 7b) of

mice heterozygous for CerS2 H/A (Het CerS2 H/A Het) and mice homozygous for CerS2 H/A (Homo CerS2 H/A) compared to wild type littermate controls.”

We have also further edited our manuscript to clarify our use of the Het CerS2 H/A mouse (**page 12, line 264**):

“We next explored whether transgenic catalytic inactivation of CerS2 in the CerS2 H/A mouse (two consecutive histidine for alanine substitutions in the CerS2 catalytic center)²⁹ affected synthesis of the ceramides. Lipidomics was used to analyse the concentration of the ceramide species in the skeletal muscle (Fig. 6a, Supplementary Figure 7a) and plasma (Fig. 65b, Supplementary Fig 7b) of mice heterozygous for CerS2 H/A (Het CerS2 H/A Het) and mice homozygous for CerS2 H/A (Homo CerS2 H/A) compared to wild type littermate controls. Tissues of the Homo CerS2 H/A mouse exhibit a compensatory increase in ceramide 34:1 concentration (Supplementary Figure 7a & b)²⁹. To limit compensatory effects and generate a more physiological model we also used the Het CerS2 H/A mouse which is not characterised by compensatory increases in shorter chain ceramides (Supplementary Figure 7a & b). The concentrations of ceramide 40:1 and 42:1 were depleted in plasma and muscle of Het CerS2 H/A, and Homo CerS2 H/A mice in a gene-dosage responsive manner.”

Our data show that ceramide 34:1 is not increased in the blood or skeletal muscle of heterozygous CerS2 H/A mice, but is elevated in a compensatory manner in the new data provided for the homozygous CerS2 H/A mouse. This is consistent with our data which shows that ceramide 34:1 has a limited effect on UPR gene expression. Ceramide 34:1 shows activity in inducing only Atf4 expression (**Page 8, line 165, Supplementary Fig 3b**).

“However, treatment of myocytes with ceramide 34:1 had a limited effect on UPR gene expression, suggesting specificity within lipid class (Supplementary Figure 3b).”

Our analysis shows that both ceramide 40:1 and 42:1 concentration increase UPR gene expression in cells, and are decreased alongside UPR gene expression in the muscle of Het CerS2 H/A mice. A decrease in UPR expression, in the presence of unchanged ceramide 34:1 in the Het CerS2 muscle would further suggest, as we observe, that ceramide 34:1 has little effect on the regulation of the UPR in our models.

In response to the reviewer’s comment and our new data showing that the Het CerS2 H/A mice exhibit decreases in the concentrations of the long-chain ceramides without the

compensatory increases in ceramide 34:1 we have also edited our discussion (**Page 19, line 415**):

“Through the use of heterozygous transgenic catalytic inactivation of CerS2, a model which represents a more physiological condition than total loss of function, preventing compensatory increases in shorter chain ceramides, we are able to directly manipulate the concentration of the long-chain ceramide signals in vivo and show that this modulates the expression of UPR genes in muscle.”

In addition, as alluded to above, to further illustrate our point in response to the reviewer’s comment, we now include data from the homozygous CerS2 H/A knockout mouse.

Lipidomic analysis of this mouse model, as is expected, shows a greater decrease in ceramide 40:1 and 42:1 plasma and muscle concentration when compared to both wild-type control and the heterozygous CerS2 H/A model. As the reviewer points out, unlike the heterozygous CerS2 H/A model, the homozygous CerS2 H/A mouse does have a compensatory increase in ceramide 34:1 concentration in plasma and muscle. Regardless, the heterozygous CerS2 H/A mouse (without compensatory increases in ceramide 34:1) and the homozygous CerS2 H/A mouse demonstrate a gene-dosage responsive decrease in UPR gene expression in response to decreased CerS2 catalytic activity and decreasing long-chain ceramide 40:1 and 42:1 concentrations, supporting our hypothesis that the long chain ceramides drive UPR gene expression.

2. The authors do not acknowledge that C36:1 are the predominate ceramide species in skeletal muscle. This has been documented by numerous animal and human studies. This is not acknowledged in the manuscript and the measurement of this ceramide species is not shown or discussed.

As the reviewer points out ceramide 36:1, derived from CerS1 activity, are the predominant ceramide species in muscle, and others have observed that ceramide 36:1 concentration is higher in the muscle of obese mice². Here, as we discuss in our study introduction, we set out to investigate cell non-autonomous signalling in the communication of lipotoxicity-induced ER stress in skeletal muscle. In doing so we screened media conditioned on palmitate treated C2C12 and human primary myocytes (**Fig1**). We do not detect ceramide 36:1 in the media of either C2C12 or human primary myocytes, suggesting this species is not active as the cell non-autonomous UPR-inducing signal (**page 8, paragraph 1, line 151 Supplementary Fig 3a**):

“Palmitate increased media concentrations of diacylglycerides (DG), lysophosphatidylcholines (LPC) and ceramides (Fig. 2a; Supplementary Fig 3a).”

and **(page 8, paragraph 1, line 166, Supplementary Fig 3c):**

“Ceramide 40:1 and 42:1 (and ceramide 34:1) were also increased in concentration in media collected from HSkMCs treated with palmitate (Fig. 2h; Supplementary Figure 3c).”

Furthermore, in response to the reviewer’s previous comment, we include data for ceramide 36:1 as part of the ceramide profiles for the:

-skeletal muscle and plasma of western diet fed mice (**line 190, page 9, New Supplementary Fig 5a & b**):

“The long-chain ceramide signals were increased in the skeletal muscle (Fig. 3a, Supplementary Figure 5a) and blood plasma (Fig. 3b, Supplementary Figure 5b) of western diet-fed mice.”

-the plasma and muscle of heterozygous and homozygous CerS2 H/A mice (**(page 12, line 266, Supplementary Fig 7a & 7b)**):

“Lipidomics was used to analyse the concentration of the ceramide species in the skeletal muscle (Fig. 6a, Supplementary Figure 7a) and plasma (Fig. 65b, Supplementary Fig 7b) of mice heterozygous for CerS2 H/A (Het CerS2 H/A Het) and mice homozygous for CerS2 H/A (Homo CerS2 H/A) compared to wild type littermate controls.”

-the plasma and muscle of humans with diabetes (**page 9, line 198, New Supplementary Fig 5e & f**):

“Lipidomic analysis of the biopsies demonstrated greater concentrations of ceramides 40:1 and 42:1 in the muscle from patients with T2D (Fig. 3c, Supplementary Figure 5e). Lipidomic analysis of the blood plasma from a random subset of patients identified increased concentrations of ceramides 40:1 and 42:1 in people with T2D (Fig. 3d, Supplementary Figure 5f).

-the intracellular ceramide profile for C2C12 myocytes and human primary skeletal myocytes treated with palmitate (**page 10, line 213, Supplementary Fig 6 a & b**):

“Palmitate increased intra-myocyte concentrations of ceramide 40:1 and 42:1 consistent with increased secretion of these species (Fig. 4b, Supplementary Fig 6a). Similar increases in intra-myocyte ceramide 40:1 and 42:1 concentration following palmitate treatment were observed in primary HSkMCs (Fig. 4c, Supplementary Fig 6b).”

In agreement with the study by Turpin-Nolan et al.² we observe increased ceramide 36:1 concentrations in the muscle of obese mice (**Page 9, line 190, New Supplementary Fig 5a**). We do not observe increased concentrations of this species in the muscle or plasma of humans with diabetes or the plasma of obese mice (**New Supplementary Fig 5 b, e & f**).

Therefore, in light of the reviewer’s suggestion to include discussion of this ceramide species in the manuscript, we have made the following addition to the manuscript’s Discussion (**Page 19, line 410**):

“For instance, despite ceramide 36:1 being the predominate ceramide species in muscle⁴⁹, we do not identify this ceramide species in the media of skeletal myocytes in response to palmitate treatment, suggesting it is not a key signal in the palmitate-induced cell non-autonomous activation of the UPR in muscle.”

3. The ceramides C40:1 and C42:1 are presented to induce key gene markers of the UPR arms of ER stress. Does this then translate into function? No experiments are presented to show more than gene expression, does protein expression also change? Can ER stress be visualised or any functional experiments be performed to confirm gene expression findings?

In response to the reviewer’s comments we now show data confirming that the long-chain ceramides increase the protein expression of UPR markers in myotubes. In addition, using ER-targeted red fluorescent protein we provide confocal microscopy imaging data showing that ceramide 40:1 and ceramide 42:1 increase ER swelling, indicative of ER stress, in human myotubes (**Page 8, line 170, Fig 2j – m**):

“We confirmed induction of the UPR at the protein level using immunoblotting for the C/EBP Homologous Protein (CHOP) transcription factor (downstream of ATF3 and ATF4 in the UPR) and the active splice variant of the UPR transcription factor X-box binding protein 1 (XBP-1s) from protein extracts of HSkMCs treated with ceramides 40:1 and 42:1 and a combination of both ceramide species (Fig 2j & 2k). Using ER-targeted red fluorescent protein and confocal microscopy imaging, we identified ER swelling, indicative of ER stress, in HSkMCs treated with ceramide 40:1 and ceramide 42:1 or a combination of both ceramide species (Fig 2l & 2m)²⁵. ”

4. The conclusion of PERK regulating sphingolipid synthesis is based on one weak experiment. Authors must prove that this connection is regulated by PERK by using PERK KO myotubes and confirming that C40:1 and C42:1 ceramides are altered.

In response to the reviewer’s comment we include new data in our revised manuscript generated from experiments we performed to knockdown PERK expression in primary human skeletal muscle cells using siRNA. We then measured the ceramide 40:1 and ceramide 42:1 concentration in cells and in media conditioned on control and palmitate-treated myocytes with and without PERK knockdown. In support of our conclusions, we found that knockdown of PERK reduced ceramide 40:1 and 42:1 intracellular and extracellular concentrations in myocyte media and inhibited the palmitate-induced increase in the media concentrations of these long chain ceramides (**Page 12, line 253, Figure 5 c - e**):

“To confirm PERK contributed to the palmitate-mediated secretion of ceramide 40:1 and 42:1 from myocytes, we decreased PERK expression in HSkMCs by greater than 85% using siRNA (Fig. 5c). Knockdown of PERK decreased intracellular (Fig 5d) and the extracellular media (Fig 5e) ceramide 40:1 and 42:1 concentrations and inhibited the palmitate-induced increase in the myocyte media concentration of the lipids.”

5. The experiments regarding the CerS2 H/A mice do use a good animal model, they do not adequately answer the questions proposed or conclusions drawn. The CerS2 H/A mice are known to have a compensatory increase in C34:1 ceramides in response to the reduction of C40:1 and C42:1 ceramides. Does this occur in the soleus muscles used in these experiments?

As the reviewer states, and discussed above, there is a compensatory increase in ceramide 34:1 concentration in homozygous CerS2 H/A mouse liver¹ (although this has not previously been observed in skeletal muscle). However, this is not the mouse model originally used in our studies. To limit compensatory effects and generate a more physiological model we originally used the heterozygous CerS2 H/A mouse (Het CerS2 H/A). A compensatory increase in ceramide 34:1 has not previously been observed in the Het CerS2 H/A mouse. In response to the reviewer's comment we recognise, therefore, the importance of including the ceramide profile data for the muscle and plasma of the Het CerS2 H/A mouse. In addition, we have now added new data including the ceramide profiles for the plasma and muscle of the homozygous CerS2 H/A mouse for comparison with the wildtype and Het CerS2 H/A model (**page 12, line 266, supplementary Fig 7a & 7b**):

“Lipidomics was used to analyse the concentration of the ceramide species in the skeletal muscle (Fig. 6a, Supplementary Figure 7a) and plasma (Fig. 6b, Supplementary Fig 7b) of mice heterozygous for CerS2 H/A (Het CerS2 H/A Het) and mice homozygous for CerS2 H/A (Homo CerS2 H/A) compared to wild type littermate controls.”

We have further edited our manuscript to clarify our use of both the Homo CerS2 H/A and the Het CerS2 H/A mouse (**page 12, line 270**):

“Tissues of the Homo CerS2 H/A mouse exhibit a compensatory increase in ceramide 34:1 concentration (Supplementary Figure 7a & b)²⁹. To limit compensatory effects and generate a more physiological model we also used the Het CerS2 H/A mouse which is not characterised by compensatory increases in shorter chain ceramides (Supplementary Figure 7a & b).”

Our data show that ceramide 34:1 is not increased in the blood or skeletal muscle of Het CerS2 H/A mice compared to wild type controls, but is elevated in a compensatory manner in the new data provided for the Homo CerS2 H/A mouse. This is consistent with our data which shows that ceramide 34:1 has a limited effect on UPR gene expression. Ceramide 34:1 shows activity in inducing only *Atf4* expression in cells (**Page 8, line 165, Supplementary Fig 3b**).

“However, treatment of myocytes with ceramide 34:1 had a limited effect on UPR gene expression, suggesting specificity within lipid class (Supplementary Figure 3b).”

Our analysis shows that both ceramide 40:1 and 42:1 concentration increase UPR gene expression in cells (**Figs 2e, f, g & i**), and are decreased alongside UPR gene expression in the muscle of Het CerS2 H/A mice (**page 13, line 274, Fig 6c**):

“The concentrations of ceramide 40:1 and 42:1 were depleted in plasma and muscle of Het CerS2 H/A, and Homo CerS2 H/A mice in a gene-dosage responsive manner. We then assessed whether heterozygous and homozygous catalytic inactivation of CerS2 and subsequent decreases in the long-chain ceramides altered skeletal muscle expression of UPR genes. The expression of both Atf4 and Hspa5 was significantly decreased in the skeletal muscle of Het CerS2 H/A, and Homo CerS2 H/A mice (Fig. 6c). ”

A decrease in UPR expression, in the presence of unchanged ceramide 34:1 in the Het CerS2 muscle (or indeed an increase in ceramide 34:1 concentration in the muscle of the Homo CerS2 H/A mice) would further suggest, as we observe, that ceramide 34:1 has little effect on the regulation of the UPR in our models.

In response to the reviewer’s comment and our new data showing that the Het CerS2 H/A mice exhibit decreases in the concentrations of the long-chain ceramides without the compensatory increases in ceramide 34:1 we have also edited our discussion (**Page 19, line 415**):

“Through the use of heterozygous transgenic catalytic inactivation of CerS2, a model which represents a more physiological condition than total loss of function, preventing compensatory increases in shorter chain ceramides, we are able to directly manipulate the concentration of the long-chain ceramide signals in vivo and show that this modulates the expression of UPR genes in muscle.”

As alluded to above, to further respond to the reviewer’s comment, we have introduced new data from the homozygous CerS2 H/A knockout mouse in this iteration of the manuscript. Lipidomic analysis of this mouse model, as is expected, shows a greater decrease in ceramide 40:1 and 42:1 plasma and muscle concentration when compared to the heterozygous CerS2 H/A model. As the reviewer points out, unlike the heterozygous CerS2 H/A model, the homozygous CerS2 H/A mouse does have a compensatory increase in ceramide 34:1 concentration in plasma and muscle. Regardless, the heterozygous CerS2 H/A mouse (without compensatory increases in ceramide 34:1) and the homozygous CerS2 H/A mouse demonstrate a gene-dosage responsive decrease in UPR gene expression in response to decreased CerS2 catalytic activity and decreasing long-chain ceramide 40:1 and 42:1

concentrations, supporting our hypothesis that the long chain ceramides drive UPR gene expression.

Also, the plasma measures of C40:1 and C42:1 ceramides are not adequate to conclude that the plasma and soleus muscle ceramide concentrations were related. The plasma concentrations of ceramides would be predominately derived from the liver, not the muscle.

In response to the reviewer's comment, we have provided new data strengthening the association between muscle ceramide 40:1 and 42:1 concentration and the plasma concentration of these long chain ceramides. We now show that the muscle and plasma concentrations of ceramide 40:1 and 42:1 are significantly positively correlated in mouse (**Page 9, line 192, Supplementary Fig 5c & 5d**):

“Muscle and plasma concentrations of ceramide 40:1 and 42:1 were significantly positively correlated in the mice (Supplementary Fig 5c & 5d).”

and in humans (**Page 10, line 202, Supplementary Fig 5g & 5h**):

“Muscle and plasma concentrations of ceramide 40:1 and 42:1 were significantly positively correlated in the patients (Supplementary Fig 5g & 5h).”

We agree with the reviewer and do not attempt to make this conclusion based on these data. Instead we conclude that (**Page 10, line 203**):

“These data highlight that increases in skeletal muscle and blood plasma long-chain ceramides are associated with metabolic disease in mice and humans.”

In light of the reviewer's comment we have further clarified this with the addition of the following statement in our discussion section (**Page 17, line 368**):

“Here, we observe an increase in the concentrations of long-chain ceramides 40:1 and 42:1 in the blood plasma and muscle of western diet-induced obese mice and humans with T2D. Confirming that ceramides 40:1 and 42:1 are enriched in muscle by metabolic disease in mouse and humans. We do not preclude that circulating concentrations of long-chain ceramides in this setting may be contributed to by additional tissues besides skeletal muscle.”

Therefore, experiments extracting muscle specific exported ceramides need to be performed or exosomes/EVs derived from muscle cultures could be performed in coculturing experiments to confirm that the C40:1 and C42:1 ceramides do indeed activate the muscle UPR.

In response to the reviewer's suggestion we have performed additional experiments to confirm that the long chain ceramides 40:1 and 42:1, derived from CerS2, activate the muscle UPR. We have used siRNA to knockdown the expression of *CERS2* in primary human skeletal myocytes with and without co-treatment with 100 μ M palmitate. Using this approach we confirmed that knockdown of *CERS2* decreased intracellular and media ceramide 40:1 and 42:1 concentrations and inhibited the palmitate-induced increase in the myocyte media concentration of the ceramides (**Page 11, line 233, Fig 4h - 4j**):

“To confirm long-chain ceramides 40:1 and 42:1 are generated through CERS2 in response to palmitate treatment in myocytes, we decreased CERS2 expression in HSkMCs by 70% using siRNA (Fig 4h). Knockdown of CERS2 decreased intracellular (Fig 4i) and extracellular media (Fig 4j) ceramide 40:1 and 42:1 concentrations and inhibited the palmitate-induced increase in the myocyte media concentration of the lipids. These data suggest increased ceramide 40:1 and ceramide 42:1 concentrations in response to palmitate are generated through the de novo pathway.”

We then performed an additional experiment in which we transferred conditioned media collected from HSkMCs treated with 100 μ M palmitate or BSA control, with and without siRNA-mediated *CERS2* knockdown, to naïve HSkMCs. We found that the induction of UPR gene expression induced by media conditioned on palmitate-treated HSkMCs was inhibited by *CERS2* knockdown confirming that the *CERS2* generated long chain ceramides are non-cell autonomous UPR-inducing lipid signals (**Page 11, line 240, Fig 4k**):

“We then transferred conditioned serum- and palmitate-free media collected from HSkMCs treated with 100 μ M palmitate or BSA control, with and without siRNA-mediated CERS2 knockdown, to naïve HSkMCs. The induction of UPR gene expression induced by media conditioned on palmitate treated HSkMCs was inhibited by CERS2 knockdown (Fig 4k) in agreement with the effect of CERS2 knockdown on ceramide 40:1 and ceramide 42:1 media concentrations. These data confirm that the CERS2 generated long-chain ceramides are cell non-autonomous UPR-inducing lipid signals.”

6. The conclusions drawn from Figure 7 do not fit the data presented. The gene expression pattern of intracellular ceramides produced by palmitate does not fit with the UPR gene expression. Authors conclude that dihydroceramides drive UPR however, the UPR genes increase only with palmitate or palmitate + fenretinide. If ceramides were solely responsible for UPR activation, then why is there no increase in UPR gene expression in fenretinide only controls? Fenretinide would inhibit all dihydroceramides and therefore should alter some of the UPR gene expression as the DES1 inhibitor increases the intracellular C40:1 and C42:1 ceramides around 8 fold in Figure 7f.

We thank the reviewer for pointing out this seeming discrepancy. However, we can confirm that the mean expression of *Atf3*, *Atf4*, *Hspa5*, and *Edem1* are greater in the fenretinide treatment group compared to control, consistent with our conclusions. When we plot only the control vs fenretinide data and analyse with a two-tailed Student's T-test, which we have presented below for the purposes of peer-review, we observe that *Atf3* and *Hspa5* expression is significantly increased in the fenretinide treatment group. Due to the nature of one-way ANOVA with multiple comparisons post hoc correction and the substantial (synergistic) differential increase in expression of the UPR markers when treated with both fenretinide and palmitate in combination, compared to the comparatively modest increases with fenretinide or palmitate alone, these increases are not identified as significant when all data are presented together with one-way ANOVA with multiple comparison testing post hoc correction.

*Activating transcription factor 3 (Atf3), activating transcription factor 4 (Atf4), Heat Shock Protein Family A (Hsp70) Member 5 (Hspa5) and ER Degradation Enhancing Alpha-Mannosidase Like Protein 1 (Edem1) UPR gene expression in control and 5µM fenretinide treated C2C12 myotubes. Statistical analysis was performed using a two-tailed Student's T-test (n = 3). * P ≤ 0.05, ** P < 0.01, *** P < 0.001, **** P < 0.0001. Data are expressed as mean ± SEM with individual data points shown*

Nevertheless, we acknowledge the reviewer's important point that more investigation of the involvement of dihydroceramides in palmitate-ceramide-induced UPR gene expression is required. Therefore, we have strengthened the data previously generated with pharmacological inhibition of Des1 (fenretinide) with the addition of data generated with genetic manipulation of *DEGS1* (the gene encoding Des1) using siRNA. We now present data in which we knockdown expression of *DEGS1* in human primary skeletal muscle cells by greater than 50% (**New Fig 8e**). Reflecting the pharmacological inhibitor experiment, knockdown of *DEGS1* in combination with palmitate treatment substantially increased the intracellular concentration of dihydroceramide 40:0 and 42:0 (**New Fig 8g**). In further agreement with the pharmacological inhibitor study, knockdown of *DEGS1* with palmitate treatment synergistically increases expression of UPR genes in the human primary myotubes (**New Fig 8h**) (**Page 15, line 338**):

“To confirm a role for dihydroceramides in lipotoxicity-induced UPR activation we used siRNA to decrease DEGS1 expression (the gene encoding Des1) by > 50% in the translational primary HSkMC model (Fig 8e). DEGS1 knockdown in HSkMCs decreased the intracellular concentration of ceramides 40:1 and 42:1 (Fig 8f). Concomitant palmitate treatment of human primary myotubes with DEGS1 knockdown increased the intracellular concentrations of dihydroceramides 40:0 and 42:0 (Fig 8g). Palmitate treatment in combination with DEGS1 knockdown synergistically enhanced UPR expression in human myotubes (Fig 8h).”

Moreover, we agree with the reviewer that dihydroceramides may not represent an exclusive mechanism for palmitate-ceramide induced cell non-autonomous UPR activation and we do not exclude the possibility of other signalling mechanisms having a role. Therefore, we conclude that this work implicates dihydroceramides in the cell non-autonomous ceramide-induced activation of the UPR rather than suggesting this is the sole mechanism (**page 16, line 345**):

“These data implicate dihydroceramides in the cell non-autonomous ceramide-induced activation of UPR signalling in lipotoxicity.”

Minor concerns:

7. Methods are missing for the addition of ceramide derivatives to the cell culture media. It is not clear how the lipophobic ceramides were added to cells, or if they even incorporated into the cells.

We thank the reviewer for pointing out our oversight. We now include the ceramide preparation methods used for treatment of the myocytes. We used the well-established dodecane/ethanol system which efficiently delivers exogenous phospholipids to cells for the examination of biological effects^{3,4}. We have now included these details in the methods section (**Page 22, line 472**):

“Preparation of ceramides

Preparation of ceramides followed standard protocols in the literature^{47, 48}. Briefly, ethanol and dodecane were mixed at a ratio of 98:2, followed by vortexing and prewarming to 37°C. Meanwhile, ceramide 40:1 and ceramide 42:1 (Avanti Polar Lipids Inc) were dissolved in chloroform-methanol 1:1. The required volume was then dried down under N₂ gas. The pre-warmed ethanol-dodecane mixture was added to the dried ceramide 40:1 and ceramide 42:1 such that the final stock concentration was 2.5 mM. This mixture was thoroughly vortexed and incubated at 37°C for a further 20 min followed by further vortexing. The stock solution was then diluted to the required concentration using ethanol/dodecane. The stock solution of ceramide 40:1 or ceramide 42:1 were diluted in cell culture medium (37°C) to the required treatment concentration followed by vortex mixing. Ethanol/dodecane was included as a vehicle control at the same concentration as the treatment.”

Moreover, supporting effective delivery of ceramides to cells, we demonstrate in both C2C12 murine myocytes and human primary skeletal muscle cells in **Figure 8a** and **Figure 8c** respectively, that treatment with ceramide 40:1 increases intracellular ceramide 40:1 concentration (but not ceramide 42:1) and, likewise, treatment with ceramide 42:1 increases

intracellular ceramide 42:1 concentration (but not ceramide 40:1) (**Page 15, Line 320, Figure 8a & 8c**):

“Myotubes were treated with 10 μM ceramide 40:1, 10 μM ceramide 42:1, a combination of both species, or the vehicle control. Cells were then analysed using lipidomics. Ceramide 40:1 treatment increased intracellular concentrations of ceramide 40:1, and ceramide 42:1 treatment increased intracellular concentrations of ceramide 42:1 (Fig. 8a). This indicates ceramides are imported into the myotubes..... Similar effects on intracellular ceramide 40:1 and 42:1were seen in long-chain ceramide-treated primary HSkMCs (Fig. 8c & d)”

8. The introduction lacks a section on ceramides and specifically, ceramides in skeletal muscle.

We certainly appreciate the reviewer’s point of view regarding possible inclusion of a section on ceramides in the introduction, although, this was not an oversight on our part. As we hope the reviewer appreciates, the original aim of this study was not to investigate a role of ceramides in palmitate-induced activation of the UPR but rather to investigate cell non-autonomous signalling in the communication of lipotoxicity-induced ER stress in skeletal muscle as stated in our introduction (**Page 4, line 66**):

“We hypothesised that cell non-autonomous signalling may be important in the communication of lipotoxicity-induced ER stress in skeletal muscle.”

The discovery that long-chain ceramides represent one potential signal mediating this process is made during our investigations (first implicated in **Fig 2**) presented in the results and was not an *a priori* assumption of this research. For this reason, we felt that discussion of ceramides was more appropriately located in our discussion section (**Page 17, line 366**):

Ceramides have previously been implicated in both the development of metabolic disease and the regulation of ER stress. Blood plasma and tissue ceramide and dihydroceramide concentrations are associated with a range of cardiometabolic diseases^{35,36,37,38,39}. Here, we observe an increase in the concentrations of long-chain ceramides C40:1 and C42:1 in the blood plasma and muscle of western diet-induced obese mice and humans with type 2 diabetes. Confirming that ceramide 40:1 and ceramide 42:1 are enriched in muscle by metabolic disease in mouse and humans. We do not preclude that circulating concentrations of long-chain ceramides in this setting may be contributed to by additional tissues besides

skeletal muscle. Ceramides have been identified as key mechanistic contributors to the lipotoxicity-induced development of adipose tissue and hepatic insulin resistance⁴⁰. Inhibition of ceramide synthesis ameliorates palmitate-induced ER stress in pancreatic β -cells⁴¹. Similarly, palmitate induced the production of long chain ceramides, and siRNA knockdown of CerS 5 and 6 prevented myristate-induced ER stress, in intestinal epithelial cells⁴². A short-chain ceramide analogue, infused into the hypothalamus of rats increases hypothalamic ER stress, and reduces the thermogenic capacity of brown adipose tissue, thus, highlighting that ceramide-induced ER stress in one tissue can have a metabolic impact on distal tissues⁴³.”

Nevertheless, we recognise the importance of the need to strengthen the discussion of skeletal muscle ceramides, highlighted by the reviewer. Therefore, we have included additional discussion of ceramides in skeletal muscle in our discussion section (**Page 18, line 381**)

“Evidence is emerging that skeletal muscle ceramide metabolism has a key role in the regulation of systemic physiology. Skeletal muscle ceramides are elevated in both patients with, and animal models of, obesity and diabetes and negatively correlate with insulin sensitivity^{44,45,46}. In addition, muscle-specific disruption of ceramide synthesis by CerS1 improves systemic insulin sensitivity in diet induced obese mice⁴⁷.“

9. The authors use the term “lipotoxic lipid species” however, numerous studies have now shown that most metabolic tissues are remarkably plastic and are able to adapt to increased lipid concentrations of TAGs, or even distinct ceramide species.

In response to the reviewer’s comment we have edited our manuscript removing the term “lipotoxic lipid species”.

10. Why does the palmitic acid concentration change in each experiment? Ie 50um, 100um and 200um concentrations are used throughout different experiments but it is not explained why.

We apologise for our lack of clarity. In our experiments we have endeavoured to replicate physiological conditions. We also aim to be translational in our research and use both murine derived C2C12 myotubes and also primary human skeletal muscle cells. The physiological

concentration of palmitate in circulation differs between mouse and human. The concentration of palmitate in mouse plasma ranges from 100 – 300 μM ^{5,6}. Therefore, in experiments using murine C2C12 cells we have opted to remain within this physiological range and use palmitate concentrations of 100 μM and 200 μM . However, the plasma concentration of palmitate in humans is lower and ranges from 50 – 150 μM ^{7,8,9}. Hence, in experiments using primary human skeletal muscle cells we use palmitate concentrations of 50 μM and 100 μM reflective of this physiological range.

We have edited the manuscript to reflect this physiological difference between palmitate concentrations in experiments with murine-derived cells (**Page 5, line 88**):

“Concentrations of 100 μM and 200 μM palmitate reflect the physiological palmitate plasma concentration range in mice (100 – 300 μM)^{19,20} ...”

And human cells (**Page 6, line 105**):

“Concentrations of 50 μM and 100 μM palmitate reflect the physiological palmitate plasma concentration range in humans (50 – 150 μM)^{17,18,21} ...”

11. The controls for lipid extraction experiments regarding the culture media are lacking. Control lipid extracts and control organic extracts of media in Figure 1e should be performed for comparisons, not just comparisons of control organic to palmitate media. Therefore, please include the aqueous extract gene expression to confirm it is palmitate media lipid extracts that are increasing gene expression.

We apologise for our lack of precision regarding the description of this experiment which may have led to a misunderstanding regarding the data presented. In **Fig 1e** we directly compare the effect of the lipid containing organic component extracted from conditioned media from palmitate treated myocytes and the lipid containing organic component extracted from conditioned media from control myocytes. These were reconstituted in media and applied to naïve myocytes to investigate their effects on UPR gene expression.

In addition, we are grateful to the reviewer for their suggestion to investigate the effects of the metabolite-containing aqueous fraction from palmitate-treated myocytes. Following the reviewer’s suggestion we have performed an additional experiment comparing the effect of the metabolite containing aqueous component extracted from conditioned media from palmitate treated myocytes and the metabolite containing aqueous component extracted from conditioned media from control myocytes. These were reconstituted in media and applied to

naïve myocytes to investigate their effects on UPR gene expression. We found that the metabolite containing aqueous fraction from the conditioned media of palmitate treated myocytes also induced expression of UPR genes in naïve myocytes (**New Supplementary Fig 1e**). Therefore, in this study, we now also provide data suggesting the presence of an aqueous soluble cell non-autonomous metabolite signal capable of inducing the UPR.

We have clarified the experimental design of the data presented in **Figure 1e**, comparing the organic extracts of control and palmitate-treated myocyte media. We have further edited the legend of Fig 1e for clarity, and incorporated the new data examining the effect of the metabolite-containing aqueous fraction from conditioned media of palmitate treated myocytes on UPR gene expression in naïve myocytes into our manuscript (**page 7, line 130, Supplementary Figure 1e**):

“We hypothesised that the UPR-inducing signal may be a bioactive lipid. The lipid containing organic component and metabolite containing aqueous component of the conditioned media from both control and palmitate treated myocytes was extracted using protein-denaturing organic-aqueous solvent partition. The lipids, isolated from the organic fractions of conditioned media from both control and palmitate treated myocytes, were reconstituted in serum-free media and transferred on to naïve differentiated myotubes. The organic lipid fraction of media from palmitate-treated myotubes increased expression of UPR target genes compared to the organic lipid fraction of media from control myotubes (Fig. 1e). Next, we examined whether aqueous soluble metabolites may represent additional UPR-inducing signals. The metabolites isolated from the aqueous fractions of conditioned media from both control and palmitate treated myocytes were reconstituted in serum-free media and transferred on to naïve differentiated myotubes. The aqueous metabolite fraction of media from palmitate-treated myotubes also increased expression of UPR target genes compared to the aqueous fraction of media from control myotubes (Supplementary Fig. 1e). These data suggest that both a bioactive lipid(s) and metabolite(s) may function as cell non-autonomous UPR-inducing secreted signals.”

Investigations of the nature, signalling and physiological function of the putative aqueous soluble UPR-inducing signals will be important but do not form the focus of this study and are outside the scope of the current investigation. We consider it a priority to study the putative aqueous soluble metabolite signals in future investigations. In light of our new data

suggesting the presence of an aqueous-soluble UPR-inducing metabolite signal we have edited the discussion of our manuscript (**Page 17, Line 360**):

“However, our work does not preclude the presence of other metabolite factors, protein signals, or bioactive lipids that may contribute to cell non-autonomous UPR activation. Indeed, our data suggests the presence of an, as yet unidentified, aqueous soluble metabolite factor(s) secreted from palmitate-treated myocytes which also activate UPR gene expression.”

12. The data presented in Figure 3C does not appear to be different as stated in the text.

We are grateful for the reviewer’s comments regarding the statistical significance of the differences in muscle long-chain ceramides between control volunteers and patients with type 2 diabetes. This was an error of omission for which we apologise and have now corrected **Fig 3c** to include the significance indicators. In addition, we have edited the way in which we display the data and now use box and whisker diagrams to make the differences in these groups clearer.

13. Line 175 – Needs to be corrected to “multiple Ceramide synthases”.

This has been edited in the text to read (**page 10, line 220**):

“multiple ceramide synthases”

14. It would be informative for the authors to include the expression levels of the Ceramide synthases and other enzymes of the de novo ceramide synthesis pathway to demonstrate that the “increased generation of ceramides through the de novo synthesis pathway results in lipotoxicity”.

We acknowledge our lack of specificity in presenting these data. In response to the reviewer’s comment we wanted to clarify that our statement read (**line 238, page 11**): *“These data suggest increased ceramide concentrations in response to lipotoxicity are generated through the de novo pathway.”* And therefore our intention was not to assert that the

increased generation of ceramides results in lipotoxicity in relation to the data being introduced at this point in the manuscript. Our statement was made in response to our data presented in **Figures 4d** and **4e** that indicate that inhibition of two points in the ceramide *de novo* synthesis pathway (serine palmitoyl transferase, **figure 4d**; ceramide synthases, **figure 4e**) abrogates the palmitate-induced increase in ceramide 40:1 and ceramide 42:1 production in myocytes. For clarity we have now edited this to read (**page 11, line 238**):

“These data suggest increased ceramide 40:1 and ceramide 42:1 concentrations in response to palmitate are generated through the de novo pathway.”

In the *de novo* pathway long-chain ceramides are predominantly synthesised by CerS2¹⁰. Therefore, we chose to provide data showing that in both C2C12 murine myocytes and human primary myocytes, palmitate treatment increases expression of CerS2 (**Page 11, line 232, Figure 4f & g**):

palmitate-treatment of C2C12 myotubes and HSkMCs induced *CerS2* expression (Fig 4f & g).”

In response to the reviewer’s suggestion we have performed an additional experiment. We have acquired new data in support of the specificity of the involvement of CerS2 in the *de novo* pathway-mediated enhanced production of the long-chain ceramides 40:1 and 42:1 in myocytes following palmitate treatment. The synthase isoform CerS4 overlaps with CerS2 in its specificity for the production of long-chain ceramides¹¹ (Ceramide 40:1 with particular relevance to the current study). However, CerS4 production of long-chain ceramides is unaffected by fumonisin B1¹¹. As our data in **Figure 4d** demonstrates that palmitate-induced production of long-chain ceramides is sensitive to fumonisin B1 inhibition, this suggests the mechanism is independent of CerS4. Therefore, in addressing the reviewer’s comment we include new data using qPCR to examine the expression of CerS4 in response to palmitate treatment in C2C12 myocytes. The expression of CerS4 was found to decrease in response to higher concentrations of palmitate (**page 11, line 228, New Supplementary Fig 6c**):

“However, CerS4 overlaps with CerS2 in its specificity for the production of long-chain ceramides²⁷. CerS4 production of long-chain ceramides is unaffected by FBI²⁷, this suggests that the mechanism for palmitate-induced ceramide 40:1 and 42:1 production is independent of CerS4. The expression of CerS4 was decreased in response to palmitate treatment in C2C12 myocytes (Supplementary Fig 6c).”

To further support our conclusion that increased ceramide concentrations in response to palmitate are generated through the *de novo* pathway and specifically through CerS2 we have performed an additional experiment to knockdown *CERS2* in human primary myocytes using siRNA. We show that genetic depletion of *CERS2* result in inhibition of the increase in long chain ceramide concentrations induced by palmitate in myocytes and myocyte media. This confirms the importance of de novo ceramide production via *CERS2* in the palmitate-mediated secretion of ceramides from muscle cells (**Page 11, line 233, Figure 4h-j**):

“To confirm long-chain ceramides 40:1 and 42:1 are generated through CERS2 in response to palmitate treatment in myocytes, we decreased CERS2 expression in HSkMCs by 70% using siRNA (Fig 4h). Knockdown of CERS2 decreased intracellular (Fig 4i) and extracellular media (Fig 4j) ceramide 40:1 and 42:1 concentrations and inhibited the palmitate-induced increase in the myocyte media concentration of the lipids.”

We then performed another experiment in which we transferred conditioned media collected from HSkMCs treated with 100 μ M palmitate or BSA control, with and without siRNA-mediated *CERS2* knockdown (depleted media long-chain ceramides), to naïve HSkMCs. We found that the induction of UPR gene expression induced by media conditioned on palmitate treated HSkMCs was inhibited by *CERS2* knockdown confirming that the *CERS2* generated long-chain ceramides are cell non-autonomous UPR-inducing lipid signals (**Page 11, line 240, Fig 4k**):

“We then transferred conditioned serum- and palmitate-free media collected from HSkMCs treated with 100 μ M palmitate or BSA control, with and without siRNA-mediated CERS2 knockdown, to naïve HSkMCs. The induction of UPR gene expression induced by media conditioned on palmitate treated HSkMCs was inhibited by CERS2 knockdown (Fig 4k) in agreement with the effect of CERS2 knockdown on ceramide 40:1 and ceramide 42:1 media concentrations. These data confirm that the CERS2 generated long-chain ceramides are cell non-autonomous UPR-inducing lipid signals.”

15. Line 217-8. The statement that the bioactive lipid profiles of EVs has not previously been reported is false. Please refer to the published article.

Burrello J, Biemmi V, Dei Cas M, Amongero M, Bolis S, Lazzarini E, et al. Sphingolipid

composition of circulating extracellular vesicles after myocardial ischemia. Scientific reports. 2020;10(1):16182.

We thank the reviewer for highlighting our oversight. In light of the reviewer's comments we now include citations to two recent review articles on the topic^{12, 13}, as well as the reference the reviewer suggests, and have edited this line to read (**Page 13, line 291**):

“There is developing recognition that bioactive lipid signals may be partitioned into EVs^{30,31,32}”

16. Line 226. The authors state that the microvesicle release was unaffected by CERT inhibition but Figure 6c shows that the CERT inhibitor HPA-12 increased microvesicle particles in controls.

We thank the reviewer for their comment. We did not think that the CERT inhibitor (HPA-12) data contributed to the conclusions drawn in our manuscript and we have therefore removed these data from **Figures 7b & 7c**.

17. Ceramide concentrations were not shown in the depleted media experiments, therefore how are we to know that ceramide levels were not altered in these experiments and therefore would account for the reduction in UPR signalling?

We thank the reviewer for pointing out our omission. As we find that the long-chain ceramides are enriched in EVs that induce UPR activation we would expect that depletion of these EVs from media would result in a decrease in the concentration of the ceramides in the media. We have now performed an additional experiment using LC-MS to measure the concentrations of the long-chain ceramides 40:1 and 42:1 in the EV-depleted media from palmitate treated myocytes. In support of our data showing that depletion of EVs from palmitate-treated myocyte media abrogates the capacity of the media to induce the UPR in myocytes (**Figure 7f**), we find that depletion of the EVs decreases the concentration of ceramide 40:1 and ceramide 42:1 in the media (**Page 14, line 313, Figure 7i**):

“Reciprocally, LC-MS analysis of EV-depleted media demonstrated a ~92% reduction in long chain ceramides 40:1 and 42:1 (Fig 7i).”

Reviewer #2 (Remarks to the Author):

In the current manuscript, the authors reported that the long-chain ceramide was extracellularly transported via EVs in ER stress. They also showed that the ceramide, which was transported by EVs, induced UPR activation in naïve myotubes. Overall, the experiment was well performed, and the manuscript was well organized. However, there are several technical issues that need to be addressed below.

We thank the reviewer for their assessment that our experiments were well performed and clearly communicated and for their insightful comments. We feel that by addressing these technical points the manuscript has been substantially strengthened.

1. MISEV 2018 guidelines currently encourage to use of operational terms for EV subtypes that refer to physical characteristics of EVs, such as size (“small EVs” and “medium/large EVs,” with ranges defined, respectively. The reviewer recommends changing the expression of “exosome” and “microvesicles” to “small EVs” and “large EVs”, respectively.

We thank the reviewer for their recommendation. In line with the reviewer’s comment we have edited the manuscript and figures throughout so that all previous references to “exosomes” now state “small EVs” and previous references to “microvesicles” now state “large EVs”.

2. The manuscript needs more information of the EV characterization. Please refer to the following, and add more information to the manuscript.

2-1. In Fig.6b and c, although the authors analyzed particle number of EVs, quantification of EVs should be addressed by at least 2 methods: protein amount, particle number, lipid amount. Please analyze the protein or lipid amount of EVs-derived from control and palmitate treated myocyte.

In response to the reviewer’s comment we have analysed the total protein concentration of EVs derived from the media of BSA-treated control and palmitate-treated myocytes. In agreement with our particle number analysis the protein concentration of EV isolates from

palmitate-treated myocyte media was significantly greater than controls (**Page 14, line 296, Supplementary Figure 9a**).

“To confirm increased EV release from myocytes in response to palmitate in accordance with the minimal information for studies of extracellular vesicles³³, the total protein concentration of EV isolates from media conditioned on BSA treated control and 200 μ M palmitate treated myocytes was measured (Supplementary Fig 9a).”

2-2. At least one protein of categories 1-3 (1. Transmembrane or GPI-anchored proteins, 2. Cytosolic proteins recovered in EVs, 3. Major components of the non-EV co-isolated structure) must be analyzed to demonstrate the EV nature and the degree of purity of an EV preparation. (Please refer to Table 3 of MISEV2018.)

In response to the reviewer’s comment we have performed immunoblotting for the plasma membrane associated transmembrane protein, transferrin receptor 2 (Tfr2), the cytosolic protein superoxide dismutase 1 (Sod) recovered in EVs, and the mitochondrial protein carnitine palmitoyltransferase 1 (Cpt1), a component of the non-EV co-isolated structure in EV isolations using the protein isolates from media extracellular vesicles, conditioned media and whole cell lysates of C2C12 myotubes (**Page 14, line 300, Supplementary Fig 9b**):

“To confirm the nature and degree of purity of the EV preparation we used immunoblotting to determine the presence of the plasma membrane associated transmembrane protein, transferrin receptor 2 (Tfr2), the cytosolic protein superoxide dismutase 1 (Sod) recovered in EVs, and the mitochondrial protein carnitine palmitoyltransferase 1 (Cpt1), a component of the non-EV co-isolated structure in EV isolations, conditioned media and whole cell myocyte protein isolations (Supplementary Fig 9b).”

3. In Fig.6 d-f, the authors should investigate whether the mRNA levels of UPR genes are upregulated in a dose-dependent manner by treating EVs-derived from palmitate treated myocyte.

We thank the reviewer for the suggestion and realise the importance of increasing the evidence for the association between long-chain ceramide containing EVs and activation of the UPR. We felt that the heterogeneous nature of an EV population (and EV contents) makes

interpreting dose response ambiguous. Therefore, we opted to use the EV enrichment and depletion approach presented to assess EV-induced UPR activation. We demonstrate that enrichment of both small EVs and large EVs from palmitate-treated myotubes activates UPR gene expression in naïve myotubes (**Page 14, line 308, Figure 7d & 7e**):

“Small EVs (Fig. 7d) and large EVs (Fig. 7e) isolated from palmitate-treated myotubes increased UPR gene expression in naïve myotubes following their reconstitution in serum-free media.”

We also demonstrate that once the EVs are depleted from conditioned media from palmitate-treated myocytes the ability of the media to induce UPR gene expression in naïve myotubes is lost (**Page 14, line 310, Figure 7f**):

“EV-depleted conditioned media, however, did not induce UPR gene expression in naïve myotubes (Fig. 7f).”

We then confirmed the presence of increased concentrations of long-chain ceramides in the small and large EVs isolated from the media of palmitate treated myotubes (**Page 14, line 311, Figure 7g & 7h**):

“Lipidomic analysis of small EVs (Fig. 7g) and large EVs (Fig. 7h) showed that palmitate-treated myotubes produced EVs enriched in ceramides 40:1 and 42:1.”

In response to the reviewer’s comment we have now further strengthened the evidence linking the bioactive long-chain ceramides to EVs and UPR activation by providing new data using LC-MS to show that the depletion of the EVs from myocyte media results in an approximately 92% decrease in the concentration of ceramide 40:1 and 42:1 in this media (**Page 14, line 313, Figure 7i**):

“Reciprocally, LC-MS analysis of EV-depleted media demonstrated a ~92% reduction in long chain ceramides 40:1 and 42:1 (Fig 7i).”

Reviewer #3 (Remarks to the Author):

The manuscript describes a study focusing on the connection of long-chain ceramides, lipotoxicity and reticulum stress in skeletal muscle. Overall, the study is well planned experimentally and it gives novel insights of the role of ceramides. However, there are several areas in the manuscript that require further consideration. The strength of the work is that it combines cell-based studies with mice and human data including also a more mechanistic study.

We thank the reviewer for their assessment of the manuscript which found it to be well planned and to provide novel insights, and for their recognition of our investigations across cell culture, murine models, human data and mechanistic studies. We also thank them for their considered points, which by addressing we feel we have further strengthened the manuscript.

However, in humans, only muscle biopsy data was shown; it would be relevant to show (as is done for mice) the data also on the matching blood-based samples.

We appreciate the reviewer's suggestion. Our original ethics approval did not allow for the use of human plasma in this research. In response to the reviewer's comment we have amended the ethics and now include data from plasma collected from a subset of our control volunteers and patients with type 2 diabetes. We have generated new data examining the concentration of ceramide 40:1 and ceramide 42:1 in the blood plasma of a subset of our patients with type 2 diabetes and control volunteers. As in skeletal muscle (and muscle and plasma of mice), the concentration of ceramide 40:1 and ceramide 42:1 were greater in patients with type 2 diabetes compared with control volunteers (**page 10, line 200, Figure 3d & Supplementary Fig 5f**):

“Lipidomic analysis of the blood plasma from a random subset of patients identified increased concentrations of ceramides 40:1 and 42:1 in people with type 2 diabetes (Fig. 3d, Supplementary Figure 5f).”

The conditions chosen for the experiments (concentrations, exposure times) have not been well justified/explained, making it more challenging to evaluate whether the results present a realistic situation.

We apologise for our lack of clarity. We have edited the manuscript to clarify the conditions used in our experiments with a particular focus on the concentrations and exposure times for the lipid species investigated.

In our experiments we have endeavoured to replicate physiological conditions. We also aim to be translational in our research and use both murine derived C2C12 myotubes and also primary human skeletal muscle cells.

In relation to our use of palmitate, the physiological concentration of palmitate in circulation differs between mouse and human. The concentration of palmitate in mouse plasma ranges from 100 – 300 μM ^{5,6}. Therefore, in experiments using murine C2C12 cells we have opted to remain within this physiological range and use palmitate concentrations of 100 μM and 200 μM . However, the plasma concentration of palmitate in humans is lower and ranges from 50 – 150 μM ^{7,8,9}. Hence, in experiments using primary human skeletal muscle cells we use palmitate concentrations of 50 μM and 100 μM reflective of this physiological range.

Moreover, much of the previous research in this area has investigated the effects of saturated fatty acids at high (often non-physiological) doses in an acute setting (typically no longer than 24 hours of exposure)^{14, 15, 16, 17}. However, physiologically *in vivo*, the myotubes within the skeletal muscle will be exposed to circulating palmitate concentrations chronically. In addition, obesity and diabetes are chronic disorders, with sustained increases in circulating plasma free fatty acid concentrations^{18, 19}. Therefore, our study aims to investigate a more physiologically relevant approach using *in vitro* models of chronic palmitate exposure (6 days) at physiological concentrations to better understand the effects of prolonged lipotoxicity on cell non-autonomous UPR activation.

In response to the reviewer's comment, therefore, we have edited the manuscript to reflect this physiological difference between palmitate concentrations in experiments with murine and human cells and to better communicate the chronic nature of the *in vitro* palmitate exposure (**Page 5, line 88**):

“Concentrations of 100 μ M and 200 μ M palmitate reflect the physiological palmitate plasma concentration range in mice (100 – 300 μ M^{19,20}, and chronic treatment (6 days) models continual palmitate exposure in vivo.”

and **(Page 6, line 105)**:

“Concentrations of 50 μ M and 100 μ M palmitate reflect the physiological palmitate plasma concentration range in humans (50 – 150 μ M)^{17,18,21}, and chronic treatment (6 days) models continual palmitate exposure in vivo.”

To further clarify the differences when equivalent experiments were performed in both murine-derived C2C12 and human skeletal myocytes, we have labelled figure panels to indicate this.

In relation to the concentrations of the lipid species Diacylglycerol 32:0 (0 – 10 μ M), Lysophosphocholine 16:0 (0 – 100 μ M) and Lysophosphocholine 18:0 (0 – 50 μ M) investigated, these were physiological and selected based on dose ranges in blood plasma reported in the literature cited^{7, 20}.

Likewise the concentrations of the ceramide species investigated, ceramide 40:1 and ceramide 42:1 (0 – 10 μ M), were also physiological and selected based on dose ranges in plasma reported in the literature cited^{20, 21, 22}.

In response to the reviewer’s comments we have, therefore, edited the text of our manuscript to make it clearer that experimental concentrations of the lipids used in our study correspond to the physiological concentration range of the lipids **(Page 8, line 158)**:

“DG 32:0 (0 – 10 μ M) (Fig. 2b), LPC 16:0 (0 – 100 μ M) (Fig. 2c) and LPC 18:0 (0 – 50 μ M) (Fig. 2d), at a dose response of physiological concentrations found in plasma^{17,22}, had no significant effect on UPR gene expression. However, treatment of myotubes with ceramides 40:1 (Fig. 2e) and 42:1 (Fig. 2f), at a dose response in the physiological range found in plasma^{22,23,24}, increased expression of Atf4 and Hspa5.”

In addition, the literature cited misses several key publications in the area. For example, an article by S. Choi et al (2018) showing that palmitate increased several long- chain ceramides in intestinal epithelial cells, while a shorter chain fatty acid myristate

resulted in changes in shorter chain ceramides; the study also concentrating on ER stress.

We apologise if we were not clear in presenting the reference highlighted by the reviewer. However, we had included the manuscript by S Choi et al. (FASEBJ 2018) as a reference in the discussion section of the previous iteration of our manuscript:

“Similarly siRNA knockdown of CerS 5 and 6 prevented myristate-induced ER stress, in intestinal epithelial cells³⁵”.

In response to the reviewer’s comment we have now expanded our discussion of this study in the latest version of our manuscript to include additional points related to this work that have been highlighted by the reviewer (**page 18, line 376**):

“Similarly, palmitate induced the production of long chain ceramides, and siRNA knockdown of CerS 5 and 6 prevented myristate-induced ER stress, in intestinal epithelial cells⁴².”

We have also increased our references to the literature throughout the document including a new section on skeletal muscle ceramides in metabolic disease incorporated into our discussion section (**Page 18, line 381**):

“Evidence is emerging that skeletal muscle ceramide metabolism has a key role in the regulation of systemic physiology. Skeletal muscle ceramides are elevated in both patients with, and animal models of, obesity and diabetes and negatively correlate with insulin sensitivity^{44,45,46}. In addition, muscle-specific disruption of ceramide synthesis by CerS1 improves systemic insulin sensitivity in diet induced obese mice⁴⁷.”

Another limitation is that the study focuses on the skeletal muscle only, while discussing the link with metabolic syndrome. The other vital tissues/organs are not covered, and there is also very little discussion on that. This would be important, as the results from the mice indicated larger changes in circulation that were observed in the muscle, thus suggesting that there are other tissues that may have a stronger contribution.

We thank the reviewer for this important point. As the reviewer states our present study focusses on skeletal muscle. The skeletal muscle is a key tissue in systemic metabolic homeostasis²³. More importantly lipotoxicity-induced ER stress in skeletal muscle has been shown to cause metabolic dysfunction and contributes to the development of metabolic disease^{14, 24, 25, 26, 27}. In our introduction we highlight that data in the literature suggests the

presence of a cell non-autonomous mechanism for the propagation of UPR activation and that the mediator of this signal has not yet been identified (**page 3, line 60**):

“However, recent studies suggest that UPR signalling can be propagated in both a paracrine manner and systemically by cell non-autonomous signals^{9,10}. The nature of these extracellular paracrine and endocrine signals remains unknown.”

For these reasons, we hypothesised that cell non-autonomous signalling may be important in the communication of lipotoxicity-induced ER stress in skeletal muscle. We introduce the reasons for concentrating on skeletal muscle in our introduction (**Page 4, line 64**):

“Skeletal muscle is a key regulator of systemic metabolic homeostasis¹¹. Lipotoxicity-induced ER stress in skeletal muscle causes metabolic dysfunction and contributes to the development of metabolic disease^{12,13,14,15,16}. We hypothesised that cell non-autonomous signalling may be important in the communication of lipotoxicity-induced ER stress in skeletal muscle. We demonstrate the presence of lipotoxicity-induced long-chain ceramides secreted in response to lipotoxicity which function as a paracrine signal to activate the UPR in myotubes. We describe the mechanism through which the long-chain ceramide signal is linked to UPR activation, synthesised, transported extracellularly and initiates UPR activation. This study is the first to identify the nature of a cell non-autonomous signal capable of propagating ER stress. We develop our understanding of the control of UPR signalling beyond a cell autonomous state, with clear implications in the development and propagation of metabolic disease”.

However, we fully agree with the reviewer that the mechanisms of cell non-autonomous signalling via ceramides transported in extracellular vesicles, which we identify in skeletal muscle in the present study, may also contribute to lipotoxic mechanisms of cardiometabolic disease in other tissues. We pay particular attention to the discussion of the role of ceramides in lipotoxicity in additional tissues of relevance in our discussion (**Page 18, line 373**):

“Ceramides have been identified as key mechanistic contributors to the lipotoxicity-induced development of adipose tissue and hepatic insulin resistance⁴⁰. Inhibition of ceramide synthesis ameliorates palmitate-induced ER stress in pancreatic β -cells⁴¹. Similarly, palmitate induced the production of long chain ceramides, and siRNA knockdown of CerS 5 and 6 prevented myristate-induced ER stress, in intestinal epithelial cells⁴². A short-chain ceramide analogue, infused into the hypothalamus of rats increases hypothalamic ER stress,

and reduces the thermogenic capacity of brown adipose tissue, thus, highlighting that ceramide-induced ER stress in one tissue can have a metabolic impact on distal tissues⁴³.”

However, we acknowledge that we did not explicitly make the point that mechanisms of cell non-autonomous signalling via ceramides may also contribute to lipotoxic mechanisms of cardiometabolic disease in other tissues. Therefore, in response to the reviewer’s comment we have edited our discussion to further highlight our reasoning for focussing on the skeletal muscle and state the potential that cell non-autonomous signaling via ceramides transported in extracellular vesicles may contribute to lipotoxic mechanisms of cardiometabolic disease in other tissues and state that future work is required to explore this (**page 18, line 389**):

“In the present study we focus on skeletal muscle, as a key regulator of systemic metabolic homeostasis¹¹ where lipotoxicity-induced ER stress causes metabolic dysfunction and contributes to the development of metabolic disease^{12, 13, 14, 15, 16}. This led to our hypothesis that cell non-autonomous signalling may be important in the communication of lipotoxicity-induced ER stress in skeletal muscle. Our work in skeletal muscle may have relevance to the mechanisms of lipotoxicity-induced ER stress in other tissues such as adipose tissue, liver, pancreas and the hypothalamus, where future work may identify that cell non-autonomous ceramide signalling contributes to dysfunction in metabolic disease in these tissues.”

In addition our conclusion regarding the changes in circulating levels of the long chain ceramides in mouse models of metabolic disease do not claim that these are solely contributed to by the skeletal muscle, rather we conclude that (**Page 10, line 203**):

“These data highlight that increases in skeletal muscle and blood plasma long-chain ceramides are associated with metabolic disease in mice and humans.”

In light of the reviewer’s comment we have further clarified that we do not exclude the contribution of other tissues to the circulating long-chain ceramide pool besides skeletal muscle with the addition of the following statement in our discussion section (**Page 17, line 368**):

“Here, we observe an increase in the concentrations of long-chain ceramides 40:1 and 42:1 in the blood plasma and muscle of western diet-induced obese mice and humans with type 2 diabetes. Confirming that ceramides 40:1 and 42:1 are enriched in muscle by metabolic disease in mouse and humans. We do not preclude that circulating concentrations of long-

chain ceramides in this setting may be contributed to by additional tissues besides skeletal muscle.”

In further response to the reviewer’s comment, we have also provided new data strengthening the association between muscle ceramide 40:1 and 42:1 concentration and the plasma concentration of these long chain ceramides. We now show that the muscle and plasma concentrations of ceramide 40:1 and 42:1 are significantly positively correlated in mouse **(Page 9, line 192, Supplementary Figure 5c & 5d):**

“Muscle and plasma concentrations of ceramide 40:1 and 42:1 were significantly positively correlated in the mice (Supplementary Fig 5c & 5d).”

and humans **(Page 10, line 202, Supplementary Figures 5g & 5h):**

“Muscle and plasma concentrations of ceramide 40:1 and 42:1 were significantly positively correlated in the patients (Supplementary Fig 5g & 5h).”

Moreover our decision to focus on skeletal muscle is further supported by the emerging recognition that skeletal muscle ceramides play a role in regulating whole-body physiology including insulin sensitivity². We have now included discussion of this literature in our discussion section to support the relevance of skeletal muscle ceramides to metabolic disease **(Page 18, line 381):**

“Evidence is emerging that skeletal muscle ceramide metabolism has a key role in the regulation of systemic physiology. Skeletal muscle ceramides are elevated in both patients with, and animal models of, obesity and diabetes and negatively correlate with insulin sensitivity^{44, 45, 46}. In addition, muscle-specific disruption of ceramide synthesis by CerSI improves systemic insulin sensitivity in diet induced obese mice⁴⁷.”

Detailed comments:

Results, first chapter: Here, the presentation of the results is not fully clear, and there is a mix of discussion and results. The discussion parts should be moved to the Discussion section.

In response to the reviewer’s comment we have extensively edited the first section of the results for clarity and have removed sections that might be considered discussion **“Pages 5 - 7”**.

The explanation of the myotube differentiation and exposure is not well described, and as it now reads, it sounds that the same experiment was carried out twice. In experimental section it is stated that the differentiation was first carried out for 2 days, after which the myotubes were exposed to palmitate but in the results, it is stated that the differentiation is done for six days and then they were exposed to palmitate.

We apologise for our lack of clarity. We have edited the results section for clarity to better reflect the experiment and the description in the methods section (**page 5, line 83**):

“To confirm UPR induction in response to chronic lipotoxicity in muscle cells, murine C2C12 myotubes were exposed to physiological concentrations of palmitate for the final 6 days of an 8 day differentiation protocol^{17,18}. Cells were treated with either 100 or 200 μ M palmitate conjugated to fatty-acid free bovine serum albumin (BSA), or BSA vehicle control.”

It is also unclear how the palmitate concentrations and the exposure times were chosen; this is crucial information and should be explained in detail.

We apologise for our lack of specificity in the manuscript. In our experiments, we have endeavoured to replicate physiological conditions. We also aim to be translational in our research and use both murine derived C2C12 myotubes and also primary human skeletal muscle cells. The physiological concentration of palmitate in circulation differs between mouse and human. The concentration of palmitate in mouse plasma ranges from 100 – 300 μ M^{5, 6}. Therefore, in experiments using murine C2C12 cells we have opted to remain within this physiological range and use palmitate concentrations of 100 μ M and 200 μ M. However, the plasma concentration of palmitate in humans is lower and ranges from 50 – 150 μ M^{7, 8, 9}. Hence, in experiments using primary human skeletal muscle cells we use palmitate concentrations of 50 μ M and 100 μ M reflective of this physiological range.

Moreover, much of the previous research in this area has investigated the effects of saturated fatty acids at high (often non-physiological) doses in an acute setting (typically no longer than 24 hours of exposure)^{14, 15, 16, 17}. However, physiologically *in vivo* the myotubes within the skeletal muscle will be exposed to circulating palmitate concentrations chronically. In addition, obesity and diabetes are chronic disorders, with sustained increases in circulating plasma free fatty acid concentrations^{18, 19}. Therefore, our study aims to investigate a more physiologically relevant approach using *in vitro* models of chronic palmitate exposure (6 days) at physiological concentrations to better understand the effects of prolonged lipotoxicity on cell non-autonomous UPR activation.

In response to the reviewer's comment, therefore, we have edited the manuscript to reflect this physiological difference between palmitate concentrations in experiments with murine and human cells and to better communicate the chronic nature of the *in vitro* palmitate exposure (**Page 5, line 88**):

“Concentrations of 100 μ M and 200 μ M palmitate reflect the physiological palmitate plasma concentration range in mice (100 – 300 μ M^{19,20}, and chronic treatment (6 days) models continual palmitate exposure in vivo.”

and (**Page 6, line 105**):

“Concentrations of 50 μ M and 100 μ M palmitate reflect the physiological palmitate plasma concentration range in humans (50 – 150 μ M)^{17,18,21}, and chronic treatment (6 days) models continual palmitate exposure in vivo.”

To further clarify the differences when equivalent experiments were performed in both murine-derived C2C12 and human skeletal myocytes, we have labelled figure panels to indicate this.

Figure 1 b vs c, the difference for several genes seem to be larger with lower palmitate concentration, except for Atf4 and EDEM1. Thus, it would be of interest to see the impact at even lower concentration.

We apologise for our lack of clarity. **Figures 1b** and **1c** are not directly comparable and represent two distinct biological systems. We aim to be translational in our research and use both murine derived C2C12 myotubes and also primary human skeletal muscle cells. **Figure 1b** is the effect of transfer of media from C2C12 murine myotubes treated with 200 μ M palmitate on the expression of the UPR markers in naive murine C2C12 cells. In contrast, **Fig 1c** is the effect of transfer of media from 100 μ M palmitate treated human primary skeletal muscle cells on the expression of the UPR markers in naive human primary skeletal myocytes. As discussed in the point above, the different concentrations used reflect the differences in physiological plasma palmitate concentrations between mice and humans. The concentration of palmitate in mouse plasma ranges from 100 – 300 μ M^{5,6}. Therefore, in experiments using murine C2C12 cells we have opted to remain within this physiological range and use a palmitate concentration of 200 μ M in the experiment in **Fig 1b**. However, the plasma concentration of palmitate in humans is lower and ranges from 50 – 150 μ M^{7,8,9}.

Hence, in the experiment shown in **Fig 1c** we use a palmitate concentration of 100 μ M reflective of the human physiological range.

Therefore it would be inappropriate to directly compare dose-dependent effect sizes between the two models. We used both murine and human systems to demonstrate effects in mice translate to human cells and tissue. We have clarified the murine and human nature of **Figure 1b** and **Figure 1c**, respectively, in the text (**Page 5, line 95**):

“To assess a role for cell non-autonomous secreted signals during lipotoxicity-induced cell stress, C2C12 myotubes were exposed to 200 μ M palmitate or the BSA vehicle during differentiation to induce ER stress. Culture media was then changed to serum-free and palmitate-free media for 24 hours. Compounds accumulating in the media would therefore originate from the myotubes. Lipidomic profiling confirmed that the conditioned media was exogenous palmitate-free (Supplementary Fig. 1b). This conditioned media was transferred to naïve differentiated C2C12 myotubes (Fig. 1a). Conditioned media from palmitate treated C2C12 myotubes increased expression of UPR genes Atf4, Hspa5 and Edem1 (Fig. 1b). To determine the relevance of this phenotype in a human model, primary human skeletal muscle cells (HskMCs) were exposed to 50 or 100 μ M palmitate, or the BSA vehicle during a 6 day differentiation. Concentrations of 50 μ M and 100 μ M palmitate reflect the physiological palmitate plasma concentration range in humans (50 – 150 μ M)^{17, 18, 21}, and chronic treatment (6 days) models continual palmitate exposure in vivo. Palmitate treatment increased the expression of UPR genes ATF3, ATF4, HSPA5 and EDEM1 (Supplementary Figure 1c). Conditioned serum- and palmitate-free media collected from HskMCs treated with 100 μ M palmitate or BSA was transferred to naïve HskMCs. Conditioned media from palmitate-treated HskMCs increased UPR gene expression in human myocytes (Fig. 1c). These data suggest the presence of cell non-autonomous UPR-inducing signal secreted by both murine and human myocytes in response to chronic lipotoxicity.”

In addition we have confirmed that palmitate exposure with the distinct murine and human physiological ranges dose-responsively increases expression of UPR genes in the murine C2C12 cells (**Supplementary Fig 1a**) and human primary skeletal muscle cells (**Supplementary Fig 1c**).

We have also edited the legend for **figure 1b & 1c** for clarity (**page 35, line 768**):

“b. Activating transcription factor 3 (Atf3), activating transcription factor 4 (Atf4), Heat Shock Protein Family A (Hsp70) Member 5 (Hspa5) and ER Degradation Enhancing Alpha-

Mannosidase Like Protein 1 (Edem1) UPR gene expression in C2C12 myotubes receiving conditioned media from C2C12 myotubes treated with 200 μ M palmitate or the BSA vehicle.... c. UPR gene expression in primary human skeletal muscle cells (HSkMCs) receiving conditioned media from HSkMCs treated with 100 μ M palmitate or the BSA vehicle...

Furthermore we have added headings to the figure panels for **figure 1b & 1c** to indicate that **figure 1b** is data from C2C12 myotubes and **figure 1c** is data obtained from human primary skeletal muscle cells.

Page 6, lines 104-113: The treatment of the media for protein denaturation: boiling of the media will have a major impact on the metabolites, as many of them are (particularly bioactive lipids) not stable at elevated conditions so this is not most likely a suitable method.

In agreement with the reviewer we recognised the potential for confounding factors with the boiling protein denaturation approach and did not consider it to be convincing enough if used in isolation. Therefore, we opted to also use the biphasic separation approach to partition metabolites and lipids into aqueous and organic soluble fractions (**Fig 1e**). As the reviewer acknowledges below (“Protein precipitation with an organic solvent is better”) and this approach has a distinct advantage.

Nevertheless, we still feel the denaturation experiment adds useful information and since the denatured media retains the ability to induce the UPR in naïve cells this suggests the signal is stable through (short term) boiling, especially as no such signal was observed in control boiled media (**Fig 1d**).

In response to the reviewers comment we have added a caveat to the text (**Page 6, line 119**):

“The UPR-inducing signal was preserved in boiled media conditioned from palmitate treated cells with no evidence of altered UPR-inducing activity in boiled control media. However, boiling denaturation has the potential to destabilize small molecules.”

Moreover, in response to the reviewer’s comment we have performed a new experiment to further characterise the physicochemical nature of the UPR-inducing signal and support the conclusion that the palmitate-induced cell non-autonomous UPR-inducing signal is a small molecule(s). Using molecular weight cut off dialysis of 1000 Da we dialysed conditioned

media from control and palmitate-treated myocytes. The dialysate (containing the dialysed molecules of 1kDa and below) was dried down under N₂, reconstituted in serum-free media and transferred to naïve myocytes. The UPR-inducing signal in conditioned media from palmitate-treated myocytes was observed in the dialysate reconstituted media (**Supplementary Figure 1d**). Therefore the signal is a molecule of 1000 Da or less (**Page 6, line 122**):

“Therefore, to further characterise the physicochemical nature of the UPR-inducing signal, molecular weight cut off dialysis of 1 kDa was used to dialyse conditioned media from control and palmitate-treated myocytes. Dialysate (containing the dialysed molecules of 1kDa and below) was dried down under N₂, reconstituted in serum-free media and transferred to naïve myocytes. The UPR-inducing signal in conditioned media from palmitate-treated myocytes was observed in the dialysate reconstituted media (Supplementary Figure 1d). These data suggest the signal is a molecule of 1 kDa or less.”

Protein precipitation with an organic solvent is better. In the experimental section, it is stated that also the aqueous fraction was used in the exposure studies, but there are no results stated in the results section. Please explain also what was the impact of the aqueous fraction.

We are grateful to the reviewer for their suggestion to investigate the effects of the metabolite-containing aqueous fraction from palmitate-treated myocytes. The mention of the aqueous fraction in the methods section was a typographical error. Nevertheless, following the reviewer’s suggestion we have performed an additional experiment comparing the effect of the metabolite containing aqueous component extracted from conditioned media from palmitate treated myocytes and the metabolite containing aqueous component extracted from conditioned media from control myocytes. These were reconstituted in media and applied to naïve myocytes to investigate their effects on UPR gene expression. We found that the metabolite containing aqueous fraction from the conditioned media of palmitate treated myocytes also induced expression of UPR genes in naïve myocytes (**New Supplementary Fig 1e**). Therefore, in this study, we now also provide data suggesting the presence of an aqueous soluble cell non-autonomous metabolite(s) capable of inducing the UPR.

We have incorporated the new data examining the effect of the metabolite-containing aqueous fraction from conditioned media of palmitate treated myocytes on UPR gene

expression in naïve myocytes into our manuscript (**Page 7, line 137, Supplementary Figure 1e**):

“Next, we examined whether aqueous soluble metabolites may represent additional UPR-inducing signals. The metabolites isolated from the aqueous fractions of conditioned media from both control and palmitate treated myocytes were reconstituted in serum-free media and transferred on to naïve differentiated myotubes. The aqueous metabolite fraction of media from palmitate-treated myotubes also increased expression of UPR target genes compared to the aqueous fraction of media from control myotubes (Supplementary Fig. 1e). These data suggest that both a bioactive lipid(s) and metabolite(s) may function as cell non-autonomous UPR-inducing secreted signals.”

Investigations of the nature, signalling and physiological function of the putative aqueous soluble UPR-inducing signals will be important but do not form the focus of this study and are outside the scope of the current investigation. We consider it a priority to study the putative aqueous soluble metabolite signals in future investigations. In light of our new data suggesting the presence of an aqueous-soluble UPR-inducing metabolite signal we have edited the discussion of our manuscript to acknowledge this (**Page 17, Line 360**):

“However, our work does not preclude the presence of other metabolite factors, protein signals, or bioactive lipids that may contribute to cell non-autonomous UPR activation. Indeed, our data suggests the presence of an, as yet unidentified, aqueous soluble metabolite factor(s) secreted from palmitate-treated myocytes which also activate UPR gene expression.”

Did you consider a more precise separation of the lipid fraction, using e.g. LC based fractionation (e.g.by HILIC)? This would enable the separation of the other species that were showing difference in the overall lipid profiles (DG, LPC), but did not seem to be the main drivers.

We thank the reviewer for their suggestion. We did consider the use of lipid fractionation using a HILIC based method we have previously published which separates the lipid species by class³². However, we found that these methods required the inclusion of mobile phase modifiers (e.g. ammonium formate, ammonium acetate) to facilitate accurate identification and separation to confirm distinct lipid content of fractions, which severely limit use of the separated fractions for downstream cell-based assays of the UPR. These sample matrix problems require extensive clean-up protocols which result in loss of lipid species. This

limitation of the approach was compounded by questions over the purity of fractions since, even with HILIC separation, there is overlap of closely-related species within fractions (e.g. ceramides and dihydroceramides; overlapping lysophosphocholine species). These limitations in the use of liquid-chromatography techniques for downstream bioassay analysis have been reviewed extensively elsewhere³³.

These limitations led us to an alternative approach using genetic manipulation to confirm the identity of the cell non-autonomous UPR-activating lipid signal. To further support our conclusion that long-chain ceramides synthesised and secreted in response to palmitate represent the cell non-autonomous UPR-inducing lipid signal, we have performed an additional experiment to knockdown *CERS2* (the enzyme generated ceramide 40:1 and 42:1) in human primary myocytes using siRNA. We show that genetic depletion of *CERS2* results in inhibition of increased palmitate-induced long chain ceramide concentrations in myocytes and myocyte media (**Page 11, line 233, Figure 4h-j**):

“To confirm long-chain ceramides 40:1 and 42:1 are generated through CERS2 in response to palmitate treatment in myocytes, we decreased CERS2 expression in HSkMCs by 70% using siRNA (Fig 4h). Knockdown of CERS2 decreased intracellular (Fig 4i) and extracellular media (Fig 4j) ceramide 40:1 and 42:1 concentrations and inhibited the palmitate-induced increase in the myocyte media concentration of the lipids.”

We then performed another experiment in which we transferred conditioned media collected from HSkMCs treated with 100 μ M palmitate or BSA control, with and without siRNA-mediated *CERS2* knockdown (depleted media long-chain ceramides), to naïve HSkMCs. We found that the induction of UPR gene expression induced by media conditioned on palmitate treated HSkMCs was inhibited by *CERS2* knockdown confirming that the *CERS2* generated long-chain ceramides are cell non-autonomous UPR-inducing lipid signals (**Page 11, line 240, Fig 4k**):

“We then transferred conditioned serum- and palmitate-free media collected from HSkMCs treated with 100 μ M palmitate or BSA control, with and without siRNA-mediated CERS2 knockdown, to naïve HSkMCs. The induction of UPR gene expression induced by media conditioned on palmitate treated HSkMCs was inhibited by CERS2 knockdown (Fig 4k) in agreement with the effect of CERS2 knockdown on ceramide 40:1 and ceramide 42:1 media concentrations. These data confirm that the CERS2 generated long-chain ceramides are cell non-autonomous UPR-inducing lipid signals.”

Page 6-7: The concentrations of DG32:0 and LPC16:0 and ceramides in the treatment of myotubes, how were these selected? Based on the experimental results?

We apologise for our lack of clarity. The concentrations of the lipid species DG 32:0 (0 – 10 μ M), LPC 16:0 (0 – 100 μ M) and LPC 18:0 (0 – 50 μ M) investigated were physiological and selected based on dose ranges in blood plasma reported in the literature cited^{7,20}.

Likewise the concentrations of the ceramide species investigated, ceramide 40:1 and ceramide 42:1 (0 – 10 μ M), were also physiological and selected based on dose ranges in plasma reported in the literature cited^{20, 21, 22}.

In response to the reviewer's comments we have, therefore, edited the text of our manuscript to make it clearer that experimental concentrations of the lipids used in our study correspond to the physiological concentration range of the lipids (**Page 8, line 158**):

“DG 32:0 (0 – 10 μ M) (Fig. 2b), LPC 16:0 (0 – 100 μ M) (Fig. 2c) and LPC 18:0 (0 – 50 μ M) (Fig. 2d), at a dose response of physiological concentrations found in plasma^{17,22}, had no significant effect on UPR gene expression. However, treatment of myotubes with ceramides 40:1 (Fig. 2e) and 42:1 (Fig. 2f), at a dose response in the physiological range found in plasma^{22,23,24}, increased expression of Atf4 and Hspa5.”

Page 8, Fig 3: The enrichment of the ceramides in murine samples, the difference for murine was larger in serum than in muscle for Cer 40:1, suggesting that this difference is related to something else than changes in the muscle metabolome.

In response to the reviewers comment, we have now provided new data strengthening the association between muscle ceramide 40:1 and 42:1 concentration and the plasma concentration of these long chain ceramides. We now show that the muscle and plasma concentrations of ceramide 40:1 and 42:1 are significantly positively correlated in mouse (**Page 9, line 192, Supplementary Fig 5c & 5d**):

“Muscle and plasma concentrations of ceramide 40:1 and 42:1 were significantly positively correlated in the mice (Supplementary Fig 5c & 5d).”

and in humans (**Page 10, line 202, Supplementary Fig 5g & 5h**):

“Muscle and plasma concentrations of ceramide 40:1 and 42:1 were significantly positively correlated in the patients (Supplementary Fig 5g & 5h).”

Nevertheless, we agree that increased circulating concentrations of the long-chain ceramides in response to western diet in mice and type 2 diabetes in humans may be contributed to by other tissues in addition to skeletal muscle. Our primary conclusion from these data was that a western diet in mice and type 2 diabetes in humans drives an increase in the concentrations of the long-chain ceramides ceramide 40:1 and ceramide 42:1 in plasma and skeletal muscle (**page 10, line 203**):

“These data highlight that increases in skeletal muscle and blood plasma long-chain ceramides are associated with metabolic disease in mice and humans.”

In response to the reviewer’s comment we now explicitly acknowledge that circulating concentrations of the long-chain ceramides in type 2 diabetes in humans and diet-induced obesity in mice may be contributed to by other tissues in addition to skeletal muscle (**page 17, line 368**):

“Here, we observe an increase in the concentrations of long-chain ceramides 40:1 and 42:1 in the blood plasma and muscle of western diet-induced obese mice and humans with type 2 diabetes. Confirming that ceramides 40:1 and 42:1 are enriched in muscle by metabolic disease in mouse and humans. We do not preclude that circulating concentrations of long-chain ceramides in this setting may be contributed to by additional tissues besides skeletal muscle.”

Moreover, as discussed above, our decision to focus on skeletal muscle is further supported by the emerging recognition that skeletal muscle ceramides play a role in regulating whole-body physiology including insulin sensitivity². We have now included discussion of this literature in our discussion section to support the relevance of skeletal muscle ceramides to metabolic disease (**Page 18, line 381**):

“Evidence is emerging that skeletal muscle ceramide metabolism has a key role in the regulation of systemic physiology. Skeletal muscle ceramides are elevated in both patients with, and animal models of, obesity and diabetes and negatively correlate with insulin sensitivity^{44, 45, 46}. In addition, muscle-specific disruption of ceramide synthesis by CerS1 improves systemic insulin sensitivity in diet induced obese mice⁴⁷.”

In humans, no results are shown for blood-based samples, this should be added in the results.

Responding to the reviewer's request for additional data, and following amendments to our ethical approval, we have generated new data examining the concentration of ceramide 40:1 and ceramide 42:1 in the blood plasma of a subset of our patients with type 2 diabetes and control volunteers. As in skeletal muscle (and muscle and plasma of mice), the concentration of ceramide 40:1 and ceramide 42:1 were greater in patients with type 2 diabetes compared with control volunteers (**page 10, line 200, Figure 3d & Supplementary Fig 5f**):

“Lipidomic analysis of the blood plasma from a random subset of patients identified increased concentrations of ceramides 40:1 and 42:1 in people with type 2 diabetes (Fig. 3d, Supplementary Figure 5f).”

In humans, the difference in the muscle does not seem to be significant?

We are grateful for the reviewer's comments regarding the statistical significance of the differences in muscle long-chain ceramides between control volunteers and patients with type 2 diabetes. This was an error of omission for which we apologise and have now corrected **Figure 3c** to include the significance indicators. In addition, we have edited the way in which we display the data and now use box and whisker diagrams to make the differences in the groups clearer.

Page 11-12, analysis of ceramides and dihydroceramides. How did you verify the identity of the dihydroceramides?)Overall, more information on the accuracy of the lipidomic measurements should be given here.

We apologise for our lack of detail regarding the lipidomic methods. We have added extensive details of our use of tandem mass spectrometry fragmentation and commercially available standards to assign and identify lipid species. The identification of the key lipid species discussed in our manuscript (LPC 16:0, LPC 18:0, DG 32:0, ceramide 40:1 (N-(docosanoyl)-ceramide; ceramide d18:1/22:0) and ceramide 42:1 (N-(tetracosanoyl)-ceramide; ceramide d18:1/24:0, dihydroceramide 42:0 and dihydroceramide 40:0) were performed to Metabolomics Standards Initiative level one identification criteria using commercially available standards^{34,35} (**Page 29, line 632**):

“To overcome limitations in LC-MS annotation of isobaric lipid species, and to facilitate the identification of the fatty acyl chains esterified to complex lipids, a tandem mass spectrometry fragmentation method was used in a second scan event. Ions were fragmented using collision induced dissociation at a normalised collision energy of 35. Precursor ions were selected from a mass list, with the most intense ion on the list fragmented in each scan. The minimum signal required was set to 5000 counts and the isolation width was set to 1. The activation time was 10 ms and the activation Q was set to 0.25. Fragmentation spectra were acquired in centroid mode at a resolution of 15,000 using the FTMS analyser. The subsequent fragmentation spectra were used to confirm the identity of the lipid assignments. Assignment of lipid identities for select species were also confirmed by comparison with commercially available standards for LPC 16:0 (855675), LPC 18:0 (855775), DG 32:0 (800816), ceramide 40:1 (N-(docosanoyl)-ceramide; Ceramide d18:1/22:0) (860501) and ceramide 42:1 (N-(tetracosanoyl)-ceramide; Ceramide d18:1/24:0) (860524), dihydroceramide 42:0 (860628) (Avanti Polar Lipids Inc) and dihydroceramide 40:0 (sc-210991) (Santa Cruz Biotechnology) and therefore meet Metabolomics Standards Initiative level one identification criteria^{34, 35}.”

Discussion:

Here, the discussion is focusing on the ceramides, however, in the current study the focus was only on the muscle tissue. Liver is one of the key organs in lipid homeostasis and it is tightly linked with the metabolic diseases. Why was to focus solely on skeletal muscle? What is the potential contribution of the other tissues , particularly liver?

As discussed above our present study focusses on skeletal muscle. The skeletal muscle is a key tissue in systemic metabolic homeostasis²³. More importantly lipotoxicity-induced ER stress in skeletal muscle has been shown to cause metabolic dysfunction and contributes to the development of metabolic disease^{14, 24, 25, 26, 27}. In our introduction we highlight that data in the literature suggests the presence of a cell non-autonomous mechanism for the propagation of UPR activation and that the mediator of this signal has not yet been identified **(page 3, line 61):**

“..UPR signalling can be propagated in both a paracrine manner and systemically by cell non-autonomous signals^{31, 36}. The nature of these extracellular paracrine and endocrine signals remains unknown.”

For these reasons we hypothesised that cell non-autonomous signalling may be important in the communication of lipotoxicity-induced ER stress in skeletal muscle. We introduce the reasons for concentrating on skeletal muscle in our introduction (**Page 4, line 64**):

“Skeletal muscle is a key regulator of systemic metabolic homeostasis²³. Lipotoxicity-induced ER stress in skeletal muscle causes metabolic dysfunction and contributes to the development of metabolic disease^{14, 24, 25, 26, 27}. We hypothesised that cell non-autonomous signalling may be important in the communication of lipotoxicity-induced ER stress in skeletal muscle. We demonstrate the presence of lipotoxicity-induced long-chain ceramides secreted in response to lipotoxicity which function as a paracrine signal to activate the UPR in myotubes....”

However, we fully agree with the reviewer that the mechanisms of cell non-autonomous signalling via ceramides transported in extracellular vesicles, which we identify in skeletal muscle in the present study, may also contribute to lipotoxic mechanisms of cardiometabolic disease in other tissues. We pay particular attention to additional tissues of relevance in our discussion (**Page 18, Line 373**):

“Ceramide have been identified as key mechanistic contributors to the lipotoxicity-induced development of adipose tissue and hepatic insulin resistance³⁷. Inhibition of ceramide synthesis ameliorates palmitate-induced ER stress in pancreatic β -cells³⁸. Similarly, palmitate induced the production of long chain ceramides, and siRNA knockdown of CerS 5 and 6 prevented myristate-induced ER stress, in intestinal epithelial cells³⁹. A short-chain ceramide analogue, infused into the hypothalamus of rats increases hypothalamic ER stress, and reduces the thermogenic capacity of brown adipose tissue, thus, highlighting that ceramide-induced ER stress in one tissue can have a metabolic impact on distal tissues⁴⁰.”

However, we acknowledge that we did not explicitly make the point that mechanisms of non-cell autonomous signalling via ceramides may also contribute to lipotoxic mechanisms of cardiometabolic disease in other tissues. Therefore, in response to the reviewer's comment we have edited our discussion to further highlight our reasoning for focussing on the skeletal muscle and state the potential that non-cell autonomous signalling via ceramides transported in extracellular vesicles may contribute to lipotoxic mechanisms of cardiometabolic disease in other tissues and that future work is required to explore this (**page 18, line 389**):

“In the present study we focus on skeletal muscle, as a key regulator of systemic metabolic homeostasis²³ where lipotoxicity-induced ER stress causes metabolic dysfunction and contributes to the development of metabolic disease^{14, 24, 25, 26, 27}. This led to our hypothesis

that cell non-autonomous signalling may be important in the communication of lipotoxicity-induced ER stress in skeletal muscle. Our work in skeletal muscle may have relevance to the mechanisms of lipotoxicity induced ER stress in other tissues such as adipose tissue, liver, pancreas and the hypothalamus where future work may identify that non-cell autonomous ceramide signalling contributes to dysfunction in metabolic disease in these tissues.”

Moreover our conclusion regarding the changes in circulating levels of the long chain ceramides in mouse models of metabolic disease do not claim that these are solely contributed to by the skeletal muscle. Instead we conclude that (**Page 10, line 203**):

“These data highlight that increases in skeletal muscle and blood plasma long-chain ceramides are associated with metabolic disease in mice and humans.”

In light of the reviewer’s comment we have further clarified this with the addition of the following statement in our discussion section (**Page 17, Line 368**):

“Here, we observe an increase in the concentrations of long-chain ceramides 40:1 and 42:1 in the blood plasma and muscle of western diet-induced obese mice and humans with type 2 diabetes. Confirming that ceramides 40:1 and 42:1 are enriched in muscle by metabolic disease in mouse and humans. We do not preclude that circulating concentrations of long-chain ceramides in this setting may be contributed to by additional tissues besides skeletal muscle.”

As previously outlined above, in response to the reviewer’s comment, we have also provided new data strengthening the association between muscle ceramide 40:1 and 42:1 concentration and the plasma concentration of these long chain ceramides. We now show that the muscle and plasma concentrations of ceramide 40:1 and 42:1 are significantly positively correlated in mouse (**Page 9, line 192, Supplementary Fig 5c & 5d**):

“Muscle and plasma concentrations of ceramide 40:1 and 42:1 were significantly positively correlated in the mice (Supplementary Fig 5c & 5d).”

and in humans (**Page 10, line 202, Supplementary Fig 5g & 5h**):

“Muscle and plasma concentrations of ceramide 40:1 and 42:1 were significantly positively correlated in the patients (Supplementary Fig 5g & 5h).”

Moreover, again as discussed above, our decision to focus on skeletal muscle is further supported by the emerging recognition that skeletal muscle ceramides play a role in regulating whole-body physiology including insulin sensitivity². We now include discussion of this literature in our discussion section to highlight the relevance of skeletal muscle ceramides to metabolic disease (**Page 18, line 381**):

“Evidence is emerging that skeletal muscle ceramide metabolism has a key role in the regulation of systemic physiology. Skeletal muscle ceramides are elevated in both patients with, and animal models of, obesity and diabetes and negatively correlate with insulin sensitivity^{44, 45, 46}. In addition, muscle-specific disruption of ceramide synthesis by CerSI improves systemic insulin sensitivity in diet induced obese mice⁴⁷.”

Page 11, line 217-218: The authors state that the partition of bioactive lipids into EVs has not been previously reported. This statement is not very clear. There are several studies of the lipid composition of the EVs, and the difference in the composition e.g. in cancer cells.

We thank the reviewer for highlighting our oversight. In light of the reviewer’s comments we now include citations to recent articles on the topic^{12, 13} and have edited this line to read (**Page 13, line 291**):

“There is developing recognition that bioactive lipid signals may be partitioned into EVs^{30, 31, 32}”

Page 15, line 309-3010: The authors state that the dihydroceramides have been linked with gastric carcinoma cells. There are other studies as well, linking these compounds with ER stress, e.g. in glioblastomas

We thank the reviewer for emphasising the importance of dihydroceramides and ER stress in cancer. In light of the reviewers comment we have edited our statement to be more generally applicable to cancer and included additional citations of work related to dihydroceramides and ER stress in gastric carcinoma and glioblastoma (**Page 20, line 422**):

“Although a role for dihydroceramide accumulation in ER stress has been reported in several cancers including gastric carcinoma cells and glioblastoma^{33, 53, 54}”

Supplementary Figure 2. The text of lipid species is too small to be readable

We agree with the reviewer that this is an ineffective and impractical display format and have edited the figure to remove this feature.

1. Bickert A, *et al.* Inactivation of ceramide synthase 2 catalytic activity in mice affects transcription of genes involved in lipid metabolism and cell division. *Biochim Biophys Acta Mol Cell Biol Lipids* **1863**, 734-749 (2018).
2. Turpin-Nolan SM, *et al.* CerS1-Derived C18:0 Ceramide in Skeletal Muscle Promotes Obesity-Induced Insulin Resistance. *Cell Rep* **26**, 1-10 e17 (2019).
3. Novgorodov SA, Gudz TI, Obeid LM. Long-chain ceramide is a potent inhibitor of the mitochondrial permeability transition pore. *J Biol Chem* **283**, 24707-24717 (2008).
4. Wijesinghe DS, *et al.* Chain length specificity for activation of cPLA2alpha by C1P: use of the dodecane delivery system to determine lipid-specific effects. *J Lipid Res* **50**, 1986-1995 (2009).
5. Eguchi K, *et al.* Saturated fatty acid and TLR signaling link beta cell dysfunction and islet inflammation. *Cell Metab* **15**, 518-533 (2012).
6. Takahashi H, *et al.* Long-chain free fatty acid profiling analysis by liquid chromatography-mass spectrometry in mouse treated with peroxisome proliferator-activated receptor alpha agonist. *Biosci Biotechnol Biochem* **77**, 2288-2293 (2013).
7. Psychogios N, *et al.* The human serum metabolome. *PLoS One* **6**, e16957 (2011).
8. Wishart DS, *et al.* HMDB: a knowledgebase for the human metabolome. *Nucleic Acids Res* **37**, D603-610 (2009).

9. Lopes SM, Trimbo SL, Mascioli EA, Blackburn GL. Human plasma fatty acid variations and how they are related to dietary intake. *Am J Clin Nutr* **53**, 628-637 (1991).
10. Laviad EL, *et al.* Characterization of ceramide synthase 2: tissue distribution, substrate specificity, and inhibition by sphingosine 1-phosphate. *J Biol Chem* **283**, 5677-5684 (2008).
11. Riebeling C, Allegood JC, Wang E, Merrill AH, Jr., Futerman AH. Two mammalian longevity assurance gene (LAG1) family members, trh1 and trh4, regulate dihydroceramide synthesis using different fatty acyl-CoA donors. *J Biol Chem* **278**, 43452-43459 (2003).
12. Skotland T, Sagini K, Sandvig K, Llorente A. An emerging focus on lipids in extracellular vesicles. *Adv Drug Deliv Rev* **159**, 308-321 (2020).
13. Boilard E. Extracellular vesicles and their content in bioactive lipid mediators: more than a sack of microRNA. *J Lipid Res* **59**, 2037-2046 (2018).
14. Salvado L, *et al.* Oleate prevents saturated-fatty-acid-induced ER stress, inflammation and insulin resistance in skeletal muscle cells through an AMPK-dependent mechanism. *Diabetologia* **56**, 1372-1382 (2013).
15. Ozcan U, *et al.* Endoplasmic reticulum stress links obesity, insulin action, and type 2 diabetes. *Science* **306**, 457-461 (2004).
16. Panzhinskiy E, Hua Y, Culver B, Ren J, Nair S. Endoplasmic reticulum stress upregulates protein tyrosine phosphatase 1B and impairs glucose uptake in cultured myotubes. *Diabetologia* **56**, 598-607 (2013).
17. Hage Hassan R, *et al.* Endoplasmic reticulum stress does not mediate palmitate-induced insulin resistance in mouse and human muscle cells. *Diabetologia* **55**, 204-214 (2012).

18. Opie LH, Walfish PG. Plasma free fatty acid concentrations in obesity. *N Engl J Med* **268**, 757-760 (1963).
19. Boden G. Free fatty acids, insulin resistance, and type 2 diabetes mellitus. *Proc Assoc Am Physicians* **111**, 241-248 (1999).
20. Quehenberger O, *et al.* Lipidomics reveals a remarkable diversity of lipids in human plasma. *J Lipid Res* **51**, 3299-3305 (2010).
21. Basit A, Piomelli D, Armirotti A. Rapid evaluation of 25 key sphingolipids and phosphosphingolipids in human plasma by LC-MS/MS. *Anal Bioanal Chem* **407**, 5189-5198 (2015).
22. Kasumov T, *et al.* Ceramide as a mediator of non-alcoholic Fatty liver disease and associated atherosclerosis. *PLoS One* **10**, e0126910 (2015).
23. Baskin KK, Winders BR, Olson EN. Muscle as a "mediator" of systemic metabolism. *Cell Metab* **21**, 237-248 (2015).
24. Peter A, *et al.* Individual stearyl-coa desaturase 1 expression modulates endoplasmic reticulum stress and inflammation in human myotubes and is associated with skeletal muscle lipid storage and insulin sensitivity in vivo. *Diabetes* **58**, 1757-1765 (2009).
25. Deldicque L, *et al.* The unfolded protein response is activated in skeletal muscle by high-fat feeding: potential role in the downregulation of protein synthesis. *Am J Physiol Endocrinol Metab* **299**, E695-705 (2010).
26. Kars M, *et al.* Tauroursodeoxycholic Acid may improve liver and muscle but not adipose tissue insulin sensitivity in obese men and women. *Diabetes* **59**, 1899-1905 (2010).

27. Salvado L, *et al.* PPARbeta/delta prevents endoplasmic reticulum stress-associated inflammation and insulin resistance in skeletal muscle cells through an AMPK-dependent mechanism. *Diabetologia* **57**, 2126-2135 (2014).
28. Boon J, *et al.* Ceramides contained in LDL are elevated in type 2 diabetes and promote inflammation and skeletal muscle insulin resistance. *Diabetes* **62**, 401-410 (2013).
29. Strackowski M, *et al.* Relationship between insulin sensitivity and sphingomyelin signaling pathway in human skeletal muscle. *Diabetes* **53**, 1215-1221 (2004).
30. Adams JM, 2nd, *et al.* Ceramide content is increased in skeletal muscle from obese insulin-resistant humans. *Diabetes* **53**, 25-31 (2004).
31. Schinzel R, Dillin A. Endocrine aspects of organelle stress-cell non-autonomous signaling of mitochondria and the ER. *Curr Opin Cell Biol* **33**, 102-110 (2015).
32. Rohwedder A, Knipp S, Roberts LD, Ladbury JE. Composition of receptor tyrosine kinase-mediated lipid micro-domains controlled by adaptor protein interaction. *Sci Rep* **11**, 6160 (2021).
33. Weller MG. A unifying review of bioassay-guided fractionation, effect-directed analysis and related techniques. *Sensors (Basel)* **12**, 9181-9209 (2012).
34. Members MSIB, *et al.* The metabolomics standards initiative. *Nat Biotechnol* **25**, 846-848 (2007).
35. Sumner LW, *et al.* Proposed minimum reporting standards for chemical analysis Chemical Analysis Working Group (CAWG) Metabolomics Standards Initiative (MSI). *Metabolomics* **3**, 211-221 (2007).

36. Mahadevan NR, Rodvold J, Sepulveda H, Rossi S, Drew AF, Zanetti M. Transmission of endoplasmic reticulum stress and pro-inflammation from tumor cells to myeloid cells. *Proc Natl Acad Sci U S A* **108**, 6561-6566 (2011).
37. Chaurasia B, *et al.* Targeting a ceramide double bond improves insulin resistance and hepatic steatosis. *Science* **365**, 386-392 (2019).
38. Boslem E, *et al.* A lipidomic screen of palmitate-treated MIN6 beta-cells links sphingolipid metabolites with endoplasmic reticulum (ER) stress and impaired protein trafficking. *Biochem J* **435**, 267-276 (2011).
39. Choi S, *et al.* Myristate-induced endoplasmic reticulum stress requires ceramide synthases 5/6 and generation of C14-ceramide in intestinal epithelial cells. *FASEB J* **32**, 5724-5736 (2018).
40. Contreras C, *et al.* Central ceramide-induced hypothalamic lipotoxicity and ER stress regulate energy balance. *Cell Rep* **9**, 366-377 (2014).

Reviewers' Comments:

Reviewer #1:

Remarks to the Author:

The authors have done a good job addressing my comments. The manuscript is much improved.

Reviewer #2:

Remarks to the Author:

The authors revised their manuscript as according to the comments and now the manuscript becomes web balanced and convinced.

Reviewer #3:

Remarks to the Author:

The authors have significantly improved the manuscript, and have added data from novel experiments. They have also provided with a detailed response to the comments. The study now is based on detailed data on both cellular studies, accompanied with data on mice and humans. Overall, a very well-done study. The results on the aqueous fraction are also interesting and while the current data is more than enough for the current manuscript, I hope that the author follow-up that data as well in future work.

Response to Reviewers' Comments

Reviewer #1 (Remarks to the Author):

The authors have done a good job addressing my comments. The manuscript is much improved.

We thank the author for their time and suggestions in helping us to improve our manuscript.

Reviewer #2 (Remarks to the Author):

The authors revised their manuscript as according to the comments and now the manuscript becomes web balanced and convinced.

We thank the author for their suggestions and input which has helped our manuscript become well balanced and convincing.

Reviewer #3 (Remarks to the Author):

The authors have significantly improved the manuscript, and have added data from novel experiments. They have also provided with a detailed response to the comments. The study now is based on detailed data on both cellular studies, accompanied with data on mice and humans. Overall, a very well-done study. The results on the aqueous fraction are also interesting and while the current data is more than enough for the current manuscript, I hope that the author follow-up that data as well in future work.

We thank the reviewer for their contribution to improving our manuscript and their kind description of a “very well-done study”. Although beyond the scope of the current manuscript, as recognised by the reviewer, it is our intention to perform future studies investigating the aqueous factor inducing cell non-autonomous UPR activation.